# `HeavyWater` and `SimplexWater`: Distortion-free LLM Watermarks for Low-Entropy Distributions

**Dor Tsur**[*]
Ben Gurion University

**Carol Xuan Long**[*]
Harvard University

**Claudio Mayrink Verdun**
Harvard University

**Hsiang Hsu**
JPMorgan Chase

**Richard Chen**
JPMorgan Chase

**Haim Permuter**
Ben Gurion University

**Sajani Vithana**[†]
Harvard University

**Flavio P. Calmon**[†]
Harvard University

## Abstract

Large language model (LLM) watermarks enable authentication of text provenance, curb misuse of machine-generated text, and promote trust in AI systems. Current watermarks operate by changing the next-token predictions output by an LLM. The updated (i.e., watermarked) predictions depend on random side information produced, for example, by hashing previously generated tokens. LLM watermarking is particularly challenging when next-token predictions are near-deterministic. In fact, over 90% of next-token distributions are low-entropy, with more than half of the probability mass on a single token. In this paper, we propose an optimization framework for watermark design for this regime. Our goal is to understand how to most effectively use random side information to optimize the detection-quality trade-off. Our analysis informs the design of two new watermarks: `HeavyWater` and `SimplexWater`. Both watermarks are tunable, gracefully trading-off between detection accuracy and text distortion. They can also be applied to any LLM and are agnostic to side information generation. We evaluate the performance of `HeavyWater` and `SimplexWater` across several benchmarks, demonstrating that they achieve superior detection with minimal degradation in downstream quality across generation tasks. Our theoretical analysis also reveals surprising new connections between LLM watermarking and coding theory.

## 1 Introduction

Watermarking large language models (LLMs) consists of embedding a signal into the text generation process that allows reliable detection of machine-generated text. Over the past two years, there have been increasing calls for LLM watermarking by both policymakers [1–3] and industry [4]. Watermarks enable authentication of text provenance [5], promote trust in AI systems [6], and can address copyright and plagiarism issues [7, 8]. Ideally, watermarks should be detectable directly from text without access to the underlying LLM. Watermarks should also be tamper-resistant – i.e., robust to minor edits or paraphrasing of watermarked text [9] – and incur little degradation in text quality.

LLMs are watermarked by changing their next-token distributions according to random side information (see Fig. 2 for a visualization). Side information is typically generated by hashing previous tokens and secret keys, then using the hash to seed a random number generator to produce a sample $s$ [10–12]. The side information $s$ is then used to change the distribution from which the next token $x$ is sampled. Watermarks are detected by mapping a token $x$ and corresponding side information $s$ to a score $f(x, s)$. By averaging scores across a sequence of tokens, we can perform statistical tests

---

[*]Equal contributions.
[†]Equal contributing senior authors.

39th Conference on Neural Information Processing Systems (NeurIPS 2025).

to decide if the text is watermarked. Watermarking is particularly challenging when the next-token distribution has low entropy, as is often the case in tasks such as code generation [13].

**Our goal** *is to design watermarks that optimally use side information to maximize detection accuracy and minimize distortion of generated text.* Our starting point is a minimax optimization framework for watermark design. This framework allows us to jointly optimize how side information is embedded into next-token distributions and the score function used for watermark detection. When restricting the optimization to binary-valued scores, we uncover a surprising new connection between LLM watermarking and coding theory: *designing minimax-optimal watermarks is equivalent to constructing codes with large Hamming distance between codewords.* We use this finding to design a new watermark called `SimplexWater` based on Simplex codes [14]. In the low-entropy regime, `SimplexWater` is optimal across binary watermarks and empirically outperforms competing methods that use binary scores (cf. Red-Green [15], and Correlated Channel [16] in Fig. 1).

Next, we relax the binary restriction on the score function. We consider score functions whose outputs are independently and identically (i.i.d.) drawn from a continuous distribution prior to the start of watermarking. Interestingly, we show that the Gumbel watermark [17] is a specific instantiation of our framework when scores are drawn from a Gumbel distribution. We prove that, in the low-entropy regime, watermark detection depends on the weight of the tail of the distribution from which scores are sampled. This observation leads to `HeavyWater`, whose scores are drawn from a heavy-tailed distribution. `HeavyWater` outperforms competing state-of-the-art watermarks in terms of detection-distortion trade-offs (cf. Gumbel [17] in Fig. 1).

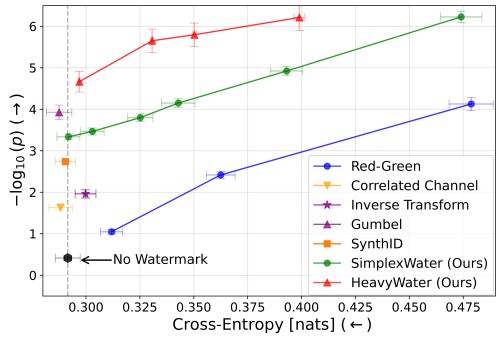

**Figure 1:** `HeavyWater` and `SimplexWater` demonstrate favorable detection performance (measured by p-values) with minimal distortion to the base unwatermarked model (measured by Cross-Entropy). See Section 5 for details.

In practice, `SimplexWater` and `HeavyWater` are implemented by solving an optimal transport (OT) problem [18] that maximizes the average score across all couplings between the side information and the next-token distributions. We efficiently solve the OT problem using Sinkhorn's algorithm [19]. Since the OT preserves the average next-token distribution, both watermarks are *distortion-free*, and a user without knowledge of the side information would not perceive an average change in generated text. `SimplexWater` and `HeavyWater` are also *tunable*: we provide a simple scheme to further increase watermark detection accuracy at a small distortion cost by upweighting high-score tokens.

Our **main contributions** are:

- We introduce `SimplexWater`, a binary score watermark rooted in coding theory, and derive optimality guarantees across all binary-score watermarks.

- We analyze watermarks that use randomly generated score functions drawn from a continuous distribution, and show that the Gumbel watermark [17] is a special case of this construction.

- We prove that the detection power of watermarks with randomly generated score functions depends on the tail of the score function. This leads to `HeavyWater`, a new watermark whose scores are randomly drawn from a heavy-tailed distribution.

- We demonstrate the favorable performance of `HeavyWater` and `SimplexWater` across an array of models, datasets, and tasks, relative to state-of-the-art watermarks.

**Related Works**

We discuss the related work most closely related to ours next, and provide a broader survey of the watermarking literature in Appendix A.1. The first LLM watermark was proposed in [15] – referred here as the Red-Green watermark. This method partitions the token vocabulary into two lists, which are then used to reweigh the token distribution via exponential tilting. The Red-Green watermark was extended via different random side information generation schemes in [10]. Watermarking has since been extensively studied, with more recent work introducing new sampling schemes to improve

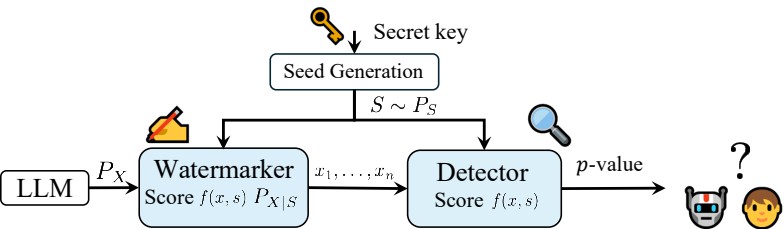

**Figure 2:** Visualization of the components of watermarking design.

watermarking [20] and multi-draft watermark generation methods [21]. The use of threshold tests over average scores to detect watermarks – a common paradigm in the literature, a design decision we make here – was questioned in [22, 23]. [23], in particular, shows that more sophisticated statistical tests can boost watermark detection. We note that our watermarks can be paired with multi-draft methods such as [21] to boost detection, though we do not explore this here.

**Distortion-free Watermarking.** In watermarking, distortion quantifies how much a watermark distribution differs from the original LLM output and is a proxy for textual quality. [17] uses the Gumbel-max trick to design a distortion-free watermark. This scheme was relaxed to a soft reweigh of the next-token distribution [24] following the idea from the Red-Green watermark. The Gumbel watermark achieves excellent performance in our benchmarks, though it is outperformed by `HeavyWater`. Another popular distortion-free watermark is [13], which uses inverse transform sampling. SynthID [11], in turn, produces watermarked tokens via a strategy called tournament sampling. This watermark was extensively evaluated in real-world user tests, and we also select it as a competing benchmark. More recently, [16] proposed a binary-score watermark based on partitioning the token vocabulary into an arbitrary number of sets, followed by a simple binary test for watermark detection. The method in [16] is simple to implement and incurs little runtime overhead. However, it underperformed both `HeavyWater` and `SimplexWater` in our benchmarks. Additional distortion-free watermarks include [25], which combines a surrogate model with the Gumbel-max trick to boost watermark detection.

**Low-Entropy Watermarks.** Several schemes were proposed to address the challenge of watermarking low-entropy distributions [10, 26]. [27] proposes a resampling method that is applied to adapt the Red-Green watermark to low-entropy distributions, while [28] turns to semantic sentence-level watermarks. [29] uses a logit-free method that injects the texts with words from an input-dependent set, and [30] weights the tokens' scores according to their entropy. We share a similar design goal of optimizing the watermark for the low-entropy regime.

In practice, side information will not be perfectly random: it will depend on previous tokens and a secret shared key through various hashing schemes [15, 31]. Hashing schemes range from simple `max`/`min` operations to elaborate adaptive context windows [11] and semantic hashing [32]. We view side information generation as a separate yet no less important problem. Instead, we focus on *how to optimally use* side information, and we develop a principled and theory-guided approach for watermark design.

## 2  LLM Watermarking Framework

In this section, we outline an optimization formulation for LLM watermarking. We denote random variables using capital letters (e.g., $X$ and $S$), their realizations by lower-case letters (e.g., $x$ and $s$), and their distributions by sub-scripted $P$ or $Q$ (e.g., $P_X$ and $P_S$). We consider an LLM with token vocabulary $\mathcal{X} = [1:m] \triangleq \{1, \ldots, m\}$. During generation, the $t$-th token is drawn auto-regressively from a distribution $Q_{X_t|X_1,\ldots,X_{t-1}}$ with support in $\mathcal{X}$. We assume watermarking is performed on a per-token basis, so we denote $P_X = Q_{X_t|X_1,\ldots,X_{t-1}}$ for simplicity.

Consider two parties: the *watermarker* and the *detector* (see Fig. 2). For each token generated by the model, both parties share random side information, represented by a random variable $S \sim P_S$ over a discrete alphabet $\mathcal{S} = [1:k]$. The watermarker has access to the next-token distribution $P_X$. Their goal is to change $P_X$ using the side information $S$. The detector, in turn, identifies the watermark from a sequence of observed tokens and side information pairs $(X, S)$ and does not know $P_X$. We divide the watermarking procedure into three main components.

The first component is **random side information generation**. Both the watermarker and the detector must produce the same side information $S$. In practice, this is accomplished by employing hashing strategies: previous tokens and a secret key are combined and hashed to produce a seed for a random number generator [11, 15]. This generator then serves as the distribution $P_S$ from which side information $S$ is sampled. Popular hashing strategies consider min, max or sum operations over some finite context window of previously generated tokens. Note that both the watermarker and the detector can produce the side information since it only depends on previously generated text and shared keys.

The second component deals with **how randomness is used**. We begin by designing a score function $f : \mathcal{X} \times \mathcal{S} \to \mathbb{R}$ that maps a token $x$ and corresponding side information $s$ onto a score $f(x, s)$. The score function $f$ is fixed and known to both the watermarker and the detector. In the watermarking stage, we use $f$ and the shared randomness $s$ to change the next-token distribution. Specifically, the watermarker samples from the conditional distribution $P_{X|S=s}$ instead of $P_X$. The distribution $P_{X|S=s}$ is designed to increase the probability of tokens with high scores $f(\cdot, s)$. We call the altered distribution $P_{X|S=s}$ the *watermarked distribution*.

The last component is **watermark detection**. In the detection stage, the detector receives a sequence of tokens and side information realizations $\{(x_t, s_t)\}_{t=1}^n$, from which they compute a sequence of scores $\{f(x_t, s_t)\}_{t=1}^n$. We assume the sequence of tokens is declared as watermarked if the averaged score $\frac{1}{n} \sum_{t=1}^n f(x_t, s_t) \geq \tau$, where $\tau$ is a threshold that trades off between specificity and sensitivity of watermark detection. See Figure 2 for a visualization of the watermarking procedure.

Most existing watermarks can be instantiated in terms of the three components above. For example[-1], in the Red-Green watermark [15], $S$ is a sequence of $m$ bits ($k = 2^m$), assigning one bit per token. A realization $s$ of side information is used to partition $\mathcal{X}$ into two lists, corresponding to the entries that are drawn as 1 or 0. Tokens marked as 1 ("green list") are upweighted, and tokens marked as 0 ("red list") are downweighted, resulting in the watermarked distribution $P_{X|S=s}$ from which the next token is sampled. The score function is binary: $f(x, s) = 1$ if $x$ is in the green list determined by $s$, or 0 otherwise. Detection is done by averaging binary scores and performing a threshold test.

**An Optimization Formulation for Watermarking.**    Given the framework above, we present an optimization formulation from which we derive and analyze `SimplexWater` and `HeavyWater`. Our focus is on optimizing the watermarked distribution $P_{X|S}$ and the score function $f(x, s)$ when $P_S$ is the uniform distribution and we generate a sample $S \sim P_S$ for each token. We also assume watermark detection is based on thresholding the average score $\frac{1}{n} \sum_{t=1}^n f(x_t, s_t)$ which, in turn, acts as a proxy for $\mathbb{E}[f(X, S)]$. When the text is watermarked, this expectation is with respect to $P_{XS} = P_{X|S} P_S$, and otherwise, it is taken with respect to $P_X P_S$.[0] Our goal is to design a pair $(f, P_{X|S})$ that maximizes the gap between average score when text is watermarked relative to when the text is not watermarked.

Watermark performance will inevitably depend on the entropy of the next-token distribution $P_X$ [10, 13]. We focus on low-entropy distributions, whose watermarking is considered to be challenging [30]. We adopt the low-entropy constraint given the following definition.

**Definition 1** (Low-entropy distributions)**.** *We use* min-entropy *as our measure of uncertainty for the next-token distribution $P_X$, defined as*

$$H_{min}(P_X) = -\log_2 \max_x P_X(x).$$

*A distribution $P_X$ is said to be* low-entropy *if its min-entropy satisfies $H_{min}(P_X) \leq 1$ bit/token, i.e., at least one token has probability mass greater than $\frac{1}{2}$.*

**Remark 1** (Watermarking in Low-Entropy Regime)**.** *Our theoretical guarantees imply favorable watermarking performance in low-entropy scenarios. The 'low-entropy' constraint used in our analysis is not merely a theoretical assumption. It is based on the empirical observation that LLMs' next-token distributions are **inherently low entropy**. Even though coding, Q&A, and summarization tasks are commonly understood as having varying degrees of 'entropy' – with coding assumed to have lower entropy – we still observe $> 90\%$ of next-token predictions across these tasks falling*

---

[-1]See Appendix A.2 in which we instantiate additional watermarks within the proposed setting.

[0]Observe that $P_X P_S$ corresponds to the null hypothesis (non-watermarked text): the generated tokens are independent of the side information $S$.

*well within the low-entropy considered in our theoretical analysis for practical temperature values (see Appendix C.1 Figures D.14 and D.15). Refer to [33, Page 13] and [34, Page 4] for a similar discussion.*

Given the aforementioned assumptions, the optimal score function $f$ and the joint distribution $P_{XS}$ that maximize detection performance for the worst-case distribution in $\mathcal{P}_\lambda$ are the solution of the optimization problem

$$\mathsf{D}_{\mathsf{gap}}(m, k, \lambda, \mathcal{F}) = \max_{\substack{f \in \mathcal{F} \\ \text{Score}}} \min_{\substack{P_X \in \mathcal{P}_\lambda \\ \text{LLM} \\ \text{distribution}}} \max_{\substack{P_{XS} \\ \text{Watermarked} \\ \text{distribution}}} \left( \mathbb{E}_{P_{X|S}P_S}\left[f(X, S)\right] - \mathbb{E}_{P_X P_S}\left[f(X, S)\right] \right). \quad (1)$$

We call (1) the *maximum detection gap*. We restrict the inner maximization in (1) to couplings between the marginals $(P_X, P_S)$, i.e., $P_{X|S}$ induces a joint distribution $P_{X,S}$ on $\mathcal{X} \times \mathcal{S}$ whose marginals are $P_X$ and $P_S$. As a result, the inner maximization is an OT problem [35, 36]. We denote the class of couplings of $(P_X, P_S)$ as $\Pi_{X,S}$. By limiting watermarked distributions to couplings, we ensure that the watermark is *distortion-free*: we have $\mathbb{E}_S[P_{X,S}] = P_X$. From the perspective of a user who does not know $S$, distortion-free watermarks incur (in theory) no perceptible change in text quality. In Section 3 we propose a simple scheme that distorts the coupling solution to further increase $\mathsf{D}_{\mathsf{gap}}$.

**Proposed method.** Following the optimization (1), we propose two watermarking schemes in Sections 3 and 4, respectively. Both watermarks differ on the nature of the score function $f$, but share the same underlying algorithmic structure given in Algorithm 1 and described next.

Our watermarks receive as input the next-token distribution $P_X$, side information sample $s \in \mathcal{S}$, and a score function $f$. Given a pair $(P_X, f)$, the inner maximization in (1) amounts to an OT problem between $P_X$ and $P_S$, which is set to be uniform on $[1 : k]$. The OT problem is given by

$$P_{XS}^* = \arg\max_{P_{XS} \in \Pi_{X,S}} \left(\mathbb{E}_{P_{XS}}\left[f(X, S)\right] - \mathbb{E}_{P_X P_S}\left[f(X, S)\right]\right). \quad (2)$$

We solve (2) using Sinkhorn's algorithm [19]. Note that the score function $f$ can be equivalently denoted as the $(m \times k)$-dimensional OT cost matrix $\mathbf{C}$, which is defined as $\mathbf{C}_{x', s'} = -f(x', s')$ for $(x', s') \in [1 : m] \times [1 : k]$. Sinkhorn's algorithm yields a coupling $P_{X,S}$, from which the watermarked distribution is obtained via $P_{X|S=s}(\cdot|s) = k \cdot P_{X,S}(\cdot, s)$. The watermarked distribution is used to sample the next token.

**Tilting.** The watermarked distribution returned by the OT optimization does not change the average next-token probabilities. To further increase watermark detectability, we propose a tilting operation to $P_{X|S=s}$, which we denote by tilt. The tilting operation increases the probability of higher scoring tokens, at the cost of incurring distortion to $P_{X|S=s}$. This is done by increasing the probability of $x \in \mathcal{X}$ with $f(x, s) > \mathbb{E}_{P_{X,S}}[f(X, S)]$ and decrease the probability of $x \in \mathcal{X}$ with $f(x, s) < \mathbb{E}_{P_{X,S}}[f(X, S)]$. The amount of tilting (and thereby the level is distortion) is determined by a parameter $\delta \in (0, 1)$. The structure of tilting depends on the structure of $f$ and is later defined for each watermark separately. See Algorithm 1 for the list of steps.

**Detection.** Watermark detection is performed as follows: Given a token sequence $(x_1, \dots, x_n)$ and a score $f$ we recover the set of side information samples $(s_1, \dots, s_n)$ and compute $f(x_t, s_t)$ for each observed pair $(x_t, s_t)$. We declare the text watermarked if the average score $\frac{1}{n}\sum_{t=1}^{n} f(x_t, s_t)$ exceeds a user-defined threshold $\tau \geq 0$.

**Limitations.** As we will see shortly, (1) will inform the design of watermarking schemes with favorable empirical performance. However, this optimization framework is not without limitations. First, it is restricted to threshold tests applied to the averaged score, which are potentially suboptimal [37]. Second, we assume perfect randomness for side information generation, i.e., we can sample a uniformly distributed and i.i.d. random variable $S$ for each token. When $X$ is not watermarked, we also assume it is independent of $S$. In sequential hashing schemes this is not necessarily the case, since both $S$ and $X$ will depend on previous tokens. We emphasize that generating high-quality side information remains an important practical challenge relevant to all existing watermarks, including ours. Nevertheless, from a design perspective, this challenge is somewhat orthogonal to our guiding question of *how to optimally use* side information. We examine the impact of non-i.i.d. side information on our watermark design in Appendix D.2.

---

**Algorithm 1** `SimplexWater` and `HeavyWater` Watermark Generation

---

1: **Inputs**: Token distribution $P_X$, score function $f$, side information $s$, tilting parameter $\delta$.
2: **Outputs:** Watermarked distribution $P_{X|S=s}$
3: Calculate OT cost matrix $C_{x',s'} = -f(x',s')$, for all $(x',s') \in [1:m] \times [1:k]$
4: $P_{X,S} = \mathsf{Sinkhorn}\,(C, P_X, P_S)$, where $P_S = \mathsf{Unif}([1:k])$
5: $P_{X|S=s}(x|s) = k \cdot P_{X,S}(x,s)$
6: **if** $\delta > 0$ **then**
7: $\quad P_{X|S=s} \leftarrow \mathsf{tilt}(P_{X|S=s}, s, \delta)$
8: $\quad$ Normalize $P_{X|S=s}$
9: **end if**
10: **Return** $P_{X|S=s}$

---

## 3 `SimplexWater`: Watermark Design With Binary Scores

In this section, we consider the class of binary score functions, i.e., $\mathcal{F} = \mathcal{F}_{\mathsf{bin}} \triangleq \{f : \mathcal{X} \times \mathcal{S} \mapsto \{0,1\}\}$. Within this class, we solve (1) to obtain the optimal binary score function $f$, and present the corresponding optimal watermark, which we call `SimplexWater`. We first present a simplification of the maximum detection gap in (1) for $f \in \mathcal{F}_{\mathsf{bin}}$ and the low-entropy regime $\lambda \in [1/2, 1)$.

**Proposition 1.** *Let $\lambda \in \left[\frac{1}{2}, 1\right)$. For $f \in \mathcal{F}_{\mathsf{bin}}$, define the vector $f_i = [f(i,1), \ldots, f(i,k)] \in \{0,1\}^k$ for each $i \in \mathcal{X}$. Then,*

$$\mathsf{D}_{\mathsf{gap}}(m,k,\lambda,\mathcal{F}_{\mathsf{bin}}) = \max_{f \in \mathcal{F}_{\mathsf{bin}}} \quad \min_{i,j \in \mathcal{X}, i \neq j} \quad \frac{(1-\lambda)d_H(f_i, f_j)}{k}, \tag{3}$$

*where $d_H(a,b) = \sum_{i=1}^{k} \mathbf{1}_{\{a_i \neq b_i\}}$ denotes the Hamming distance between $a, b \in \{0,1\}^k$ and $\mathbf{1}_{\{\cdot\}}$ is the indicator function.*

Remarkably, (3) is equivalent to a classical problem in coding theory: the design of distance-maximizing codes! In this problem [14], the goal is to design a set of $m$ binary vectors of length $k$ – called *codewords* – with maximum pairwise Hamming distance [14, 38]. There is a one-to-one equivalence between designing a code with maximum distance between codewords and designing a binary score function for watermarking LLMs: for a fixed token $x$, the score function vector $[f(x,1), \ldots, f(x,k)]$ can be viewed as a codeword. Conversely, a distance-maximizing code can be used to build a binary score function that maximizes the detection gap. This connection with coding theory allows us to use classic coding theory results – specifically the Plotkin bound [39] – to derive an upper bound for $\mathsf{D}_{\mathsf{gap}}$ in (3). This result is stated in the next theorem.

**Theorem 1** (Maximum Detection Gap Upper Bound). *Consider the class of binary score functions $\mathcal{F}_{\mathsf{bin}}$ and uniform $P_S$. Then, for any $\lambda \in \left[\frac{1}{2}, 1\right)$, the maximum detection gap can be bounded as*

$$\mathsf{D}_{\mathsf{gap}}(m,k,\lambda,\mathcal{F}_{\mathsf{bin}}) \leq \frac{m(1-\lambda)}{2(m-1)} \tag{4}$$

The proof of Theorem 1 is given in Appendix B.2. For a fixed token vocabulary size $m$, this bound remains the same for any choice of $k$. Therefore, any score function constructed with any $k$ that achieves this bound is optimal. This observation serves as the starting point for our optimal watermark design, leading to `SimplexWater`.

`SimplexWater`: **An Optimal Binary-Score Watermark.** The upper bound in (4) is achievable when the score function is constructed using a *simplex code* – a family of codes that attain the Plotkin bound [40, 41]. A simplex code is defined as follows:

**Definition 1** (Simplex Code). *For any $x, s \in [0 : m-1]$, let $\mathsf{bin}(x)$, $\mathsf{bin}(s)$ denote their binary representations respectively using $\log_2 m$ bits. A simplex code $f_{\mathsf{sim}} : [0:m-1] \times [1:m-1] \to \{0,1\}$ is characterized by*

$$f_{\mathsf{sim}}(x,s) \triangleq \mathrm{dot}(\mathsf{bin}(x), \mathsf{bin}(s)), \tag{5}$$

*where $\mathrm{dot}(\mathsf{bin}(x), \mathsf{bin}(s)) \triangleq \sum_{i=1}^{\log_2 m} \mathsf{bin}(x)_i \cdot \mathsf{bin}(s)_i$ and $\mathsf{bin}(v)_i$ denotes the $i$th bit in the binary representation of $v$.*

We present next `SimplexWater`, a watermark that employs the simplex code as its score function. `SimplexWater` operates according to the steps of Algorithm 1, where the cost matrix is derived from $f_{\text{sim}}$. Specifically, given a token distribution, side information and the cost matrix derived from $f_{\text{sim}}$, `SimplexWater` solves the OT via Sinkhorn's algorithm to compute the watermarked distribution $P_{X|S}$. The optimality of `SimplexWater` follows directly from its one-to-one correspondence to the Simplex code that attains the Plotkin bound and is stated formally in the following theorem.

**Theorem 2** (`SimplexWater` Optimality). *For any $\lambda \in \left[\frac{1}{2}, 1\right)$ the maximum detection gap upper bound* (4) *is attained by* `SimplexWater`.

The proof of Theorem 2 is given in Appendix B.3. Together, Theorems 1 and 2 imply that `SimplexWater` is minimiax optimal among all watermarks with binary-valued score functions in the low-entropy regime.

As outlined in Algorithm 1, we further increase detection by tilting the watermarked distribution obtained by `SimplexWater`. For binary-valued scores, the tilting operation (Alg. 1, step 7) is

$$\text{tilt}(P^*_{X|S=s}, s, \delta) = P^*_{X|S=s}(x, s) \left(1 + \delta \cdot (\mathbf{1}_{\{f(x,s)=1\}} - \mathbf{1}_{\{f(x,s)=0\}})\right). \tag{6}$$

In Section 5 we show that by adding mild distortion we significantly increase the detection power of `SimplexWater`.

# 4 `HeavyWater`: **Watermarking using Heavy-Tailed Distributions**

So far, we restricted our analysis to binary scores, i.e., $f(x, s) \in \{0, 1\}$. In this case, the detection gap is bounded by $m(1 - \lambda)/(2(m - 1))$ as shown in Theorem 1 – and `SimplexWater` achieves this bound. However, the binary constraint is quite restrictive. Watermarks such as [17] are not restricted to binary scores – and achieve excellent performance in practice (see Gumbel in Fig. 1). In what follows, we relax the binary constraint to improve variation in the score values, enabling us to break the binary detection barrier (4) and significantly enhance detection performance.

We consider score functions where each score $f(x, s)$ is sampled independently from a continuous distribution prior to the watermarking process[1]. Remarkably, under this formulation, the popular Gumbel watermark [17] emerges as a special case when scores are drawn from a Gumbel distribution [42]. Our framework allows us to explore a broader class of heavy-tailed distributions that improve upon the Gumbel approach (Figure 1).

`HeavyWater`: **Heavy-Tailed Score Distributions for Improved Detection.** Consider the watermarking setting where we draw each score value $f(x, s)$ i.i.d. from a continuous finite variance distribution $P_F$. This score is used in the OT problem in (2) and Algorithm 1 to obtain the watermarked distribution. Next, we show that the Gumbel watermark [17] falls into this category of watermarks when $P_F$ is a Gumbel distribution.

**Theorem 3** (Gumbel Watermark as OT). *When the score random variables $f(x, s)$, are sampled i.i.d. from Gumbel$(0, 1)$, the solution to the OT problem in* (2) *converges to the Gumbel watermark [17] as $|\mathcal{S}| = k \to \infty$.*

This connection situates the Gumbel watermark within our broader framework (1). Can the Gumbel watermark be improved by selecting a different score distribution $P_F$? Intuitively, a heavy-tailed $P_F$ (which Gumbel is not) could increase the probability of sampling large values of $f(x, s)$, thus also increasing the likelihood of watermark detection. To make this intuition precise, we analyze the maximum detection gap (1) when the score function is randomly generated instead of selected from a set $\mathcal{F}$.

For fixed $|\mathcal{X}| = m$, $|\mathcal{S}| = k$, and $\lambda \in \left[\frac{1}{2}, 1\right)$, denote the worst case detection gap achieved by some fixed distribution $P_F$ as,

$$\mathsf{D}_{\text{gap}}^{[P_F]}(m, k, \lambda) = \min_{P_X \in \mathcal{P}_\lambda} \max_{P_{XS}} \left(\mathbb{E}_{P_{XS}}\left[f(X, S)\right] - \mathbb{E}_{P_X P_S}\left[f(X, S)\right]\right) \tag{7}$$

Note that $\mathsf{D}_{\text{gap}}^{[P_F]}(m, k, \lambda)$ is a random variable due to the randomness of the $f(x, s)$ samples. We provide high-probability guarantees on the achievable maximum detection gap for any fixed distribution $P_F$ in the regime of large $k$.

---

[1]A natural first step would be to extend `SimplexWater` by using $q$-ary instead of binary codes. This, however, results in marginal performance gain, as we discuss in Appendix C.3.

**Theorem 4** (Detection Gap). *Let $\lambda \in \left[\frac{1}{2}, 1\right)$, and consider the score difference random variable $\Delta = f(x, s) - f(x', s')$ for some $(x, s) \neq (x', s')$, where $f(x, s)$ and $f(x', s')$ are sampled i.i.d. from $P_F$. Let the cumulative distribution function of $\Delta$ be $F$, and let $Q = F^{-1}$ be its inverse. Then,*

$$\lim_{k \to \infty} \mathsf{D}_{\mathsf{gap}}^{[P_F]}(m, k, \lambda) = \int_{1-\lambda}^{1} Q(u) du, \tag{8}$$

Theorem 4 shows that detection performance improves when the distribution of $\Delta = f(x, s) - f(x', s')$ have heavier tails, where both $f(x, s), f(x', s') \sim P_F$. We analyze several choices of heavy-tailed $P_F$ for maximizing $\int_{1-\lambda}^{1} Q(u) du$ (see details in Appendix C.2).

`HeavyWater`**: A watermark with heavy-tailed scores.** We propose `HeavyWater`, a watermark that leverages Theorem 4 and samples the score function from a heavy-tailed distribution. `HeavyWater` follows the steps of Algorithm 1 but uses scores that are generated by sampling i.i.d. from a heavy-tailed $P_F$ prior to generation. We explored several options for selecting $P_F$. In theory, the Gamma distribution maximizes (8) across several common heavy-tailed distribution (Appendix C.2). However, in practice, we observed that choosing $P_F$ to be lognormal achieved the highest detection accuracy.[2] Assuming that $P_F$ is normalized to have zero-mean, the tilting operation in `HeavyWater` is given by

$$\mathsf{tilt}(P_{X|S=s}^*, s, \delta) = P_{X|S=s}^*(x, s)\left(1 + \delta \cdot \mathsf{sign}(f(x, s))\right).$$

## 5   Numerical Experiments

**Experimental Setting.**   We evaluate the detection-distortion-robustness tradeoff of `HeavyWater` and `SimplexWater` following the experimental benchmarking guidelines in Waterbench [43] and MarkMyWords [44]. We use two popular prompt-generation datasets: Finance-QA [45] (covering Q&A tasks) and LCC [46] (covering coding tasks). We evaluate on three open-weight models: Llama2-7B [47], Llama3-8B [48], and Mistral-7B [49]. Implementation details are provided in Appendix C.4, and full results, including ablation studies, are presented in Appendix D.

**Baseline Watermarks.** We benchmark `SimplexWater` and `HeavyWater` against the Gumbel watermark [17], the Inverse-transform watermark [13], the Correlated Channel watermark [16], and the SynthID watermark [11] with binary scores and $K = 15$ competition layers. We select these methods based on code availability or ease of implementation. These watermarks are distortion-free (i.e., they preserve the average next token distribution). To our knowledge, they also do not have an out-of-the-box method for trading off distortion with detection accuracy. We also consider the Red-Green watermark [10], which can be tuned for such trade-offs.

**Evaluation Metrics.** We use **p-value** as the primary detection power metric (plotted as $-\log_{10} p$) because it offers a threshold-independent measure of statistical power of a watermark. Moreover, p-values are the metric of choice in [11, 13, 15, 43, 44, 50, 51]. The p-value measures the probability of obtaining a given test statistic – or a more extreme one – under the unwatermarked (null) distribution. This provides a direct connection to the false-alarm rate, avoids the need to choose arbitrary detection thresholds, and is well-suited for comparing different watermarking schemes on equal footing. When the distribution of the test statistic is not available, we use a z-test, which approximates the statistic as Gaussian under the Central Limit Theorem. For methods like Gumbel [17], where the test statistic distribution is known analytically, we use the exact form. At a generation length of $T = 300$, we observe that the Gaussian approximation provided by the z-test is very accurate, ensuring that p-values remain reliable even when the exact distribution is unknown.

Distortion is measured using the estimated **cross entropy** (CE) between the watermarked and unwatermarked distribution, defined as $-\frac{1}{n} \sum_{t=1}^{n} \log P_X(\widetilde{x}_t)$, where $(\widetilde{x}_t)_{t=1}^{n}$ are watermarked tokens and $P_X$ is the LLM's distribution. This is proportional to relative perplexity – a commonly used proxy for quality evaluation [21, 22]. We also measure **watermark size**, proposed by [44] and defined as the number of tokens it takes to detect the watermark with a certain p-value. We provide results for task-specific metrics for generation quality in Appendix D.5, including document summarization [52], document and long-form QA [53, 54], and code completion [46] as curated by [43].

---

[2]Since detection is evaluated via $z$-scores and $p$-values which standardize by mean and variance, the specific mean and variance of the selected distributions do not affect detection accuracy.

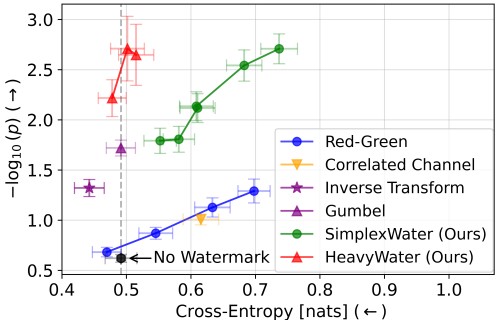

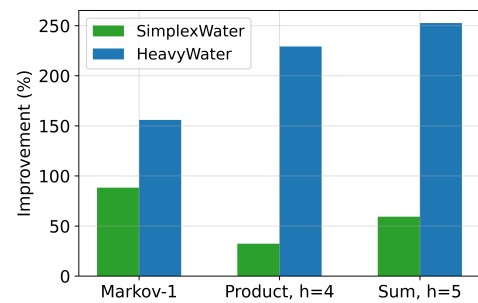

(a) Detection-Distortion Tradeoff. Mistral-7B, Coding.

(b) Watermark improvement over Red-Green [10] with several hashing strategies, $\delta = 2$. Llama2-7B, Coding.

**Figure 3: Left:** Tradeoff between detection (measured by $p$-value) and distortion (measured by Cross-Entropy) — `SimplexWater` and `HeavyWater` achieve higher detection rates while preserving token distributions close to the base unwatermarked model. **Right:** Detection gained by employing our watermark under various randomness generation schemes and several sliding window sizes $h$. Both `SimplexWater` and `HeavyWater` provide a significant improvement of up to $250\%$ and a decrease in distortion.

**Detection-Distortion Tradeoff.** We compare the tradeoff between watermark detection (p-value) against distortion (cross entropy). The results are given in Figures 1 and 3a. We make two key observations. First, considering watermarks with binary scores, `SimplexWater` Pareto dominates the red-green method. Second, `HeavyWater` provides the top performance along the baseline watermarks, incurring zero distortion. When we tune the watermarked distribution, both `SimplexWater` and `HeavyWater` obtain significant gains in detection, with mild distortion incurred.

Table 1: Results on HumanEval Functional Coding Dataset (pass@k, higher is better)

| Watermark Scheme | pass@1 (%) ↑ | pass@5 (%) ↑ | pass@10 (%) ↑ |
|---|---|---|---|
| No Watermark | 14.0 | 20.7 | 28.1 |
| **HeavyWater (Ours)** | **13.1** | **18.9** | **27.8** |
| Gumbel | 14.3 | 20.5 | 25.6 |
| **SimplexWater (Ours)** | **13.7** | **22.7** | **25.3** |
| Inverse Transform | 13.8 | 22.0 | 27.5 |
| Red/Green $\delta = 3$ | 11.6 | 19.5 | 23.2 |

**Additional Quality Metrics for Textual Tasks.** In addition to cross entropy, we have reported additional performance metrics for text quality: ROUGE-L and F1 scores for four additional tasks in Appendix D.5. In Table D.3, we assess the impact of watermarking on generation quality across four additional tasks from WaterBench [43]: LongformQA, Multi-news Summarization, Knowledge Memorization, and Knowledge Understanding. For these, we report ROUGE-L scores (LongformQA and Multi-news) and F1 scores (Knowledge Memorization and Knowledge Understanding). Our methods (SimplexWater and HeavyWater) maintain competitive detection performance across all datasets, with a generation metric comparable to unwatermarked text (see Table D.3). Together, these results provide significant evidence that our watermarks preserve output quality in textual tasks.

**Additional Quality Metrics for Coding Tasks.** We report additional tasks and metrics for low-entropy generation in coding. For LCC code completion tasks[46], since human-written codes are used as the ground truth, **Edit_Similarity** is the generation-quality metric. We benchmark our watermarking methods against prior work using this metric, and HeavyWater achieves the highest Edit_Similarity score for the code completion task (see Table D.4). For the HumanEval dataset [46], we take the standard metric **pass@**$K$ with $K$=1,5,10 generation metric. As shown in Table 1, SimplexWater achieves the highest score on pass@5 and HeavyWater performs best on pass@10. These results demonstrate that our watermarks preserve both functional correctness and syntactic quality of generated code.

**Watermark Size.** We consider $p = 10^{-3}$, which corresponds to a $0.1\%$ false-positive rate and report the average generation length to reach this $p$ value. As seen in Figure 4, where Llama2-7B is run on Q&A dataset, both the `SimplexWater` and `HeavyWater` schemes provide the lowest watermark size

in terms of the number of tokens required to attain a watermark detection strength, as measured by the p-value.

**Gain and Performance Under Hashing.** We illustrate how `SimplexWater` and `HeavyWater` can be coupled with different side information generation methods to boost watermark detection. We consider an experiment in which we replace the Red-Green watermark cost function and watermarked distribution with ours and report the gain in detection under several seed hashing methods adopted in [10]. As seen in Figure 3b, simply switching Red-Green ($\delta = 2$) with our methods – while keeping the same hashing scheme – introduces significant detection gain (up to $250\%$ improvement in the aggregated $-\log_{10} p$ value).

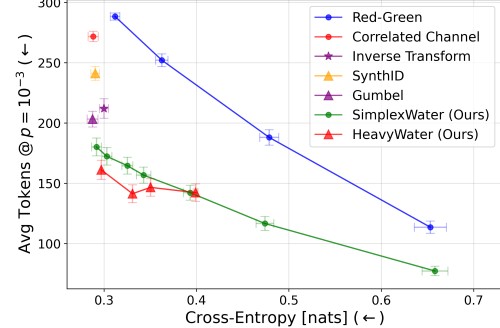

**Figure 4:** Our watermarks require fewer tokens to reach a given detection strength (p-value) with zero distortion.

**Additional Results.** The following experimental results and ablation studies are included in Appendix D: 1) An evaluation on the WaterBench dataset [43] in which we show that our methods do not degrade textual quality across an array of generation metrics (Appendix D.5); 2) A robustness study in which evaluate tamper resistance of our watermarks (Appendix D.3), and 3) computational overhead measurements, where we compare generation wall-clock times of various watermarking schemes (Appendix D.4).

## 6 Conclusion

We developed two new watermarking schemes by characterizing and analyzing an optimization problem that models watermark detection in low-entropy text generation. When restricting attention to binary scores, we draw connections to coding theory, which we then use to develop `SimplexWater`. We prove that `SimplexWater` is minimax optimal within our framework. When we sample the scores i.i.d. at random, we show that watermark detection depends on the tail of the score distribution. This leads to `HeavyWater`, where the scores are sampled from a heavy-tailed distribution. We further show that the Gumbel watermark is a special case of this construction. Both `SimplexWater` and `HeavyWater` demonstrate favorable performance on an array of LLM watermarking experiments when compared to various recent schemes. An interesting direction for future work is further optimizing the score function. This could be done by exploring other heavy-tailed score distributions, or using alternative strategies to optimize $f(x, s)$ such as backpropagating through (1). We also restrict our analysis to threshold tests – more powerful statistical tests could improve detection accuracy (see [23]). Our work contributes to better authentication of text provenance and fostering trust in AI systems. Agnostic to how side information is generated, our watermarks can be paired with new research in side information generation strategies that are more robust to tampering and adversarial attacks.

**Disclaimer.** This paper was prepared for informational purposes by the Global Technology Applied Research center of JPMorgan Chase & Co. This paper is not a product of the Research Department of JPMorgan Chase & Co. or its affiliates. Neither JPMorgan Chase & Co. nor any of its affiliates makes any explicit or implied representation or warranty and none of them accept any liability in connection with this paper, including, without limitation, with respect to the completeness, accuracy, or reliability of the information contained herein and the potential legal, compliance, tax, or accounting effects thereof. This document is not intended as investment research or investment advice, or as a recommendation, offer, or solicitation for the purchase or sale of any security, financial instrument, financial product or service, or to be used in any way for evaluating the merits of participating in any transaction.

**Acknowledgment.** This material is based upon work supported by the National Science Foundation under Grant No FAI 2040880, CIF 2231707, and CIF 2312667, as well as JP Morgan Chase.

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

# Supplementary material for
# `HeavyWater` **and** `SimplexWater`**: Watermarking Low-Entropy Text Distributions**

**Table of contents:**

# A  Additional Information on Related Works

## A.1  Additional Information on Related Work

**Optimization Framework**    Several optimization frameworks have been proposed for watermark analysis. [16, 25, 26, 33] adopts a hypothesis-testing framework for analyzing the statistical power of watermarking schemes. [33] goes beyond the vanilla threshold test to determine the optimal detection rule by solving a minimax optimization program. The authors of [25] consider an optimization under an additional constraint of controlled false-positive error. The watermarking scheme follows by learning a coupling that spreads the LLM distribution into an auxiliary sequence of variables (with alphabet size greater than $m$). While the proposed scheme is the optimal solution for the considered optimization, the scheme requires access to the (proxy of) LLM logits on both generation and detection ends. The authors of [55] consider a multi-objective optimization — maximization of the green list bias while minimizing the log-perplexity. Building on the hypothesis testing framework, we optimize over classes of score functions, by observing that the detection power of a watermark boils down to the separation of expected scores between the null and alternative hypotheses.

**Hashing Schemes in LLM Watermarking**    Current LLM watermarking schemes derive their per-token pseudorandom seed through five recurring hashing patterns. LeftHash hashes the immediately preceding token and feeds the digest to a keyed Psaudorandom function (PRF), yielding a light-weight, self-synchronising seed that survives single-token edits [15]. Window-hash generalises this by hashing the last $h$ tokens, expanding the key-space and hindering brute-force list enumeration at the cost of higher edit sensitivity [7, 15]. SelfHash (sometimes dubbed right-hand hashing") appends the *candidate* token to the left context before hashing, so the seed depends on both history and the token being scored; this hardens the scheme against key-extraction attacks and is used in multi-bit systems such as MPAC [31]. Orthogonally, counter-based keystreams drop context altogether and set the seed to $\mathrm{PRF}(K, \text{position})$, a strategy adopted by inverse-transform and Gumbel watermarks to preserve the original LM distribution in expectation [13, 17]. Finally, adaptive sliding-window hashing—popularised by *SynthID-Text*—hashes the last $H = 4$ tokens together with the secret key and *skips watermarking whenever that exact window has appeared before* ($K$-sequence repeated context masking"), thereby avoiding repetition artefacts while retaining the robustness benefits of a short window [11]. Semantic extensions build on these primitives: SemaMark quantises sentence embeddings [32], while Semantic-Invariant Watermarking uses a learned contextual encoder [56]. Collectively, these hashing families balance secrecy, robustness to editing or paraphrasing, and computational overhead, offering a rich design space for practical watermark deployments. Both `SimplexWater` and `HeavyWater` are agnostic to and can be applied on top of any hashing scheme. We provide the detection gain over Red-Green using various hashing schemes in Fig. 3b.

**Additional Distortion-Free Watermarks**    An array of recent works propose distortion-free watermarks that preserve the original LLM distribution [13, 16, 17, 20, 51, 57–59]. In addition to the ones that we have introduced in Section 1, [58] constructs undetectable watermarks using one-way functions, which is a cryptography-inspired technique. [59] proposes a watermark using error-correcting codes that leverages double-symmetric binary channels to obtain the watermarked distribution. [51] designs a distortion-free watermark based on multiple draws from a black-box LLM, which involves fixing a score function, drawing multiple tokens in each step, and outputting the highest-score token.

## A.2  Instantiation of Existing Watermarks As Score, Distribution and Randomness Design

Existing watermarks can be represented under the proposed outlook of side information, score function and watermarked distribution from Section 2. We next demonstrate that by instantiating several popular schemes through our lenses.

The **Red-Green watermark** [15] randomly partitions $\mathcal{X}$ into a green list and a red list. Here, $S$ is a set of $m$ binary random variables, representing the random list assignment. The function $f$ has binary outputs, which are used to increase the probability of green list tokens though exponential tilting of $P_X$.

The **SynthID** watermark [11] employs tournament sampling: a tournament between a set of $N^m$ token candidates along $m$-layers with $N$ competing token groups. Each tournament is performed given a sample of shared randomness (denoted $r$ [11]). On the $\ell$th layer the winners are taken to

| Watermark | # Bits |
|---|---|
| Red-Green [15] | $m$ |
| Inverse Transform [13] | $m$F |
| Gumbel [17] | $m$F |
| SimplexWater (**ours**) | $\log(m)$ |
| HeavyWater (**ours**) | $\log(k)$ |

Table A.2: Amount of random bits generated per step in popular watermarks. $m$ is the vocabulary size, F is the floating point precision and $k$ is the side information alphabet size.

be the token with highest score $f_\ell(x, s)$ within each $N$-sized set. The value of the score function $f$ is obtained from the side information for each token candidate. Different domains of $f$ induce a different construction of the function (See [11, Appendix E] for more information). The watermarked distribution $P_{X|S=s}$ is the obtained with a closed form in specific cases of $(f, N, m)$ and can be directly used instead of the instantiating a tournament. This form of watermarking is termed *vectorized tournament sampling*, in which the tournament is not applied, but the induced conditional distribution is employed instead. In this paper we consider vectorized tournament sampling with binary-valued scores, as this is the formulation given in [11] with a closed form, and the one that was utilized in their proposed experiments.

In the **Gumbel watermark** [17], the side information consists of $m$ i.i.d., uniform random variables on $[0, 1]$, each one corresponding to a single element in the vocabulary $x \in \mathcal{X}$. The score function $f$ is obtained by assigning each $x \in \mathcal{X}$ with a value $l_x = -\frac{1}{P_X(x)} \log(u_x)$, where $u_x$ the uniform variable corresponding to $x$ and $P_X(x)$ is the probability of $x \in \mathcal{X}$ under the unwatermarked model. The watermarked distribution is then given by assigning probability 1 to $argmax_{x \in \mathcal{X}} l_x(x)$ and probability 0 to the rest of $x \in \mathcal{X}$ (i.e., a singleton distribution).

In the **Correlated Channel** watermark [16], the side information corresponds to a partition of $\mathcal{X}$ into $k \geq 2$ lists and a single shared uniform variables $S'$ that is uniformly distributed on $[1 : k]$. The score function is then an indicator of the matching between the additional variable realization and the token random assignment, i.e., $f(x, s) = \mathbf{1}(B(x) = s)$, where $B(x) \in [1 : k]$ is the assignment of $x$ into one of the $k$ lists. The watermarked distribution is then given in closed form by solving the maximum coupling problem.

## A.3 Discussion on Randomness Efficiency

Given a hashing procedure that determines the random number generator seed value, we sample the side information $S \sim P_S$ over $\mathcal{S}$. The size of $\mathcal{S}$ determines the amount of side information we are required to sample. As described in Appendix A.2, each watermarks corresponds to a different size fo $\mathcal{S}$. We interpret that as *randomness efficiency*. That is, the bigger $\mathcal{S}$ is, the more bits of randomness we are required to extract from the random seed. We argue that a byproduct of our method is randomness efficiency, i.e., SimplexWater and HeavyWater require less random bits to sampled from the random seed compared to existing schemes. To that end, we compare with several popular watermarks (see Table A.2 for a summary).

The Red-Green watermark [15] corresponds to sampling a single bit for each element in $\mathcal{X}$, whose value determines the token's list assignment (red or green), thus resulting in a total of $m$ bits. The inverse transform watermark [13] requires sampling a single uniform and sampling a random permutation of $[1 : m]$. Thus, it asymptotically requires F $\log(m!)$ bits, where F is the resolution of the floating point representation used to sample the sampled uniform variable in bits (e.g. 32 for float32). For the Gumbel watermark, we sample a uniform variable for each $x \in \mathcal{X}$, resulting in a total of $m$F bits.

In SimplexWater, the side information size is $|\mathcal{S}| = m - 1$. To that end, we required $\log(m)$ bits to sample a single $s \in [1 : m - 1]$. Furthermore, HeavyWater is not constrained to a specific size of $|\mathcal{S}| = k$, and in the proposed experiments we take $k = 1024$ which is significantly smaller than both $m$ and $2^F$. The resulting amount of bits to be sampled from the random seed is $\log(1024) = 10$ bits. Consequently, we observe that HeavyWater is the most randomness-efficient watermark across considered schemes, while also being the best-performing watermark across considered experiments (see Section 5).

Minimizing the number of random bits extracted from a random number generator directly reduces computational and energy overhead in watermark embedding as less information is to be stored on the GPU. This eases implementation in resource-constrained hardware by lowering entropy demands and memory usage. Additionally, it limits side-channel leakage by shrinking an adversary's observable output [60], which may lead to more secure watermarking. We leave an extensive study and quantification of the benefits of randomness efficiency to future work.

# B Proofs for Theorems from Section 3 and 4.

We prove Proposition 1, Theorem 1, Theorem 2, Theorem 3 and Theorem 4

## B.1 Proof of Proposition 1

In this section, we prove Proposition 1, which connects the watermark design problem to a coding theoretic problem when the score function class is chosen to be binary.

**Proposition 1 (restated):** Let $\lambda \in \left[\frac{1}{2}, 1\right)$. For $f \in \mathcal{F}_{\mathsf{bin}}$, define the vector $f_i = [f(i,1), \ldots, f(i,k)] \in \{0,1\}^k$ for each $i \in \mathcal{X}$. Then,

$$\mathsf{D}_{\mathsf{gap}}(m,k,\lambda,\mathcal{F}_{\mathsf{bin}}) = \max_{f \in \mathcal{F}_{\mathsf{bin}}} \quad \min_{i,j \in \mathcal{X}, i \neq j} \quad \frac{(1-\lambda)d_H(f_i, f_j)}{k}, \tag{B.9}$$

where $d_H(a,b) = \sum_{i=1}^{k} \mathbf{1}_{\{a_i \neq b_i\}}$ denotes the Hamming distance between $a, b \in \{0,1\}^k$ and $\mathbf{1}_{\{\cdot\}}$ is the indicator function.

*Proof of Prop.1.* Recall that we consider $P_S \sim \mathcal{U}(1:k)$. For any $P_X$, let $\Psi(P_X) = \max_{P_{XS}} \left( \mathbb{E}_{P_{XS}}[f(X,S)] - \mathbb{E}_{P_X P_S}[f(X,S)] \right)$.

**Claim 1.** $\operatorname{argmin}_{P \in \mathcal{P}_\lambda} \Psi(P) \in \mathcal{P}_{\mathsf{spike},\lambda}$, *where*

$$\mathcal{P}_{\mathsf{spike},\lambda} = \{P \in \Delta_m \mid \{P_X(x_1), P_X(x_2), \ldots, P_X(x_m)\} = \{\lambda, 1-\lambda, 0, 0, \ldots, 0\}\}. \tag{B.10}$$

*Here $P_X(x_i)$ denotes the $x_i$th element of the $P_X$ probability vector with $(x_1, \ldots, x_m)$ representing any permutation of $(1, \ldots, m)$. $\Delta_m$ is the $m$-dimensional probability simplex.*

*Proof of Claim 1.* To prove Claim 1, we first show that $\Psi(P_X)$ is concave in $P_X$. By definition, for any given $P_X$ and uniform $P_S$, we have,

$$\Psi(P_X) = \max_{P_{XS}} \left( \mathbb{E}_{P_{XS}}[f(X,S)] - \mathbb{E}_{P_X P_S}[f(X,S)] \right).$$

Define $P_{XS}^*$ as the optimal joint distribution that achieves the maximum for a given $P_X$. Then, we have:

$$\Psi(P_X) = \mathbb{E}_{P_{XS}^*}[f(X,S)] - \mathbb{E}_{P_X P_S}[f(X,S)].$$

Now, consider the mixture $P_X^\theta = \theta P_X^{(1)} + (1-\theta)P_X^{(2)}$ for some $\theta \in [0,1]$ and $P_X^{(1)}, P_X^{(2)} \in \Delta_m$. Given $P_{XS}^{(1)*}$ and $P_{XS}^{(2)*}$, the maximizing couplings of $\Psi(P_X^{(1)})$ and $\Psi(P_X^{(2)})$ respectively, we define a mixed joint distribution: $P_{XS}^{\theta*} = \theta P_{XS}^{(1)*} + (1-\theta)P_{XS}^{(2)*}$. Note that,

$$\sum_s P_{XS}^{\theta*} = \theta \sum_s P_{XS}^{(1)*} + (1-\theta) \sum_s P_{XS}^{(2)*} \tag{B.11}$$

$$= \theta P_X^{(1)} + (1-\theta)P_X^{(2)} \quad \text{(as } P_{XS}^{(i)*} \text{ is the optimal coupling of } P_X^{(i)} \text{ and } P_S.\text{)} \tag{B.12}$$

$$= P_X^{(\theta)} \tag{B.13}$$

$$\sum_x P_{XS}^{\theta*} = \theta \sum_x P_{XS}^{(1)*} + (1-\theta) \sum_x P_{XS}^{(2)*} \tag{B.14}$$

$$= \frac{\theta}{k} + (1-\theta)P_X^{(2)} \quad \text{(as } P_{XS}^{(i)*} \text{ is the optimal coupling of } P_X^{(i)} \text{ and } P_S.\text{)} \tag{B.15}$$

$$= \frac{1}{k} \tag{B.16}$$

which shows that $P_{XS}^{\theta*}$ is a valid coupling of $P_X^{(\theta)}$ and uniform $P_S$. Using the linearity of the expectation operation, the expectation under the mixed distribution $P_{XS}^{\theta*}$ is:

$$\mathbb{E}_{P_{XS}^{\theta*}}[f(X,S)] = \theta\mathbb{E}_{P_{XS}^{(1)*}}[f(X,S)] + (1-\theta)\mathbb{E}_{P_{XS}^{(2)*}}[f(X,S)].$$

Similarly, for the independent case:

$$\mathbb{E}_{P_X^\theta P_S}[f(X,S)] = \theta\mathbb{E}_{P_X^{(1)} P_S}[f(X,S)] + (1-\theta)\mathbb{E}_{P_X^{(2)} P_S}[f(X,S)].$$

Since $\Psi(P_X^\theta)$ by definition takes the maximum coupling among all $P_{XS}$, it upper bounds the value attained by the specific mixture $P_{XS}^{\theta*}$. Thus,

$$\begin{aligned}
\Psi(P_X^\theta) &\geq \mathbb{E}_{P_{XS}^{\theta*}}[f(X,S)] - \mathbb{E}_{P_X^\theta P_S}[f(X,S)] \\
&= \theta\mathbb{E}_{P_{XS}^{(1)*}}[f(X,S)] + (1-\theta)\mathbb{E}_{P_{XS}^{(2)*}}[f(X,S)] - \theta\mathbb{E}_{P_X^{(1)} P_S}[f(X,S)] \\
&\quad - (1-\theta)\mathbb{E}_{P_X^{(2)} P_S}[f(X,S)]) \\
&= \theta\Psi(P_X^{(1)}) + (1-\theta)\Psi(P_X^{(2)}),
\end{aligned}$$

which proves concavity.

We now show that for any $\lambda \in \left[\frac{1}{2}, 1\right]$, the minimizer of $\Psi(P_X)$ lies in the set $\mathcal{P}_{\text{spike},\lambda}$, where

$$\mathcal{P}_{\text{spike},\lambda} = \{P \in \Delta_m \mid \{P_X(x_1), P_X(x_2), \ldots, P_X(x_m)\} = \{\lambda, 1-\lambda, 0, 0, \ldots, 0\}\}, \quad \text{(B.17)}$$

Recall that $\mathcal{P}_\lambda = \{P \in \Delta_m, \|P\|_\infty \leq \lambda\}$. This is a convex set and the extreme points of this set are precisely the spike distributions in $\mathcal{P}_{\text{spike},\lambda}$, which correspond to permutations of $(\lambda, 1-\lambda, 0, \ldots, 0)$. Since $\Psi(P_X)$ is concave, its minimum over the convex set $\mathcal{P}_\lambda$ occurs at an extreme point of $\mathcal{P}_\lambda$. Hence, the minimizer of $\Psi(P_X)$ belongs to $\mathcal{P}_{\text{spike},\lambda}$, which completes the proof. $\qquad\square$

Using Claim 1, we prove Proposition 1 as follows. Let $p^\star \in \mathcal{P}_{\text{spike},\lambda}$ be the minimizer in the minimization of $\mathsf{D}_{\text{gap}}(m,k,\lambda,\mathcal{F}_{\text{bin}})$ in (1), and let $(i,j)$ be its non-zero entries, i.e., $p_i^\star = \lambda$ and $p_j^\star = 1-\lambda$. Under such $p^\star$ and uniform $P_S$, any coupling can be written as

$$P_{XS}(x,s) = \begin{cases} \lambda Q_{X|S}(s \mid i) & \text{if } x = i, \\ (1-\lambda)Q_{X|S}(s \mid j) & \text{if } x = j \\ 0 & \text{if } x \neq i, j \end{cases}$$

where $Q_{S|X} = P_{XS}/P_X$ denotes the conditional distribution of the shared randomness $S$ given the selected token $X$. The coupling constraint for each $s \in \mathcal{S}$ becomes

$$\lambda Q_{S|X}(s \mid i) + (1-\lambda)Q_{S|X}(s \mid j) = \frac{1}{k},$$

To that end, we represent such coupling as

$$Q_{S|X}(s \mid i) = a_s, \quad Q_{S|X}(s \mid j) = \frac{\frac{1}{k} - \lambda a_s}{1-\lambda}$$

for some set of parameters $(a_s)_{s \in \mathcal{S}}$, such that $a_s \in [0, \frac{1}{k\lambda}]$. Under this construction, considering the worst case $(i,j)$ pair of non-zero $P_X$ indices, we have,

$$\begin{aligned}
\mathsf{D}_{\text{gap}}(m,k,\lambda,\mathcal{F}_{\text{bin}}) &= \max_{f:\mathcal{X}\times\mathcal{S}\mapsto[0,1]} \min_{i\neq j} \max_{Q_{X|S}} \sum_{s=1}^k \lambda Q_{S|X}(s|i)f(i,s) + (1-\lambda)Q_{S|X}(s|j)f(j,s) \\
&\quad - \left(\frac{\lambda}{k}f(i,s) + \frac{1-\lambda}{k}f(j,s)\right) \quad\quad \text{(B.18)} \\
&= \max_{f:\mathcal{X}\times\mathcal{S}\mapsto[0,1]} \min_{i\neq j} \max_{a_s\in[0,\frac{1}{k\lambda}]} \sum_{s=1}^k \lambda a_s f(i,s) + (1-\lambda)\left(\frac{\frac{1}{k} - \lambda a_s}{1-\lambda}\right)f(j,s) \\
&\quad - \left(\frac{\lambda}{k}f(i,s) + \frac{1-\lambda}{k}f(j,s)\right) \quad\quad \text{(B.19)} \\
&= \max_{f:\mathcal{X}\times\mathcal{S}\mapsto[0,1]} \min_{i\neq j} \max_{a_s\in[0,\frac{1}{k\lambda}]} \sum_{s=1}^k \lambda\left(a_s - \frac{1}{k}\right)(f(i,s) - f(j,s)) \quad \text{(B.20)}
\end{aligned}$$

The design of the coupling $P_{XS}$ boils down to choosing $\{a_s\}_{s \in \mathcal{S}}$ such that (B.20) is maximized. Moreover, since $\sum_{s=1}^{k} Q_{S|X}(s|x) = 1$ for $x = i$ and $x = j$, we have,

$$\sum_{s=1}^{k} Q_{S|X}(s|x) = 1 \quad \Longrightarrow \quad \sum_{s=1}^{k} a_s = 1 \tag{B.21}$$

For a given $f$ and any two indices $(i, j)$, we characterize optimal $a_s$ as follows. Let $m_+^{i,j} = \#\{s : f(i, s) - f(j, s) = 1\}$ and $m_-^{i,j} = \#\{s : f(i, s) - f(j, s) = -1\}$ where $\#$ denotes the cardinality of a set. Assume that the score functions $f$ satisfy $m_+^{i,j} \leq k\lambda$ and $m_-^{i,j} + m_-^{i,j} < k$. These assumptions are needed to eliminate trivial edge cases for which the inner minimization in (B.20) is zero, which leads to zero detection (see Appendix B.1.1). The inner-most maximum in (B.20) is obtained when:

$$a_s = \begin{cases} \frac{1}{k\lambda}, & s : f(i, s) - f(j, s) = 1 \\ 0, & s : f(i, s) - f(j, s) = -1 \\ \frac{1 - \frac{1}{k\lambda} m_+^{i,j}}{k - m_+^{i,j} - m_-^{i,j}}, & s : f(i, s) - f(j, s) = 0 \end{cases} \tag{B.22}$$

The optimal values of $a_s$ are obtained by allocating the maximum probability mass to the highest scores $f(i, s) - f(j, s)$ in (B.20). Continuing from (B.20), we have,

$$D_{\text{gap}}(m, k, \lambda, \mathcal{F}_{\text{bin}}) = \max_{f : \mathcal{X} \times \mathcal{S} \mapsto [0,1]} \min_{i \neq j} \frac{1}{k} \sum_{s : f_i - f_j = -1} \lambda |f(i, s) - f(j, s)|$$

$$+ \frac{1}{k} \sum_{s : f_i - f_j = 1} (1 - \lambda)|f(i, s) - f(j, s)| \tag{B.23}$$

$$= \max_{f : \mathcal{X} \times \mathcal{S} \mapsto [0,1]} \min_{i \neq j} \frac{1}{k}(\lambda m_-^{i,j} + (1 - \lambda)m_+^{i,j}) \tag{B.24}$$

Let $d_{ij} = \#\{s : f(i, s) \neq f(j, s)\}$. Then, $m_-^{ij} + m_+^{ij} = d_{ij}$. Let $m_-^{ij} = \beta d_{ij}$ and $m_+^{ij} = (1 - \beta)d_{ij}$ for some $\beta \in [0, 1]$. From (B.24), we have,

$$D_{\text{gap}}(m, k, \lambda, \mathcal{F}_{\text{bin}}) = \max_{f : \mathcal{X} \times \mathcal{S} \mapsto [0,1]} \min_{i \neq j} \min_{\beta \in [0,1]} \frac{1}{k}(\lambda\beta + (1 - \lambda)(1 - \beta))d_{i,j} \tag{B.25}$$

$$= \max_{f : \mathcal{X} \times \mathcal{S} \mapsto [0,1]} \min_{i \neq j} \min_{\beta \in [0,1]} \frac{1}{k}(\beta(2\lambda - 1) + (1 - \lambda))d_{i,j} \tag{B.26}$$

$$= \max_{f : \mathcal{X} \times \mathcal{S} \mapsto [0,1]} \min_{i \neq j} \frac{d_{i,j}}{k}(1 - \lambda) \tag{B.27}$$

since the inner minimum is achieved when $\beta = 0$, as $\lambda \geq \frac{1}{2}$. The proof is complete as $d_{ij}$ is exactly the Hamming distance between $f(i, \cdot)$ and $f(j, \cdot)$.

$\square$

### B.1.1   Edge Cases

In Appendix B.1, we assumed that the score functions $f$ satisfy $m_+^{i,j} \leq k\lambda$ and $m_+^{i,j} + m_-^{i,j} < k$. In this section, we show what happens when these constraints are violated.

**Case 1:** $m_+^{i,j} > k\lambda$: In this case, we obtain the optimal values of $a_s$ as in the proof of Proposition 1 above, by assigning the probability masses to the high score cases as follows. Let $\mathcal{J} \subset m_+^{i,j}$, $|\mathcal{J}| = k\lambda$ be any subset of $k\lambda$ values of $s$, each satisfying $f(i, s) - f(j, s) = 1$. In this case, similar to case 1, the inner maximum in (B.20) is obtained when:

$$a_s = \begin{cases} \frac{1}{k\lambda}, & s \in \mathcal{J} \\ 0, & s \notin \mathcal{J} \end{cases} \tag{B.28}$$

Continuing from (B.20), we have,

$$D_{\text{gap}}(m, k, \lambda, \mathcal{F}_{\text{bin}}) = \max_{f:\mathcal{X}\times\mathcal{S}\mapsto[0,1]} \min_{i\neq j} \lambda(1-\lambda) - \frac{\lambda(m_+^{i,j} - k\lambda)}{k} + \frac{\lambda m_-^{i,j}}{k} \tag{B.29}$$

$$= \max_{f:\mathcal{X}\times\mathcal{S}\mapsto[0,1]} \min_{i\neq j} \lambda + \frac{\lambda}{k}(m_-^{i,j} - m_+^{i,j}) \tag{B.30}$$

$$= \max_{f:\mathcal{X}\times\mathcal{S}\mapsto[0,1]} \min_{i\neq j} \min_{\beta\in[0,1]} \lambda + \frac{\lambda}{k}(2\beta - 1)d_{ij} \tag{B.31}$$

$$= \max_{f:\mathcal{X}\times\mathcal{S}\mapsto[0,1]} \min_{i\neq j} \lambda\left(1 - \frac{1}{k}d_{ij}\right) \tag{B.32}$$

$$= 0 \tag{B.33}$$

where the last equality follows from the fact that the inner minimum is achieved when $d_{ij} = k$, i.e., $f(i,\cdot)$ is the all zeros vector and $f(j,\cdot)$ is the all ones vector. The score functions $f$ that result in such cases are uninteresting as the detection can not be improved.

**Case 2:** $m_+^{i,j} \leq k\lambda$ **and** $m_+^{i,j} + m_-^{i,j} = k$**:** In this case, we obtain the optimal values of $a_s$ as in Appendix B.1, by assigning the probability masses to the high score cases as follows. Note that in this case $|s : f(i,s) - f(j,s) = 0| = 0$. Therefore,

$$a_s = \begin{cases} \frac{1}{k\lambda}, & s : f(i,s) - f(j,s) = 1 \\ \frac{1 - \frac{1}{k\lambda}m_+^{i,j}}{k - m_+^{i,j}}, & s : f(i,s) - f(j,s) = -1 \end{cases} \tag{B.34}$$

Continuing from (B.24), we have,

$$D_{\text{gap}}(m, k, \lambda, \mathcal{F}_{\text{bin}}) = \max_{f:\mathcal{X}\times\mathcal{S}\mapsto[0,1]} \min_{i\neq j} \frac{(1-\lambda)m_+^{ij}}{k} - \lambda\left(\frac{1}{k} - \frac{1 - \frac{1}{k\lambda}m_+^{ij}}{k - m_+^{i,j}}\right)(k - m_+^{ij}) \tag{B.35}$$

$$= \max_{f:\mathcal{X}\times\mathcal{S}\mapsto[0,1]} \min_{i\neq j} \frac{2m_+^{ij}}{k}(1-\lambda) \tag{B.36}$$

$$= 0 \tag{B.37}$$

where the last equality follows from the fact that the inner minimum is achieved when $m_+^{ij} = 0$, i.e., $f(i,\cdot)$ is the all zeros vector and $f(j,\cdot)$ is the all ones vector. The score functions $f$ that result in such cases are uninteresting as the detection can not be improved.

## B.2 Proof of Theorem 1.

In this section, we derive an upper bound for the detection gap in (1) using the Plotkin bound from coding theory [14].

**Theorem 1 (restated):** Consider the class of binary score functions $\mathcal{F}_{\text{bin}}$ and uniform $P_S$. Then, for any $\lambda \in \left[\frac{1}{2}, 1\right)$, the maximum detection gap can be bounded as

$$D_{\text{gap}}(m, k, \lambda, \mathcal{F}_{\text{bin}}) \leq \frac{m(1-\lambda)}{2(m-1)} \tag{B.38}$$

*Proof of Thm. 1.* Thm. 1 follows directly from the Plotkin bound [14], which provides an upper bound on the minimum normalized Hamming distance between any two codewords, considering any code construction. Indeed,

$$D_{\text{gap}}(m, k, \lambda, \mathcal{F}_{\text{bin}}) = \max_{f\in\mathcal{F}_{\text{bin}}} \min_{i,j\in\mathcal{X}, i\neq j} \frac{(1-\lambda)d_H(f_i, f_j)}{k} \leq (1-\lambda)\frac{m}{2(m-1)}. \tag{B.39}$$

$\square$

## B.3 Proof of Theorem 2.

In this section, we show that `SimplexWater` achieves the upper bound in (B.39).

Theorem 2 (restated): For any $\lambda \in \left[\frac{1}{2}, 1\right)$ the maximum detection gap upper bound (4) is attained by `SimplexWater`.

*Proof.* `SimplexWater` uses the Simplex code construction in Definition 1 as the score function. The simplex code achieves the Plotkin bound [14], i.e.,

$$\min_{i \neq j} \frac{d_H(f_{\mathsf{sim}}(i, \cdot), f_{\mathsf{sim}}(j, \cdot))}{k} = \frac{m}{2(m-1)} \tag{B.40}$$

Therefore,

$$\mathsf{D}_{\mathsf{gap}}(m, k, \lambda, \mathcal{F}_{\mathsf{bin}}) \geq \min_{i,j \in \mathcal{X}, i \neq j} \frac{(1-\lambda)d_H(f_{\mathsf{sim}}(i, \cdot), f_{\mathsf{sim}}(j, \cdot))}{k} = (1-\lambda)\frac{m}{2(m-1)}. \tag{B.41}$$

Considering the upper and lower bounds in (B.39) and (B.41), we have,

$$\mathsf{D}_{\mathsf{gap}}(m, k, \lambda, \mathcal{F}_{\mathsf{bin}}) = (1-\lambda)\frac{m}{2(m-1)}, \tag{B.42}$$

which is achieved by `SimplexWater`. $\square$

## B.4 Proof of Theorem 3.

In this section, we establish that the Gumbel watermark scheme [17] can be understood within our optimal transport framework, i.e., we prove Theorem 3, restated below.

**Theorem 5** (Gumbel Watermark as OT). *When the score random variables $f(x, s)$, are sampled i.i.d. from Gumbel$(0, 1)$, the solution to the OT problem in* (2) *converges to the Gumbel watermark [17] as $|\mathcal{S}| = k \to \infty$.*

To prove this, we first write down the Kantorovich dual of our optimal transport formulation and identify the dual potentials $\{\alpha_x, \beta_s\}$ that generate the arg-max coupling. We then analyze the behavior of these potentials in the limit $k \to \infty$, showing that the resulting arg-max rule converges to the classical Gumbel-Max sampling procedure. A final concentration argument ensures that the random coupling concentrates around its expectation, thereby establishing that the OT-derived sampler coincides with the Gumbel watermarking scheme.

To understand this connection, we first review the Gumbel watermarking scheme. The Gumbel-Max trick states that sampling from a softmax distribution can be equivalently expressed as:

$$y_t = \arg\max_{y \in \mathcal{V}} \frac{u_t(y)}{T} + G_t(y) \tag{B.43}$$

where $u_t(y)$ are the logits, $T$ is the temperature parameter, and $G_t(y) \sim$ Gumbel$(0, 1)$ independently for each token position $t$ and vocabulary element $y$. A Gumbel$(0, 1)$ random variable can be generated via:

$$G_t(y) = -\log(-\log(r_t(y))) \tag{B.44}$$

where $r_t(y) \sim$ Uniform$([0, 1])$.

In the Gumbel watermarking scheme, the uniform random variables are replaced with pseudo-random values:

$$r_t(y) \sim \text{Uniform}([0, 1]) \text{ (in unwatermarked model)} \tag{B.45}$$

$$r_t(y) = F_{y_{t-m:t-1}, \mathbf{k}}(y) \text{ (in watermarked model)} \tag{B.46}$$

Here, $F_{y_{t-m:t-1}, \mathbf{k}}(y)$ uses a secret key $\mathbf{k}$ and previously generated tokens $y_{t-m:t-1}$ to deterministically generate values that appear random without knowledge of $\mathbf{k}$.

To establish the connection to our OT framework (2), in the case when randomness is i.i.d. generated, we analyze the dual formulation of the optimal transport problem. In this formulation, we seek to minimize $\sum_x P_X(x)\alpha_x + \frac{1}{k}\sum_s \beta_s$ subject to $\alpha_x + \beta_s \geq f(x, s)$. The optimal values for these dual

variables satisfy specific conditions that link to the Gumbel-Max construction. The key insight comes from the optimality condition in our framework:

$$P_X(i) = \mathbb{P}(\arg\max_j [f(j,s) - \alpha_j^*] = i) \tag{B.47}$$

This can be directly mapped to the Gumbel-Max trick by identifying that $f(j,s)$ corresponds to the Gumbel noise $G_t(y)$ and $\alpha_j^*$ corresponds to $-\frac{u_t(y)}{T}$ (the negative normalized logits). With these identifications, the Gumbel-Max sampling expression:

$$y_t = \arg\max_{y \in \mathcal{V}} \frac{u_t(y)}{T} + G_t(y) = \arg\max_{y \in \mathcal{V}}[G_t(y) - (-\frac{u_t(y)}{T})] \tag{B.48}$$

Takes exactly the same form as our framework's expression $\arg\max_j[z(j,s) - \alpha_j^*]$. Further, the probability that this argmax equals $i$ is precisely $P_X(i)$ in our framework. Therefore, when $f(j,s) \sim$ Gumbel$(0,1)$, we will show below that our optimal transport solution exactly recovers the Gumbel watermarking scheme.

The detailed proof proceeds by analyzing the convergence of the discretized dual problem to its continuous limit as $k \to \infty$. The optimality conditions in the limit confirm that our OT framework generalizes the Gumbel watermark as a special case when scores are drawn from the Gumbel distribution. We start by defining a general optimal transport problem and its Kantorovich duality.

**Definition 2** (Optimal Transport Problem). *Given probability measures $\mu$ and $\nu$ on spaces $\mathcal{X}$ and $\mathcal{Y}$ respectively, and a cost function $c : \mathcal{X} \times \mathcal{Y} \to \mathbb{R}$, the optimal transport problem seeks to find a coupling $\pi$ (a joint distribution with marginals $\mu$ and $\nu$) that minimizes the expected cost:*

$$\inf_{\pi \in \Pi(\mu,\nu)} \int_{\mathcal{X} \times \mathcal{Y}} c(x,y)\, d\pi(x,y) \tag{B.49}$$

*where $\Pi(\mu,\nu)$ is the set of all probability measures on $\mathcal{X} \times \mathcal{Y}$ with marginals $\mu$ and $\nu$.*

The Kantorovich duality is a fundamental result that provides an equivalent formulation of this problem. For more details, see [35, Chapter 5].

**Theorem 1** (Kantorovich Duality). *The optimal transport problem is equivalent to:*

$$\sup_{(\varphi,\psi) \in \Phi_c} \left\{ \int_{\mathcal{X}} \varphi(x)\, d\mu(x) + \int_{\mathcal{Y}} \psi(y)\, d\nu(y) \right\} \tag{B.50}$$

*where $\Phi_c = \{(\varphi,\psi) : \varphi(x) + \psi(y) \leq c(x,y)\}$ is the set of functions satisfying the c-inequality constraint.*

This duality relationship (i.) transforms a non-trivial constrained optimization problem over probability measures into an optimization over functions, (ii.) provides a way to certify optimality through complementary slackness conditions and (iii.) enables us to analyze the convergence properties of OT problems through the convergence of dual objective functions. In many applications, the Kantorovich duality is rewritten with the constraint reversed (potential functions sum to $\geq c$), transforming the problem into a minimization rather than maximization. We adopt this convention in our formulation below.

**Primal Formulation.** Let us consider an optimal transport problem described by the inner maximization of (1), that is, given by Equation (2). Recall that given a pair $(P_X, f)$, the inner maximization in (1) amounts to an OT problem between $P_X$ and $P_S$, which is set to follow an uniform distribution on $[1:k]$. There, the score function $f$ can be equivalently denoted as the $(m \times k)$-dimensional OT cost matrix C, which is defined as $C_{x',s'} = -f(x',s')$ for $(x',s') \in [1:m] \times [1:k]$. This matrix is only generated once.

Now, let $P_X$ be a probability distribution over a finite set $\{1,2,\ldots,m\}$ and $P_S$ be a uniform distribution with mass $1/k$ at each point in $\{1,2,\ldots,k\}$. We have a cost function $f(x,s)$ with zero mean and unit variance, i.e., $\mathbb{E}_P[z] = 0$ and $\mathbb{E}_P[z^2] = 1$. The inner optimization in (1) aims to find a coupling $P_{XS}$ that maximizes:

$$C_{P,k}^* = \max_{P_{XS}} \sum_{x,s} P_{XS}(x,s) \cdot f(x,s) \tag{B.51}$$

$$\text{s.t.} \sum_s P_{XS}(x, s) = P_X(x) \quad \forall x \tag{B.52}$$

$$\sum_x P_{XS}(x, s) = \frac{1}{k} \quad \forall s \tag{B.53}$$

$$P_{XS}(x, s) \geq 0 \quad \forall x, s \tag{B.54}$$

**Dual Formulation.** In order to analyze this primal optimal transport problem we can look at the dual by using a Kantorovich duality argument. The equivalent dual problem is given by:

$$C_{D,k}^* = \min_{\alpha_x, \beta_s} \sum_x P_X(x)\alpha_x + \sum_s \frac{1}{k}\beta_s \tag{B.55}$$

$$\text{Subject to: } \alpha_x + \beta_s \geq f(x, s) \quad \forall x, s \tag{B.56}$$

In this dual formulation, $\alpha_x$ and $\beta_s$ are the Kantorovich potentials (i.e., Lagrange multipliers) enforcing the marginal constraints $\sum_s P_{XS}(x, s) = P_X(x)$ and $\sum_x P_{XS}(x, s) = \frac{1}{k}$ respectively. For any fixed values of $\alpha_x$, we need to determine the optimal values of $\beta_s$ that minimize the dual objective function. Since we aim to minimize the objective and the coefficients of $\beta_s$ are positive ($\frac{1}{k} > 0$), we want to make each $\beta_s$ as small as possible while still satisfying the constraints. For a given $s$, the constraint becomes:

$$\beta_s \geq f(x, s) - \alpha_x \quad \forall x \tag{B.57}$$

This means that $\beta_s$ must be at least as large as $f(x, s) - \alpha_x$ for every value of $x$. To ensure all constraints are satisfied while keeping $\beta_s$ minimal, we set:

$$\beta_s = \max_{x'}[f(x', s) - \alpha_{x'}] \tag{B.58}$$

This is the smallest value of $\beta_s$ that satisfies all constraints for a given $s$. Any smaller value would violate at least one constraint, and any larger value would unnecessarily increase the objective function. Substituting this expression back into the dual objective yields:

$$C_{D,k}^* = \min_{\alpha_x} \sum_{x=1}^{m} P_X(x)\alpha_x + \frac{1}{k} \sum_{s=1}^{k} \max_{x'}[f(x', s) - \alpha_{x'}] \tag{B.59}$$

This reformulation reduces the dual problem to an unconstrained minimization over $\alpha_x$ only.

To establish convergence of the dual problem, which is the key to embedding the Gumbel watermarking scheme in our framework, we will first recall a few fundamental concepts from variational analysis and convex optimization.

**Definition 3** (Epi-convergence)**.** *A sequence of functions* $f_k : \mathbb{R}^n \to \mathbb{R} \cup \{\pm\infty\}$ *is said to epi-converge to a function* $f : \mathbb{R}^n \to \mathbb{R} \cup \{\pm\infty\}$ *if the following two conditions hold:*

*(i) For every* $x \in \mathbb{R}^n$ *and every sequence* $x_k \to x$, $\liminf_{k\to\infty} f_k(x_k) \geq f(x)$.

*(ii) For every* $x \in \mathbb{R}^n$, *there exists a sequence* $x_k \to x$ *such that* $\limsup_{k\to\infty} f_k(x_k) \leq f(x)$.

Epi-convergence is particularly important in optimization because it guarantees that minimizers and minimal values converge appropriately. For more details on epi-convergence, see [61].

**Definition 4** (Equi-lower semicontinuity)**.** *A family of functions* $\{f_k\}$ *is equi-lower semicontinuous if for every* $x \in \mathbb{R}^n$ *and every* $\varepsilon > 0$, *there exists a neighborhood* $V$ *of* $x$ *such that*

$$\inf_{y \in V} f_k(y) > f_k(x) - \varepsilon \tag{B.60}$$

*for all* $k$.

The following result connects pointwise convergence with epi-convergence for convex functions:

**Theorem 2** ([62, Theorem 2])**.** *If* $\{f_k\}$ *is a sequence of convex, continuous, and equi-lower semicontinuous functions that converge pointwise to a function* $f$ *on* $\mathbb{R}^n$, *then* $\{f_k\}$ *epi-converges to* $f$.

As a first step toward applying our framework to the Gumbel watermarking scheme, we now characterize how the optimal transport cost behaves in the limit $k \to \infty$.

**Theorem 3.** *As $k \to \infty$, the optimal value of the discrete problem converges to the expected value problem:*

$$C_{D,k}^* \to C_D^* = \min_{x \in \mathbb{R}^m} p^T x + \mathbb{E}[\max_j (f(j,s) - x_j)] \tag{B.61}$$

*where $p_j = P_X(j)$ and $f(j,s)$ are random variables with the distribution matching the problem's cost function.*

*Proof of Theorem 3.* Let us rewrite the objective function in vector notation:

$$f_k(x) = p^T x + \frac{1}{k} \sum_{s=1}^{k} \max_j [f(j,s) - x_i] \tag{B.62}$$

where $\vec{z}_j = [z_{1,j}, \ldots, z_{m,j}]$ and $z_{i,j} := f(i, s_j)$ denote the sampled scores, i.e., each column $\vec{z}_j$ corresponds to the vector of scores across tokens for a fixed side-information sample $s_j$, and $x = [\alpha_1, \ldots, \alpha_m]$.

We need to establish that the functions $f_k$ are continuous and convex. Note that

1. The term $p^T x$ is linear and therefore continuous and convex.

2. The function $g_s(x) = \max_j [f(j,s) - x_j]$ is continuous for each $j$ because:
   (a) The functions $h_j(x) = f(j,s) - x_j$ are continuous for each $j$.
   (b) The maximum of a finite number of continuous functions is continuous. It is also convex as the maximum of linear functions.

3. The sum $\frac{1}{k} \sum_{s=1}^{k} g_s(x)$ is continuous as a linear combination of continuous functions and convex as a positive linear combination of convex functions.

As $k \to \infty$, by the Law of Large Numbers, for any fixed $x$, the sample average converges to the expected value:

$$\frac{1}{k} \sum_{s=1}^{k} \max_j [f(j,s) - x_j] \to \mathbb{E}[\max_j (f(j,s) - x_j)] \tag{B.63}$$

Therefore,

$$f_k(x) \to f(x) = p^T x + \mathbb{E}[\max_i (z_i - x_i)] \tag{B.64}$$

This establishes pointwise convergence of $f_k$ to $f$. Next, we need to prove that the family of functions $\{f_k\}$ is equi-lower semicontinuous.

**Lemma B.1.** *The family of functions $\{f_k\}$ defined above is equi-lower semicontinuous under mild assumptions on the boundedness of the data points $\vec{z}_j$.*

*Proof.* First, observe that each $f_k$ is continuous (and hence lower semicontinuous). For any $x \in \mathbb{R}^n$ and $\varepsilon > 0$, consider the neighborhood

$$V = \{y : \|y - x\|_\infty < \varepsilon/2\} \tag{B.65}$$

For any $y \in V$ and any $j, i$, we have

$$z_{i,j} - y_i > z_{i,j} - x_i - \varepsilon/2 \tag{B.66}$$

Therefore,

$$\max_i [z_{i,j} - y_i] \geq \max_i [z_{i,j} - x_i - \varepsilon/2] = \max_i [z_{i,j} - x_i] - \varepsilon/2 \tag{B.67}$$

This implies

$$\frac{1}{k}\sum_{j=1}^{k}\max_i[z_{i,j} - y_i] \geq \frac{1}{k}\sum_{j=1}^{k}\max_i[z_{i,j} - x_i] - \varepsilon/2 \tag{B.68}$$

Combined with the linear term, we get

$$f_k(y) = p^T y + \frac{1}{k}\sum_{j=1}^{k}\max_i[z_{i,j} - y_i] \tag{B.69}$$

$$\geq p^T x - \|p\|_1 \cdot \varepsilon/2 + \frac{1}{k}\sum_{j=1}^{k}\max_i[z_{i,j} - x_i] - \varepsilon/2 \tag{B.70}$$

$$= f_k(x) - \|p\|_1 \cdot \varepsilon/2 - \varepsilon/2 \tag{B.71}$$

By choosing $\varepsilon$ small enough, we ensure that $\|p\|_1 \cdot \varepsilon/2 + \varepsilon/2 < \varepsilon$, which gives us

$$\inf_{y \in V} f_k(y) > f_k(x) - \varepsilon \tag{B.72}$$

This holds for all $k$, establishing the equi-lower semicontinuity of the family $\{f_k\}$. $\qquad\square$

Since we have established that the functions $f_k$ are convex, continuous, and equi-lower semicontinuous, and that they converge pointwise to $f$, we can apply Lemma 2 to conclude that $f_k$ epi-converges to $f$. By the fundamental properties of epi-convergence, if $f_k$ epi-converges to $f$, then $\min f_k \to \min f$. Also, every limit point of minimizers of $f_k$ is a minimizer of $f$. This establishes that $C_{D,k}^* \to C_D^*$ as $k \to \infty$. $\qquad\square$

**Optimality Conditions and Characterization.** Now, we analyze the optimality conditions for the dual problem. Consider our objective function:

$$f_k(\alpha) = \sum_{i=1}^{m} P_X(i)\alpha_i + \frac{1}{k}\sum_{s=1}^{k}\max_j[f(j,s) - \alpha_j] \tag{B.73}$$

To find the derivative with respect to $\alpha_i$, we note that the derivative of the first term is simply $P_X(i)$. For the second term, we need to understand how the max function behaves. At points where a unique index $j$ achieves the maximum value of $f(j,s) - \alpha_j$, the derivative is:

$$\frac{\partial}{\partial \alpha_i}\max_j[f(j,s) - \alpha_j] = \begin{cases} -1 & \text{if } i = \operatorname{argmax}_j[f(j,s) - \alpha_j] \\ 0 & \text{otherwise} \end{cases} \tag{B.74}$$

The negative sign appears because $\alpha_i$ has a negative coefficient in the expression inside the max. For simplicity, we often write the optimality condition using the indicator function $\mathbb{1}[i \in \operatorname{argmax}_j[f(j,s) - \alpha_j]]$, which equals 1 when $i$ is in the argmax set and 0 otherwise.

$$\frac{\partial}{\partial \alpha_i}\max_j[f(j,s) - \alpha_j] = -\mathbb{1}[i = \operatorname*{argmax}_j[f(j,s) - \alpha_j]] \tag{B.75}$$

If there are ties (multiple indices achieve the maximum), then the max function is not differentiable. Instead, we use the concept of subdifferential. A valid subgradient can be written as $-\mathbb{1}[i \in \operatorname{argmax}_j[f(j,s) - \alpha_j]] \cdot w_i$, where $w_i \geq 0$ are weights such that $\sum_{i \in \operatorname{argmax}} w_i = 1$.

The derivative (or subgradient) of the entire objective function with respect to $\alpha_i$ is:

$$\frac{\partial f_k}{\partial \alpha_i} = P_X(i) - \frac{1}{k}\sum_{s=1}^{k}\mathbb{1}[i \in \operatorname*{argmax}_j[f(j,s) - \alpha_j]] \tag{B.76}$$

Then, for the optimal dual variables $\alpha^*$, the first-order optimality condition gives:

$$\frac{\partial f_k}{\partial \alpha_i}(\alpha^*) = P_X(i) - \frac{1}{k}\sum_{s=1}^{k}\mathbb{1}[i \in \arg\max_j[f(j,s) - \alpha_j^*]] = 0 \tag{B.77}$$

We interpret this optimality condition as follows: the probability $P_X(i)$ equals the empirical probability that $i$ is in the argmax set of $f(j,s) - \alpha_j^*$ across all samples $s$. Rearranging, we have:

$$\frac{1}{k}\sum_{s=1}^{k}\mathbb{1}[i \in \arg\max_j[f(j,s) - \alpha_j^*]] = P_X(i) \tag{B.78}$$

To fully understand the emergence of the Gumbel-Max connection, we must carefully analyze how the discrete optimality condition converges to its continuous counterpart as $k \to \infty$. The key step is understanding how the empirical average on Equation B.78 becomes the probability statement:

$$\Pr_j[\arg\max(z_j - \alpha_j^*) = i] = P_X(i) \tag{B.79}$$

In other words, it remains to show that

$$\frac{1}{k}\sum_{s=1}^{k}\mathbb{1}\{i \in \arg\max_j[f(j,s) - \alpha_j^*]\} \longrightarrow \mathbb{E}\Big[\mathbb{1}\{i \in \arg\max_j[f(j,s) - \alpha_j^*]\}\Big]. \tag{B.80}$$

In order to show this convergence, let $\alpha_k^*$ denote the optimal dual variables for the problem with $k$ samples. From the epi-convergence of $f_k$ to $f$, we know that $\alpha_k^* \to \alpha^*$ as $k \to \infty$.

For any fixed $\alpha$, the indicator functions $\mathbb{1}\{i \in \arg\max_j[f(j,s) - \alpha_j]\}$ are independent and identically distributed across $s$, since $f(j,s)$ are sampled independently. Therefore, by the Strong Law of Large Numbers:

$$\frac{1}{k}\sum_{s=1}^{k}\mathbb{1}\{i \in \arg\max_j[f(j,s) - \alpha_j]\} \xrightarrow{a.s.} \mathbb{E}[\mathbb{1}\{i \in \arg\max_j[f(j,s) - \alpha_j]\}] \tag{B.81}$$

To address the case where $\alpha$ is replaced by $\alpha_k^*$ which depends on the samples, we decompose:

$$\left|\frac{1}{k}\sum_{s=1}^{k}\mathbb{1}\{i \in \arg\max_j[f(j,s) - \alpha_{k,j}^*]\} - \mathbb{E}[\mathbb{1}\{i \in \arg\max_j[f(j,s) - \alpha_j^*]\}]\right| \tag{B.82}$$

$$\leq \left|\frac{1}{k}\sum_{s=1}^{k}\mathbb{1}\{i \in \arg\max_j[f(j,s) - \alpha_{k,j}^*]\} - \frac{1}{k}\sum_{s=1}^{k}\mathbb{1}\{i \in \arg\max_j[f(j,s) - \alpha_j^*]\}\right| \tag{B.83}$$

$$+ \left|\frac{1}{k}\sum_{s=1}^{k}\mathbb{1}\{i \in \arg\max_j[f(j,s) - \alpha_j^*]\} - \mathbb{E}[\mathbb{1}\{i \in \arg\max_j[f(j,s) - \alpha_j^*]\}]\right| \tag{B.84}$$

The second term converges to zero by the Strong Law of Large Numbers. For the first term, we exploit the fact that the argmax function is stable under small perturbations except at ties, which occur with probability zero for continuous distributions of $f$.

Specifically, let $\Delta_k = \|\alpha_k^* - \alpha^*\|_\infty$. For any realization of $f(j,s)$, the argmax changes only if the perturbation $\Delta_k$ exceeds the minimum gap between the maximum value and the second-largest value. Let $G_s = \min_{j \neq j^*}[f(j^*,s) - \alpha_{j^*}^*] - [f(j,s) - \alpha_j^*]$, where $j^* = \arg\max_j[f(j,s) - \alpha_j^*]$. Then:

$$\mathbb{1}\{i \in \arg\max_j[f(j,s) - \alpha_{k,j}^*]\} \neq \mathbb{1}\{i \in \arg\max_j[f(j,s) - \alpha_j^*]\} \implies \Delta_k > G_s \tag{B.85}$$

Since $\alpha_k^* \to \alpha^*$, we have $\Delta_k \to 0$ as $k \to \infty$. The probability $\mathbb{P}(\Delta_k > G_s)$ converges to zero because $G_s > 0$ almost surely for continuous distributions. By dominated convergence, the first term also converges to zero.

Consequently:

$$\frac{1}{k}\sum_{s=1}^{k}\mathbb{1}\{i \in \arg\max_j[f(j,s)-\alpha_{k,j}^*]\} \xrightarrow{p} \mathbb{E}[\mathbb{1}\{i \in \arg\max_j[f(j,s)-\alpha_j^*]\}] \tag{B.86}$$

Combined with our optimality condition in equation (B.78), we obtain:

$$P_X(i) = \mathbb{E}[\mathbb{1}\{i \in \arg\max_j[f(j,s)-\alpha_j^*]\}] = \mathbb{P}(\arg\max_j[f(j,s)-\alpha_j^*]=i) \tag{B.87}$$

In other words, the random mapping $S \mapsto \arg\max_j[f(j,s)-\alpha_j^*]$ reproduces the original law $P_X$ exactly. This fact is precisely what Theorem 3 asserts: the Gumbel-Max procedure constitutes the optimal-transport coupling between $P_X$ and the side-information mechanism.

**Connection to Gumbel Watermarking.** The Gumbel watermarking scheme described above can be directly interpreted within our optimal transport framework by noticing that this same arg-max coupling is exactly what underlies the Gumbel-Max trick: adding Gumbel noise $G_t(j)$ to (negative) logits $-\frac{u_t(y)}{T}$ and taking argmax samples from the softmax. In our formulation, the cost function $f(j,s)$ corresponds to the Gumbel noise $G_t(j)$, while the dual variables $\alpha_j$ correspond to $-\frac{u_t(j)}{T}$. The optimality condition

$$P_X(i) = \mathbb{P}(\arg\max_j[f(j,s)-\alpha_j^*]=i) \tag{B.88}$$

is precisely the Gumbel-Max trick, which states that sampling from a softmax distribution is equivalent to adding Gumbel noise to logits and taking the argmax.[3] In practice, under the watermarking process, the Gumbel scheme replaces the uniform variables $r_t(y)$ with pseudo-random values $F_{y_{t-m:t-1},k}(y)$ that depend on the secret key and previously generated tokens. This creates a coupling between the original token distribution and the side information, exactly as prescribed in our optimal transport approach. Thus, when we assume i.i.d. scores, the Gumbel watermarking scheme represents a specific instantiation of our general optimal transport framework, where the coupling is designed to preserve the original distribution in expectation while maximizing detectability through heavy-tailed score distributions.

## B.5 Proof of the Detection Gap, Theorem 4.

In Section 4, we introduced the `HeavyWater` scheme by generalizing the scores beyond the binary case. Since $f(x,s)$ is random, $\mathsf{D}_{\mathsf{gap}}^{[P_F]}(m,k,\lambda)$ given by

$$\mathsf{D}_{\mathsf{gap}}^{[P_F]}(m,k,\lambda) = \min_{P_X \in \mathcal{P}_\lambda} \max_{P_{XS}} \left(\mathbb{E}_{P_{XS}}[f(X,S)] - \mathbb{E}_{P_X P_S}[f(X,S)]\right) \tag{B.89}$$

is also a random variable. Now, we will show that we can go beyond Theorem 3, improving on the watermarking scheme like Gumbel, and prove Theorem 4 that connects the asymptotic detection gap with the quantiles of the distribution of the score difference $\Delta = f(x,s) - f(x',s')$.

**Theorem 4** (Detection Gap, asymptotic randomness). *Let $\lambda \in \left[\frac{1}{2}, 1\right)$, and consider the score difference random variable $\Delta = f(x,s) - f(x',s')$ for some $(x,s) \neq (x',s')$, where $f(x,s)$ and $f(x',s')$ are sampled i.i.d. from $P_F$. Let the cumulative distribution function of $\Delta$ be $F$, and let $Q = F^{-1}$ be its inverse. Then,*

$$\lim_{k\to\infty} \mathsf{D}_{\mathsf{gap}}^{[P_F]}(m,k,\lambda) = \int_{1-\lambda}^{1} Q(u)du. \tag{B.90}$$

*Proof.* We begin by characterizing the structure of the optimal coupling $P_{XS}$ that attains the maximum in (B.91) below.

---

[3]To see this precisely, substitute the $j$th logit as $-\alpha_j^*$ on the RHS of (B.88), and simplify the RHS using the same steps as in Appendix B.1 of [24]. This shows that (B.88) holds when $-\alpha_j^* = j$th logit. Which means that the Gumbel watermark is a special case of our construction in sec. 4 (which boils down to (B.88)), with $f(x,s)$ specifically chosen has Gumbel$(0,1)$.

$$\max_{P_{XS}} \mathbb{E}_{P_{XS}}^{[k]}[f(X,S)] - \mathbb{E}_{P_X P_S}^{[k]}[f(X,S)]$$

$$\text{s.t. } \sum_{s=1}^{k} P_{x,s} = p_x, \quad x \in \{1,\dots,m\}$$

$$\sum_{x=1}^{m} P_{x,s} = \frac{1}{k}, \quad s \in \{1,\dots,k\} \tag{B.91}$$

**Lemma 1.** *The minimum in* (B.89) *is achieved by the* $P_X \in \mathcal{P}_\lambda$ *satisfying*

$$P_X(x) = \begin{cases} \lambda, & x = i, \\ 1 - \lambda & x = j, \\ 0, & x \neq i, j. \end{cases} \tag{B.92}$$

*as shown in* (B.17). *Assume that* $\lambda \geq \frac{1}{2}$. *Then, the optimal* $P_{XS}$, *solution of Equation* (B.91), *is given by*

$$\begin{bmatrix} & s_1 & \cdots & s_r & s_{r+1} & s_{r+2} & \cdots & s_k \\ \lambda & \frac{1}{k} & \cdots & \frac{1}{k} & \lambda - \frac{r}{k} & 0 & \cdots & 0 \\ 1-\lambda & 0 & \cdots & 0 & \frac{r+1}{k}-\lambda & \frac{1}{k} & \cdots & \frac{1}{k} \end{bmatrix}$$

*where* $r$ *is the integer satisfying* $\frac{r}{k} \leq \lambda < \frac{r+1}{k}$.

The optimal coupling assigns the probability mass of $x = i$ sequentially to the first $r$ side-information bins until its total mass $\lambda$ is exhausted, after which the remaining bins are filled by $x = j$. This is intuitive as $P_X(i) \geq P_X(j)$.

For a given $k$, $\max_{P_{XS}} \mathbb{E}_{P_{XS}}^{[k]}[f(X,S)]$ is given by

$$\max_{P_{XS}} \mathbb{E}_{P_{XS}}^{[k]}[f(X,S)] \tag{B.93}$$

$$= \frac{1}{k}\sum_{s=1}^{r} f(1,s) + \left(\lambda - \frac{r}{k}\right) f(1,r+1) + \left(\frac{r+1}{k} - \lambda\right) f(2,r+1) + \frac{1}{k}\sum_{s=r+2}^{k} f(2,s) \tag{B.94}$$

$$= \frac{1}{k}\sum_{s=1}^{k} f(2,s) + \left(\lambda - \frac{r}{k}\right)\Delta_{r+1} + \frac{1}{k}\sum_{s=1}^{r}\Delta_s \tag{B.95}$$

For $\mathbb{E}_{P_X P_S}^{[k]}[f(X,S)]$, we have

$$\mathbb{E}_{P_X P_S}^{[k]}[f(X,S)] = \frac{\lambda}{k}\sum_{s=1}^{k} f(1,s) + \frac{1-\lambda}{k}\sum_{s=1}^{k} f(2,s). \tag{B.96}$$

Therefore, the difference is given by

$$\max_{P_{XS}} \mathbb{E}_{P_{XS}}^{[k]}[f(X,S)] - \mathbb{E}_{P_X P_S}^{[k]}[f(X,S)] \tag{B.97}$$

$$= \frac{1}{k}\sum_{s=1}^{k} f(2,s) + \left(\lambda - \frac{r}{k}\right)\Delta_{r+1} + \frac{1}{k}\sum_{s=1}^{r}\Delta_s - \frac{\lambda}{k}\sum_{s=1}^{k} f(1,s) - \frac{1-\lambda}{k}\sum_{s=1}^{k} f(2,s) \tag{B.98}$$

$$= \frac{\lambda}{k}\sum_{s=1}^{k} f(2,s) + \left(\lambda - \frac{r}{k}\right)\Delta_{r+1} + \frac{1}{k}\sum_{s=1}^{r}\Delta_s - \frac{\lambda}{k}\sum_{s=1}^{k} f(1,s) \tag{B.99}$$

Let us define the following terms that we will analyze precisely:

$$\bar{f}_{1,k} = \frac{1}{k}\sum_{s=1}^{k} f(1,s), \quad \bar{f}_{2,k} = \frac{1}{k}\sum_{s=1}^{k} f(2,s) \quad \text{and} \quad I_k = \frac{1}{k}\sum_{s=1}^{r}\Delta_s. \tag{B.100}$$

With these definitions, we can rewrite the detection gap as:

$$\max_{P_{XS}} \mathbb{E}^{[k]}_{P_{XS}}[f(X,S)] - \mathbb{E}^{[k]}_{P_X P_S}[f(X,S)] = \lambda \bar{f}_{2,k} + \left(\lambda - \frac{r}{k}\right)\Delta_{r+1} + I_k - \lambda \bar{f}_{1,k} \quad \text{(B.101)}$$

To establish the exact limit, we need to analyze each term with greater precision.

**Exact characterization of $\lambda(\bar{f}_{2,k} - \bar{f}_{1,k})$:** Both $f(1,s)$ and $f(2,s)$ are i.i.d. sub-exponential random variables with parameters $(\nu^2, b)$. By the strong law of large numbers, we have almost surely:

$$\lim_{k\to\infty} \bar{f}_{1,k} = \mathbb{E}[f(1,1)] \quad \text{and} \quad \lim_{k\to\infty} \bar{f}_{2,k} = \mathbb{E}[f(2,1)] \quad \text{(B.102)}$$

Since $f(x,s)$ are i.i.d. with zero mean, we have $\mathbb{E}[f(1,1)] = \mathbb{E}[f(2,1)] = 0$. Therefore, almost surely:

$$\lim_{k\to\infty} \lambda(\bar{f}_{2,k} - \bar{f}_{1,k}) = 0 \quad \text{(B.103)}$$

**Exact characterization of $\left(\lambda - \frac{r}{k}\right)\Delta_{r+1}$:** By definition of $r = \lfloor \lambda k \rfloor$, we have $0 \leq \lambda - \frac{r}{k} < \frac{1}{k}$. Since $\Delta_{r+1}$ is a sub-exponential random variable with parameters $(4\nu^2, 2b)$, it is almost surely finite. Combining this with the above inequality:

$$\lim_{k\to\infty} \left(\lambda - \frac{r}{k}\right)\Delta_{r+1} = 0 \quad \text{almost surely.} \quad \text{(B.104)}$$

**Exact characterization of $I_k$:** For this term, we use order statistics. Let $\Delta_{(1)} \leq \Delta_{(2)} \leq \cdots \leq \Delta_{(k)}$ denote the ordered values for the difference of scores $\Delta_s$. Then:

$$I_k = \frac{1}{k} \sum_{j=k-r+1}^{k} \Delta_{(j)}. \quad \text{(B.105)}$$

Let $F$ be the CDF of $\Delta_s$ and $F_k$ be the empirical CDF given by

$$F_k(x) := \frac{1}{k} \sum_{s=1}^{k} \mathbf{1}\{\Delta_s \leq x\}, \quad \text{(B.106)}$$

By the Glivenko-Cantelli theorem, we have:

$$\sup_{x\in\mathbb{R}} |F_k(x) - F(x)| \leq \eta_k \quad \text{where } \eta_k \to 0 \text{ almost surely as } k \to \infty \quad \text{(B.107)}$$

For each order statistic $\Delta_{(j)}$, the empirical CDF by definition gives $F_k(\Delta_{(j)}) = \frac{j}{k}$. The bound from Glivenko-Cantelli gives:

$$\frac{j}{k} - \eta_k \leq F(\Delta_{(j)}) \leq \frac{j}{k} + \eta_k. \quad \text{(B.108)}$$

Let $Q = F^{-1}$ be the quantile function. By definition of the quantile function, if $p \leq F(x)$ then $Q(p) \leq x$, and if $F(x) \leq q$ then $x \leq Q(q)$. Applying these relationships:

$$Q\left(\frac{j}{k} - \eta_k\right) \leq \Delta_{(j)} \leq Q\left(\frac{j}{k} + \eta_k\right) \quad \text{(B.109)}$$

Now we perform a change of index. We want to rewrite the sum in $I_k$ which uses index $j$ ranging from $k - r + 1$ to $k$. Let's set $j = k - i + 1$, so $i$ ranges from $1$ to $r$. This gives:

$$\frac{j}{k} = \frac{k-i+1}{k} = 1 - \frac{i-1}{k} \quad \text{(B.110)}$$

Substituting this into our bounds on $\Delta_{(j)}$:

$$Q\left(1 - \frac{i-1}{k} - \eta_k\right) \leq \Delta_{(k-i+1)} \leq Q\left(1 - \frac{i-1}{k} + \eta_k\right) \quad \text{(B.111)}$$

Summing over $i$ from 1 to $r$ and dividing by $k$:

$$\frac{1}{k}\sum_{i=1}^{r} Q\left(1 - \frac{i-1}{k} - \eta_k\right) \leq \frac{1}{k}\sum_{i=1}^{r} \Delta_{(k-i+1)} \tag{B.112}$$

$$= \frac{1}{k}\sum_{j=k-r+1}^{k} \Delta_{(j)} \tag{B.113}$$

$$= I_k \tag{B.114}$$

$$\leq \frac{1}{k}\sum_{i=1}^{r} Q\left(1 - \frac{i-1}{k} + \eta_k\right) \tag{B.115}$$

As $k \to \infty$, we know that $\eta_k \to 0$ almost surely by the Glivenko-Cantelli theorem. The points $u_i := 1 - \frac{i-1}{k}$ for $i = 1, \ldots, r$ where $r = \lfloor \lambda k \rfloor$, form a partition of the interval $[1 - \lambda, 1]$ as follows:

$$u_1 = 1, \ u_2 = 1 - \frac{1}{k}, \ u_3 = 1 - \frac{2}{k}, \ \ldots, \ u_r = 1 - \frac{r-1}{k} \approx 1 - \lambda. \tag{B.116}$$

As $k \to \infty$, the number of points $r = \lfloor \lambda k \rfloor$ also increases, and the distance between adjacent points $\frac{1}{k} \to 0$. Therefore, these points $\{u_i\}_{i=1}^{r}$ form an increasingly fine partition of the interval $[1 - \lambda, 1]$. For any fixed $u \in [1 - \lambda, 1]$, as $k \to \infty$, there exists a sequence of indices $i_k$ such that $u_{i_k} \to u$. Specifically, we can take $i_k = \lceil k(1 - u) + 1 \rceil$, which ensures $u_{i_k} \to u$ as $k \to \infty$. Our bounds for $I_k$ can be written as:

$$\frac{1}{k}\sum_{i=1}^{r} Q(u_i - \eta_k) \leq I_k \leq \frac{1}{k}\sum_{i=1}^{r} Q(u_i + \eta_k) \tag{B.117}$$

Since $Q$ is non-decreasing, it is bounded on the compact interval $[1 - \lambda - \beta, 1 + \beta]$ for some $\beta > 0$. Let

$$M := \sup_{t \in [1-\lambda-\beta, 1+\beta]} |Q(t)| < \infty, \tag{B.118}$$

then $|Q(u_i - \eta_k)| \leq M$ and $|Q(u_i + \eta_k)| \leq M$ for all $i$ and sufficiently large $k$.

The lower sum $\frac{1}{k}\sum_{i=1}^{r} Q(u_i - \eta_k)$ is a perturbed lower Riemann sum for the integral $\int_{1-\lambda}^{1} Q(u)\,du$, and similarly the upper sum is a perturbed upper Riemann sum. As $k \to \infty$, two things happen simultaneously, namely, (i) the mesh width $\frac{1}{k} \to 0$, so the Riemann sums converge to the integral and (ii) the perturbation $\eta_k \to 0$, so $Q(u_i \pm \eta_k) \to Q(u_i)$. By the Dominated Convergence Theorem, we have

$$\lim_{k \to \infty} \frac{1}{k}\sum_{i=1}^{r} Q(u_i - \eta_k) = \int_{1-\lambda}^{1} Q(u)\,du \tag{B.119}$$

$$\lim_{k \to \infty} \frac{1}{k}\sum_{i=1}^{r} Q(u_i + \eta_k) = \int_{1-\lambda}^{1} Q(u)\,du \tag{B.120}$$

Since $I_k$ is bounded between these two quantities that converge to the same limit, we have

$$\lim_{k \to \infty} I_k = \int_{1-\lambda}^{1} Q(u)\,du \quad \text{almost surely.} \tag{B.121}$$

**Combining all terms:** From our asymptotic analysis of each term, we have:

$$\lim_{k \to \infty} \max_{P_{XS}} \mathbb{E}_{P_{XS}}^{[k]}[f(X,S)] - \mathbb{E}_{P_X P_S}^{[k]}[f(X,S)] = \lim_{k \to \infty} I_k = \int_{1-\lambda}^{1} Q(u)\,du \quad \text{almost surely.} \tag{B.122}$$

Therefore:

$$\lim_{k \to \infty} D_{\text{gap}}^{[P_F]}(m, k, \lambda) = \int_{1-\lambda}^{1} Q(u)\,du. \tag{B.123}$$

$\square$

Theorem 4 implies that distributions with heavier tails imply larger values of the integral $\int_{1-\lambda}^{1} Q(u)du$, which in turn imply higher detection gaps. Indeed, fix $\lambda$. We say that a distribution $F_2$ is *(right-)heavier-tailed* than $F_1$ when $1 - F_2(x) \leq 1 - F_1(x)$ for all large $x$, equivalently $Q_2(u) \geq Q_1(u)$ for every $u$ in a neighbourhood of 1. In particular, for all $u \in [1 - \lambda, 1]$:

$$Q_2(u) \geq Q_1(u)$$

which implies:

$$\int_{1-\lambda}^{1} Q_2(u)du \geq \int_{1-\lambda}^{1} Q_1(u)du$$

Therefore, keeping the same confidence levels:

$$\text{heavier tail} \implies \text{larger} \int_{1-\lambda}^{1} Q(u)du \implies \text{larger guaranteed detection gap.}$$

The asymptotic detection gap, given by $\int_{1-\lambda}^{1} Q(u)du$, represents the average value of the quantile function over the upper $\lambda$ fraction of the distribution. This integral captures how much signal can be extracted from the tail of the score differences. Distributions whose upper tail places more mass far from zero (resulting in larger values of the integral $\int_{1-\lambda}^{1} Q(u)du$ directly increase the detection capability of the watermark. This explains why the choice of score distribution fundamentally impacts watermark detectability. Consequently, among distributions with the same mean and variance, the heavier-tailed ones yield strictly higher guaranteed detection rates. Therefore, if we constrain ourselves to the ensemble of sub-exponential probability distributions, by choosing something with a heavier tail than Gumbel, e.g., Log-Normal, we can achieve a better detection scheme, as we show in our experiments, cf. Figure 1.

**Remark 2.** *By using Bernstein's inequality for sub-exponential variables and Dvoretzky–Kiefer–Wolfowitz inequality instead of Glivenko-Cantelli, it is possible to show a non-asymptotic version of the theorem above:*

**Theorem 5** (Detection gap, formal)**.** *Let $0 < \lambda < 1$ be fixed, write $r = \lfloor \lambda k \rfloor$ and define*

$$I_k = \frac{1}{k}\sum_{s=1}^{r}\Delta_s, \qquad \bar{f}_{1,k} = \frac{1}{k}\sum_{s=1}^{k}f(1,s), \quad \bar{f}_{2,k} = \frac{1}{k}\sum_{s=1}^{k}f(2,s),$$

*where $\Delta_s = f(1,s) - f(2,s)$. For any confidence levels $\delta, \delta', \delta^{\ddagger} \in (0,1)$ set*

$$\varepsilon_k(\delta) = \sqrt{\frac{2\nu^2\log(1/\delta)}{k}} + \frac{b\log(1/\delta)}{k}, \qquad t_*(\delta') = \max\left\{2\nu\sqrt{\log\tfrac{1}{\delta'}}, \ 2b\log\tfrac{1}{\delta'}\right\},$$

$$\eta_k(\delta^{\ddagger}) = \sqrt{\frac{\log(2/\delta^{\ddagger})}{2k}}.$$

*Then, with probability at least $1 - \delta - \delta' - \delta^{\ddagger}$,*

$$\max_{P_{XS}} \mathbb{E}_{P_{XS}}^{[k]}\big[f(X,S)\big] - \mathbb{E}_{P_X P_S}^{[k]}\big[f(X,S)\big] \geq \frac{1}{k}\sum_{i=1}^{r}Q\Big(1 - \frac{i-1}{k} - \eta_k(\delta^{\ddagger})\Big). - \Big[2\varepsilon_k(\delta) + \tfrac{t_*(\delta')}{k}\Big],$$

$$\tag{B.124}$$

*where $Q = F^{-1}$ is the quantile function of $\Delta_s$, and $\mathbb{E}^{[k]}$ denotes an expectation where the side information alphabet is of size $k$.*

# C  Additional Information and Implementation Details

## C.1  Low-Entropy Distributions in LLMs

To motivate the low-entropy regime, we compute summary statistics of token distributions of popular open-weight LLMs. We consider the densities of three statistics: infinity norm (connects directly to min-entropy[4]), entropy, and L-2 norm, all calculated on the next token prediction along a collection of responses.

---

[4]An infinity norm of $\lambda$, i.e. $\max_s P(x) < \lambda$, translates directly to min-entropy constraint of $-\log\lambda$.

We observe that 90% LLM token distributions fall into the low-entropy regime we consider with infinity norm greater than 1/2, i.e. $\max_x P(x) \geq \frac{1}{2}$, across the three open LLMs (Llama2-7B[47], Llama3-8B[63], Mistral-7B) and two on popular prompt-generation datasets (Q&A tasks from Finance-QA[45] and coding tasks from LCC[46]). We show the histogram and CDF plots in Fig. D.14 and D.15.

## C.2 Theoretical effect of different tails of distributions

To further improve the detection performance, we go beyond binary score functions to explore the flexibility in the design space that continuous score distributions offer. Motivated by the Gumbel watermark [17], we observe that we can significantly improve detection by using continuous score distributions, particularly those with heavy tails. Recall that our minimax formulation considers the low-entropy regime ($\lambda \in [1/2, 1]$), where the worst-case token distribution $P_X$ has only two non-zero elements with values $\{\lambda, 1 - \lambda\}$. Working with this distribution, consider score matrices where each entry $f(x, s)$ is sampled independently from a distribution $P_F$ with zero mean and unit variance. We additionally assume that $f$ (and hence every $\Delta_s = f(1, s) - f(2, s)$) is *sub-exponential* with parameters $(\nu^2, b)$, i.e., $\mathbb{E}[e^{\lambda f}] \leq \exp\left(\frac{\nu^2 \lambda^2}{2}\right)$ for all $|\lambda| \leq 1/b$. Many distributions, such as Gamma, Gaussian, and Lognormal satisfy this property. This formulation leads to Theorem 4, which formally characterizes the achievable maximum detection gap for any $P_F$ for large $k$, in terms of quantile tail integrals of various candidate distributions as its score distribution $P_F$. We visualize the result in Fig. C.5.

In Fig. C.5, we present the quantile tail integrals of four different distributions: Lognormal, Gamma, Gaussian, and Gumbel. A higher value on the y-axis (quantile tail integral) indicates a greater detection gap under the low-entropy regime, which translates to a greater probability of detection under adversarial token distributions. In theory and as seen from Figure C.5, drawing score functions i.i.d from either the Lognormal or Gamma distribution outperforms that from Gumbel. Recall that the significance of adopting the Gumbel score function is that we have established its equivalence with the Gumbel watermark by [17] in Theorem 3. Although the Gamma distribution maximizes the detection gap in the worst-case regime, we observe that choosing $P_F$ to be lognormal achieves the highest detection accuracy in practice, which is what we eventually adopt for `HeavyWater`.

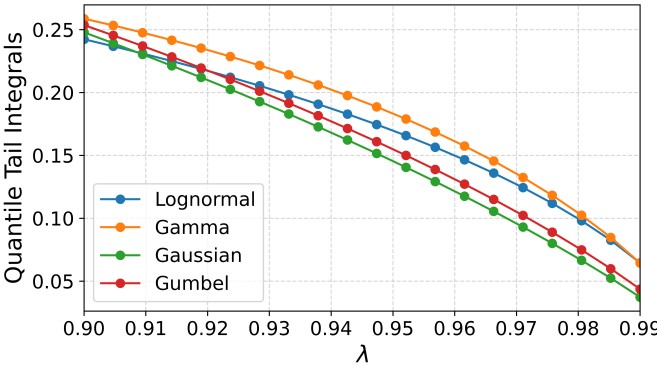

**Figure C.5:** Tail integrals of different *score difference distributions*: Higher the tail integral, better the detection.

## C.3 Q-ary Code

We provide a discussion of the direct extension of `SimplexWater` to go beyond binary-valued scores. Recall that `SimplexWater` uses the binary Simplex Code as its score function. For a given prime field-size $q$, a Q-ary `SimplexWater` adopts the corresponding Q-ary Simplex Code [14], which we define next. Besides the score function, the algorithm for Q-ary `SimplexWater` is identical to the binary case, which we have provided in the main text.

Given an alphabet of size $m$ and field size $q > 2$, the size of a Q-ary codeword is $q$ to the power of the ceiling of log $m$ with base $q$, i.e. $n = q^{\lceil \log_q m \rceil}$. For any $x, s \in [0 : n - 1]$, let $\mathsf{qary}(x)$, $\mathsf{qary}(s)$ denote their Q-ary representations respectively using $n$ bits. A Q-ary simplex code $f_{\mathsf{sim}} : [0 :$

$n-1] \times [1:n-1] \to [0:q-1$ is characterized by

$$f_{\text{sim}}(x,s) \triangleq \text{dot}(\text{qary}(x), \text{qary}(s)), \tag{C.125}$$

where $\text{dot}(\text{qary}(x), \text{qary}(s)) \triangleq \sum_{i=1}^{n} \text{qary}(x)_i \cdot \text{qary}(s)_i$ and $\text{qary}(v)_i$ denotes the $i$th bit in the Q-ary representation of $v$.

There are two main limitations of Q-ary `SimplexWater`, which motivated us to explore a continuous score function directly and ultimately led to `HeavyWater`. The first limitation is that we do not have optimality guarantees for the Q-ary code. Recall that to maximize watermark detection, the optimal code maximizes the $L_1$ distance. Simplex Code achieves the Plotkin bound, which maximizes the pair-wise Hamming distance between codewords. In the binary case, maximizing the Hamming distance and $L_1$ distance are equivalent, where $\mathbf{1}_{i,j} = |i-j|$; in the Q-ary case, this equivalence doesn't hold. Hence, we do not have an optimality guarantee in detection for Q-ary `SimplexWater`. The second limitation is that the size of Q-ary codewords is potentially very large, leading to memory issues in actual implementation on a GPU. Recall that in the binary case, we have $n = m - 1$. In the Q-ary case, however, the use of the ceiling function when converting $m$ to base-$q$ artificially inflates $n$, often far beyond the actual vocabulary size. For example, with $q = 7$ and a vocabulary size of $m = 100{,}000$, we compute $\lceil \log_7(100{,}000) \rceil \approx \lceil 5.92 \rceil = 6$, which corresponds to $n = 7^6 = 117{,}649$ codewords—more than 18% larger than the original vocabulary. Furthermore, the required alphabet size to implement a Q-arry code grows with $q$.

## C.4  Implementation Details

In this section we provide additional implementation details for `SimplexWater` and `HeavyWater`. Our code implementation employs the code from the two benchmark papers, namely WaterBench[5] [43] and MarkMyWords[6] [44].

**Score Matrix Instantiation**  We sample the score matrix once during the initialization stage of `HeavyWater` generation and it has shape $(m, k)$. Each row maps a vocabulary token to a cost vector over the side information space $\mathcal{S}$. This matrix is stored as a `torch.Tensor` and reused across watermarking calls. During generation, the score matrix is used as the cost matrix for Sinkhorn-based optimal transport computations.

**Normalization of $f$**  As described before, for each element in the vocabulary, $k$ samples for the score $f$ are drawn from a heavy-tailed distribution (log-normal in the experiment). To ensure numerical stability and suitability for optimal transport computations, each row of the score matrix (size vocab_size $* k$) is normalized to have zero mean and unit variance.

**Top-$p$ Filtering**  To reduce the computational cost of watermarking, top-$p$ filtering is applied prior to watermarking. Identical to the common definition of top-$p$, we take the minimal set of tokens whose cumulative softmax probability exceeds a threshold (e.g., $p = 0.999$). The watermarking algorithm is then restricted to this filtered subset. We emphasize that, in the considered experiments, top-$p$ filtering was applied to all considered watermarks to maintain consistency in the experimental setting.

**Detection Algorithm**  We outline in Algorithm 2 a standard watermark detection algorithm that employs a threshold-test with a score matrix F. Given the sampled token $x$ and side information $s$, the corresponding score is $\mathrm{F}_{x,s} = f(x,s)$, which is obtained in a similar fashion to its construction in generation: If `SimplexWater` is used, then it is obtained from the Simplex code, and if `HeavyWater` it is randomly sampled using the shared secret key as the initial seed. This maintains the generation of the same cost matrix F on both ends of generation and detection. A watermark is detected if the sum of scores exceeds a certain predetermined threshold.

---

[5]`https://github.com/THU-KEG/WaterBench`
[6]`https://github.com/wagner-group/MarkMyWords`

---

**Algorithm 2 Detection** using a threshold-test with a score matrix

---

1: **Input:** Token sequence $x^n$, side information $s$, seed, score matrix $F \in \mathbb{R}^{m \times k}$
2: **Outputs:** $p$-value based detection outcome
3: **for** $t = 1$ to $T$ **do**
4:     Compute score $\phi_t := F_{x_t,s}$
5: **end for**
6: $Z \leftarrow \sum_{t=1}^{T} \phi_t$
7: **if** $Z > \tau$ **then return** "Watermark Detected"
8: **else return** "No Watermark"

---

**Fresh Randomness Generation**    Fresh randomness is crucial to ensure side information is sampled independently from previous tokens to avoid seed collision. Our implementation supports several seeding strategies. In majority of our experiment, we use the 'fresh' strategy, which generates a unique seed for each token by incrementing a counter and combining it with the shared secret key. This allows both ends to share the same seed that dynamically changes, but is independent of previously generated tokens. Our implementation of `HeavyWater` and `SimplexWater` also allows for various forms of temporal or token-based encoding strategies, such as sliding-window hashing.

### C.5 Information on Optimal Transport and Sinkhorn's Algorithm

In this section, we provide preliminary information on the considered OT problem and its solution through Sinkhorn's algorithm. Let $p \in \Delta_m$ and $q \in \Delta_k$ be two probability vectors. For a given cost matrix $C \in \mathbb{R}^{m \times k}$, the OT between $p$ and $q$ is given by

$$\mathsf{OT}(p,q) = \min_{P \in \Pi_{p,q}} \sum_{i=1}^{m} \sum_{j=1}^{k} C_{i,j} P_{i,j},$$

where $\Pi_{p,q}$ is the set of joint distributions with marginals $p$ and $q$, which we call *couplings*. Efficiently solving the optimization $\mathsf{OT}(p,q)$ is generally considered challenging [36]. To that end, a common approach to obtain a solution efficiently is through entropic regularization. An entropic OT (EOT) with parameter $\epsilon$ is given by

$$\mathsf{OT}_\epsilon(p,q) = \min_{P \in \Pi_{p,q}} \left( \sum_{i=1}^{m} \sum_{j=1}^{k} C_{i,j} P_{i,j} - \epsilon H(P) \right),$$

where $H(P) \triangleq -\sum_{i,j} P_{i,j} \log(P_{i,j})$ and $C$ is the OT cost. The EOT is an $\epsilon$-strongly convex problem, which implies its fast convergence to the *unique* optimal solution. However, the EOT provides an approximate solution which converges to the unregularized solution with rate $O(\epsilon \log(1/\epsilon))$.

One of the main reasons EOT has gained its popularity is due to Sinkhorn's algorithm [64], which is a matrix scaling algorithm that has found its application to solve the dual formulation of the EOT problem [19]. Sinkhorn's algorithm looks for a pair of vectors $u, v$ that obtain the equality

$$P^* = \mathsf{diag}(u) K \mathsf{diag}(v),$$

where $P^*$ is the EOT solution and $K = \exp(-C/\epsilon)$ is called the *Gibbs Kernel*. Consequently, Sinkhorn's algorithm follows from a simple iterative procedure of alternately updating $u$ and $v$. The steps of Sinkhorn's algorithm are given in Algorithm 3. Having solved Sinkhorn's algorithm, we obtain the optimal EOT coupling. When using Sinkhorn's algorithm, the stopping criteria is often regarding the marginalization of the current coupling against the corresponding marginal, i.e., we check wether $\left\| u \odot (Kv) - p \right\|_1 \le \delta$ where $\triangleq \sum_{i=1}^{n} \left| u_i (Kv)_i - p_i \right|$ and $\delta > 0$ is some threshold. In our implementation, we solve Sinkhorn's algorithm using the Python optimal transport package [65].

---

**Algorithm 3** Sinkhorn's Algorithm for Entropic OT

---

1: **Input:** Marginals $P_X \in \Delta_m$, $P_S \in \Delta_k$, cost matrix $C \in \mathbb{R}^{m \times k}$ regularization parameter $\varepsilon > 0$, Threshold $\delta > 0$.
2: **Output:** Optimal coupling $P \in \mathbb{R}_{\geq 0}^{n \times m}$
3: Calculate kernel $K \leftarrow \exp(-C/\varepsilon)$
4: $u \leftarrow \mathbf{1}_m, \quad v \leftarrow \mathbf{1}_k$
5: **while** $\left\| u \odot (K\,v) - P_X \right\|_1 > \delta$ **do**
6: $\quad u \leftarrow a/(K\,v)$                      ▷ element-wise division
7: $\quad v \leftarrow b/(K^\intercal u)$
8: **end while**
9: $P \leftarrow \mathrm{diag}(u)\,K\,\mathrm{diag}(v)$
10: **return** $P$

---

# D  Additional Numerical Results and Ablation Study

## D.1  Ablation Study

We perform an ablation study to investigate the effect of various hyperparameters set in `SimplexWater` and `HeavyWater`. The ablation study is performed in a curated subset of prompts from the Finance-QA dataset [45] considered in Section 5.

**Sinkhorn Algorithm Parameters.** We study the effect of Sinkhorn's algorithm's parameter on the performance of the proposed watermarking scheme. We note that, while the Sinkhorn's algorithm is set with a predetermined maximum iterations parameters, in the considered experiments the algorithm runs until convergence. We study this effect through three cases.

1. We analyze the effect of Sinkhorn's algorithm's regularization parameter $\epsilon$ on the overall runtime. While lower values of $\epsilon$ provide solution that are closer to the underlying OT solution, often a smaller value of $\epsilon$ required more time for convergence of the algorithm. As seen from Figure D.6a, as expected, smaller $\epsilon$ increase overall runtime. In our experiments we chose $\epsilon = 0.05$ which resulted in overall satisfactory performance, while incurring mild runtime overhead.

2. We analyze the effect of the error threshold on runtime. The lower the error threshold, the higher the higher the accuracy in the solution and the higher the overall algorithm runtime. This is indeed the case, and the effect on the watermarking procedure runtime is visualized in Figure D.6b.

3. We analyzed the effect of the error threshold on the watermarked distribution cross entropy (see Section 5 for definition). As seen from Figure D.6c, while a lower threshold results with a higher runtime, the improvement on the cross entropy, which is a proxy for textual quality, becomes negligible from some point. In our experiments, we chose a threshold value of $10^{-5}$, which demonstrated the best performance between runtime, cross entropy and detection.

## D.2  Impact of Non i.i.d. Side Information Generation

As we previously mentioned, most of the considered experiments in Section 5 operate under a 'fresh randomness' scheme, in which we try to replicate independence between the random side information and the LLM net token distribution. However, in practice various hashing scheme are employed, often with the purpose of increasing the overall watermarking scheme's robustness to attacks. Such hashing schemes aggregate previous tokens (using some sliding window with context size $h$) and a shared secret key $r \in \mathbb{N}$. We are interested in verifying that indeed, robustness-driven seed generation scheme do not degrade the performance of our methods.

To that end, in this section we test the effect of various popular hashing schemes in the performance of our watermarks. As both `SimplexWater` and `HeavyWater` follow the same watermarking algorithm we anticipate them to demonstrate similar dependence on the hashing scheme. We therefore prioritize

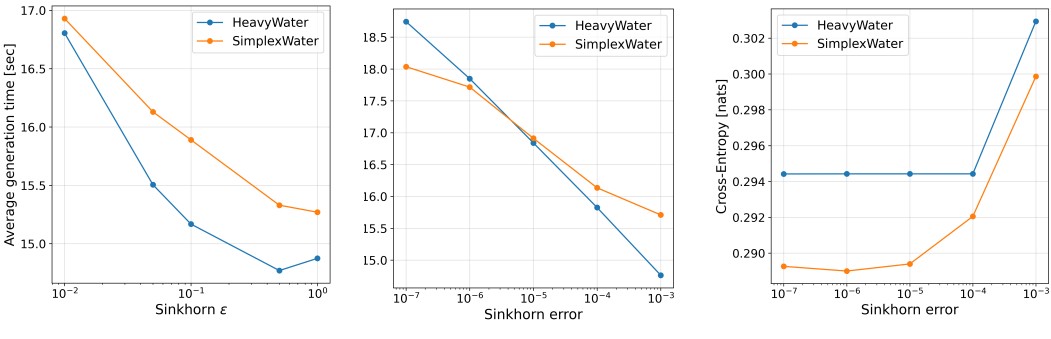

**(a)** Generation time vs. Sinkhorn regularization parameter $\epsilon$.

**(b)** Generation time vs. Sinkhorn error threshold.

**(c)** Cross Entropy of watermarked text vs. Sinkhorn error threshold.

**Figure D.6:** Effect of Sinkhorn Algorithm's parameters on Runtime and distortion.

an extensive study on a single watermark - `SimplexWater`. For a given sliding window size $h$, we consider the following seed generation functions:

1. min-hash, which takes the minimum over token-ids and multiplies it with the secret key, i.e. seed $= \min(x_{t-1}, \ldots, x_{t-h}) \cdot r$.

2. sum-hash, which takes the sum of the token-ids and multiplies it with the secret key, i.e. seed $= \mathsf{sum}(x_{t-1}, \ldots, x_{t-h}) \cdot r$.

3. prod-hash, which takes the product of the token-ids and multiplies it with the secret key, i.e. seed $= \mathsf{prod}(x_{t-1}, \ldots, x_{t-h}) \cdot r$.

4. Markov-1 scheme, which considers $h = 1$.

We present performance across the aforementioned schemes, considering several values of $h$. As seen from Figure D.8, the change of seed generation scheme is does have a significant effect on the overall detection-distortion tradeoff. Furthermore, as emphasize in Figure D.7, the size of the sliding window also results in a negligible effect on the watermark performance.

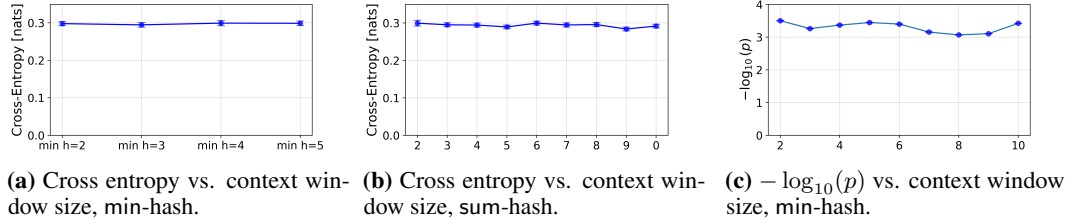

**(a)** Cross entropy vs. context window size, min-hash.

**(b)** Cross entropy vs. context window size, sum-hash.

**(c)** $-\log_{10}(p)$ vs. context window size, min-hash.

**Figure D.7:** Seed Ablation: The effect of the context window is is negligible on the performance of `SimplexWater`.

### D.3 Experiment: Robustness To Textual Attacks

The watermarks in this paper are obtained by optimizing a problem that encodes the tradeoff between detection and distortion under worst case distribution. To that end, the proposed watermarks are not theoretically optimized for robustness guarantees. However, robustness is often a byproduct of the considered randomness generation scheme, as text edit attacks mainly effect the context from which the seed is generated. A discrepancy in the seed results in a discrepancy in the shared side information sample $s$. However, regardless of the seed generation scheme, one has to choose a score function and a watermarked distribution design.

In this section, we show that, while not optimized for robustness directly, `SimplexWater` and `HeavyWater` demonstrate competitive performance in terms on robustness to common textual edit attacks. We consider the setting from the watermarking benchmark MarkMyWords [44]. We compare our performance with the Red-Green watermark and the Gumbel watermark. We choose

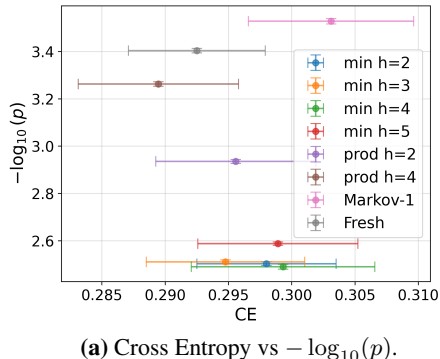
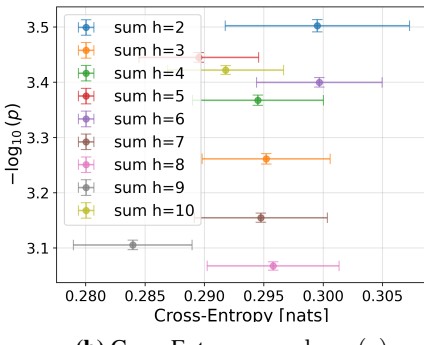

**(a)** Cross Entropy vs $-\log_{10}(p)$.

**(b)** Cross Entropy vs $-\log_{10}(p)$.

**Figure D.8:** Seed Ablation visualized in the detection-distortion plane. It is visible that the performance of our watermark is consistent across an array of hashing scheme and context window sizes.

the value of $\delta$ for the Red-Green watermark such that its cross-entropy distortion is comparable with `SimplexWater`, `HeavyWater` and Gumbel ($\delta = 1$).

We consider three attacks:

1. A Lowercase attack, in which all the characters are replaced with their lowercase version.

2. A Misspelling attack, in which words are replaced with a predetermined misspelled version. Each word is misspelled with probability $0.1$.

3. A Typo attack, in which, each character is replaced with its neighbor in the QWERTY keyboard. A character is replaced with probability $0.05$.

As seen in Figure D.9, our schemes demonstrate strong robustness under the considered attacks, resulting in the highest detection capabilities in 3 out of 4 cases and competitive detection power in the 4th. This implies that, even though `SimplexWater` and `HeavyWater` are not designed to maximize robustness, the resulting schemes show competitive resilience to common text edit attacks.

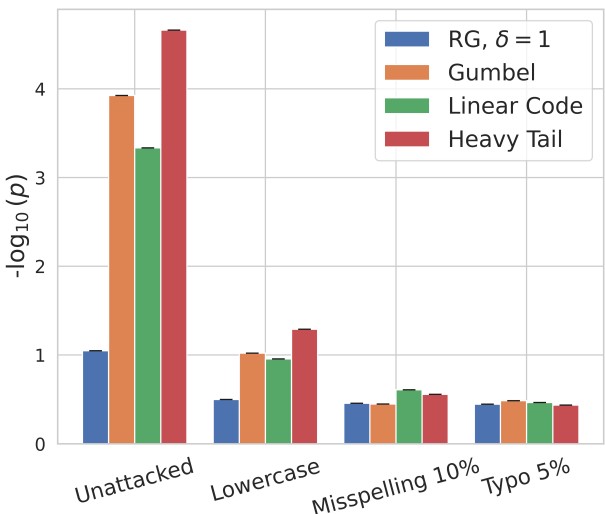

**Figure D.9:** Robustness to attacks — `HeavyWater` demonstrates equal or superior detection performance, as measured by $-\log_{10}(p)$, across a variety of attacks involving edits to generated outputs.

### D.4 Computational Overhead

We analyze the computational overhead induced by the considered watermarking scheme. Theoretically, Sinkhorn's algorithm has an iteration computational complexity of $O(km)$ for token vocabulary of size $|\mathcal{X}| = m$ and side information of alphabet $|\mathcal{S}| = k$ due to its vector-matrix operations. In practice, watermarking is a single step within the entire next token generation pipeline.

We analyze the computational overhead induced by applying `SimplexWater` and `HeavyWater`. Figure D.10 shows the overhead of watermarking in a few common watermarks - Red-Green [15], Gumbel [17], Inverse-transform [13], SynthID [11] and our watermarks. It can be seen that The Gumbel, Inverse transform and Red-Green watermarks induce a computational overhead of $\sim 10\%$, while `SimplexWater`, `HeavyWater` and SynthID induce an overhead of $\sim 30\%$. While this overhead is not negligible, our methods demonstrate superior performance over considered methods. However, replacing a 'fast', yet 'weaker' watermark with ours boils down to a difference in $\sim 20\%$ increase in generation time. We consider an implementation of the SynthID through vectorized tournament sampling with a binary score function and $15$ tournament layers, which is the method reported in the main text experiments. As we previously mentioned, we consider top-$p$ sampling with $p$=0.999. We note that, in many text generation schemes, lower top-$p$ values, which accelerate Sinkhorn's algorithm's runtime, thus further closing the computational gap.

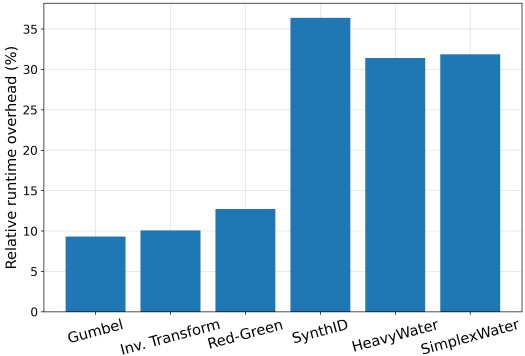

**Figure D.10:** Computational overhead over unwatermarked text generation, Llama2-7b.

### D.5 Experiment: Alternative Quality Metrics for Textual and Coding Tasks

This paper focused on distortion as the proxy for textual quality. This is a common practice in watermarking (e.g. [11, 13, 16, 17, 24, 25]). Distortion is measured by the discrepancy between the token distribution $P_X$ and the expected watermarked distribution $\mathbb{E}_S[P_{X|S}]$. In practice, distortion and textual quality are often measured with some perplexity-based measure (e.g. cross-entropy in this paper). However, as explored in the WaterBench benchmark [43], such measures are not guaranteed to faithfully represent degradation in textual quality.

To that end, WaterBench proposed an array of alternative generation metrics, whose purpose us to evaluate the quality of generated watermarked text, and are tailored for specific text generation tasks. We consider $4$ datasets from the WaterBench benchmark [43]:

1. Longform QA [54]: A dataset of 200 long questions-answer generation prompts. The considered generation metric is the ROGUE-L score.

2. Knowledge memorization: A closed-ended entity-probing benchmark drawn from KoLA [66], consisting of 200 triplets sampled at varying frequencies from Wikipedia the test an LLM's factual recall. The considered generation metric is the F1 score as it is a factual knowledge dataset.

3. Knowledge understanding [67]: A dataset of 200 questions that demonstrate the LLM's understanding of various concepts. The considered generation metric is the F1 score as it is a factual knowledge dataset.

4. Multi-news summarization: A collection of 200 long news clusters, coupled with summarization prompts. The score here is the ROGUE-L score.

As seen in Table D.3, our watermarks maintain competitive performance in the considered set of textual generation tasks, even under alternative text generation evaluation metrics.

To better evaluate quality of generated watermarked code, we include two coding-centric evaluation metrics that assess different aspects of code quality.

**Pass@K Evaluation (Functional Correctness)** We evaluate functional correctness using the HumanEval dataset [46], which is a popular benchmark for code generation. Pass@K measures the empirical probability of a solution passing all unit tests among K generated solutions. We measure pass@k with $k \in 1, 5, 10$, where we execute the generated code against unit tests provided in the dataset in a sandbox environment. As shown in the table below, our methods outperformed the competitors: (i) SimplexWater achieves the highest pass@5 (22.7%), outperforming all baselines including unwatermarked text (20.7%) and (ii) HeavyWater achieves the highest pass@10 (27.8%), nearly matching unwatermarked performance (28.1%). In all cases, performance is comparable to unwatermarked code (see Table 1).

**Edit_Sim (Edit Similarity)** We use this metric on the LCC coding dataset [46] which focuses on code completion: generating the next few lines of code given a long context. Since the dataset is sourced from GitHub, human-written ground-truth completions are available. Edit_Sim is defined as the Levenshtein similarity between generated and ground-truth code.

Let $s_1, s_2 \in \Sigma^*$ be two strings over an alphabet $\Sigma$. The *Levenshtein distance* $d_L(s_1, s_2)$ is defined as the minimum number of single-character edit operations (insertions, deletions, or substitutions) required to transform $s_1$ into $s_2$. The *Levenshtein similarity* (normalized form) is then defined as:

$$\text{LevSim}(s_1, s_2) = 1 - \frac{d_L(s_1, s_2)}{\max(|s_1|, |s_2|)}$$

where $|s|$ denotes the length of string $s$. Thus, $\text{LevSim}(s_1, s_2) \in [0, 1]$, with $\text{LevSim}(s_1, s_2) = 1$ if and only if $s_1 = s_2$.

We benchmark our watermarking methods against prior work using this metric. The results are provided in the Table D.4. Specifically, HeavyWater ties for best performance 0.52. These two metrics support that HeavyWater and SimplexWater have no significant impact on the quality of generated code relative to unwatermarked code, and perform favorably when compared to competing methods both in terms of detection accuracy and generation quality.

**Code Examples.** We also include an example of generated code before and after watermarking with HeavyWater.

**Unwatermarked Output:**

```python
def has_close_elements(numbers: List[float], threshold: float) -> bool:
    # Sort the list of numbers
    numbers = sorted(numbers)

    # Find the first two numbers in the list
    that are closer to each other than the threshold
    min_distance = float('inf')
    for i in range(1, len(numbers)):
        distance = abs(numbers[i] - numbers[i - 1])
        if distance < min_distance and distance < threshold:
            min_distance = distance

    # If we found a pair of numbers that are
    closer to each other than the threshold, return True
    return min_distance < threshold
```

**Unit Tests result:** Pass $\rightarrow$ Pass@1 = 100% on this example prompt.

**HeavyWater's Watermarked Output:**

```python
def has_close_elements(numbers: List[float], threshold: float) -> bool:
    close_pairs = []
    for i in range(len(numbers)):
```

```
4            closest = None
5            closest_distance = float("inf")
6            for j in range(i+1, len(numbers)):
7                distance = abs(numbers[i] - numbers[j])
8                if distance < closest_distance:
9                    closest = numbers[j]
10                   closest_distance = distance
11           if closest and closest_distance <= threshold:
12               close_pairs.append((numbers[i], closest))
13       return len(close_pairs) > 0
```

**Unit Tests result:** Pass → Pass@1 = 100% on this example prompt.

**Example: Code Completion under Long Context**

This example was shortened for brevity.

**Prompt: Please complete the code given below.**

```python
1  #!/usr/bin/env python
2  # -- coding: utf-8 --
3
4  from HttpUtils import App, buildOpener
5
6  class Device(object):
7      def __init__(self, token):
8          self.token = token
9          self.app = App()
10
11     def check_inspection(self):
12         data = self.app.check_inspection()
13         return data
14     [abbreviated]
15
16 class Exploration(object):
17     def __init__(self, app):
18         self.app = app
19
20     def getAreaList(self):
21         data = self.app.exploration_area()
22         return data
23     [abbreviated]
24
25 class User(object):
26     [abbreviated]
27
28 class RoundTable(object):
29     [abbreviated]
30
31 class Menu(object):
32     [abbreviated]
33
34 if __name__ == "__main__":
35     from config import deviceToken, loginId, password
```

**Answers provided by different methods:**

**Human Answer / Ground Truth:**

```
1  device = Device(deviceToken)
```

**No Watermark** → Edit_Sim = 0.94

```
1  dev = Device(deviceToken)
```

**HeavyWater / Inverse-Transform / SimplexCode** → Edit_Sim = 1

```
1  device = Device(deviceToken)
```

**Red/Green** → Edit_Sim = 0.9

```
1  d = Device(deviceToken)
```

## D.6 Alternative Detection Metric

In this section we provide results on an additional detection metric. We consider the detection probability under a false-alarm (FA) constraint. As we consider watermarks from which $p$-values can be calculated, we can impose such a FA constraint. For a given set of responses obtained from a dataset of prompts, we are interested in calculating an estimate of the detection probability at some FA constraint, given a set of $p$-values, each calculated for one of the responses. We obtain an estimate of the detection probability at a given FA constraint by taking the ratio of responses whose $p$-value is lower than the proposed $p$-value threshold, over the total number of responses.

We provide results on the FinanceQA dataset using Llama2-7b. We consider several FA values and visualize the resulting tradeoff curves in Figures D.11 and D.12. To obtain error-bars, we consider the following bootstrapping technique: Out of the 200 responses, we randomly sample 200 subsets with 150 responses and calculate the corresponding metric. From the set of 150 results we provide error-bars, considering the average value and standard deviation. It can be seen that the trends presented in Figures 1 and 3a are preserved under the considered detection metric.

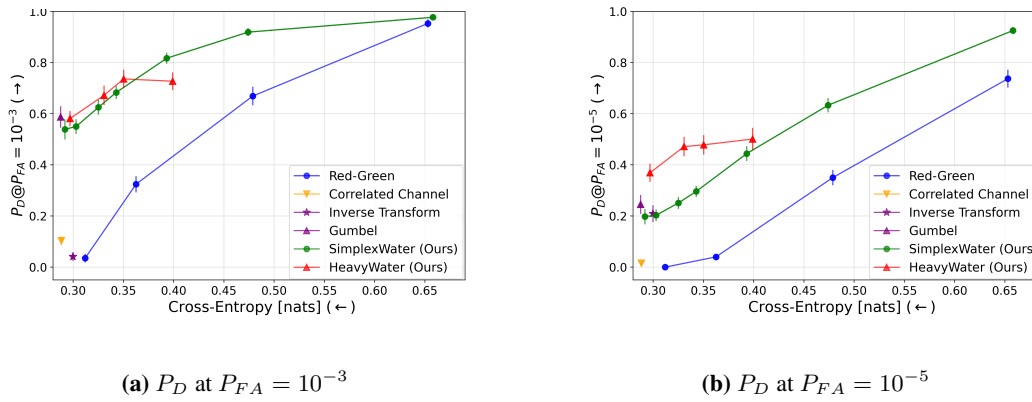

(a) $P_D$ at $P_{FA} = 10^{-3}$          (b) $P_D$ at $P_{FA} = 10^{-5}$

**Figure D.11:** Detection probability at a given false alarm constraint. LLama2-7b, Finance-QA dataset.

## D.7 Additional Detection-Distortion Tradeoff Results

We provide results that explore the detection-distortion tradeoff, in addition to ones presented in Fig. 1 and Fig. 3a. We run three models (Llama2-7b, Llama3-8b, Mistral-7b) on two tasks (Q&A and

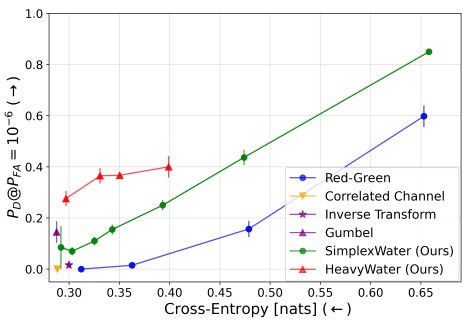

**Figure D.12:** $P_D$ at $P_{FA} = 10^{-6}$, LLama2-7b, Finance-QA dataset.

coding). We employ the popular Q&A dataset, FinanceQA, and code-completion dataset LCC. Fig. 1 shows the result for Llama2-7b on Q&A, while Fig. 3a shows the result for Mistral-7B on coding. In Fig. D.13, we present this tradeoff over the remaining datasets and LLMs.

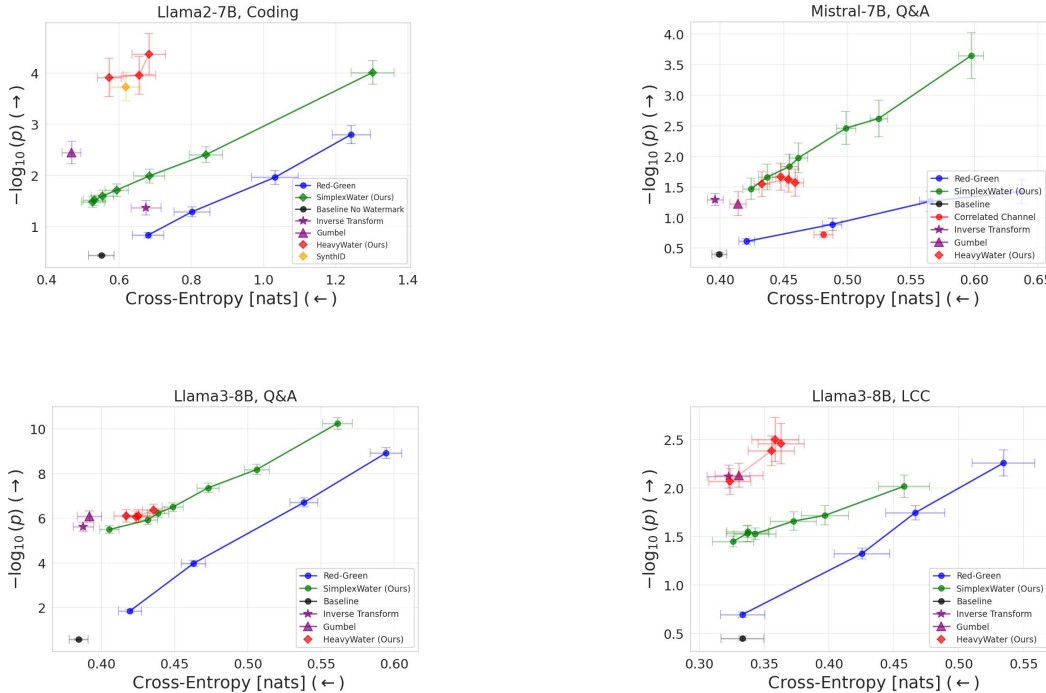

**Figure D.13:** Detection-distortion tradeoffs on multiple models and tasks.

Table D.3: Performance of Watermarking Methods across Four Datasets

| Dataset | Watermark | Gen. Metric ↑ | % Drop in GM ↓ | $-\log_{10} p$ ↑ |
|---|---|---|---|---|
| **Longform** | Gumbel | 21.20 | -0.856 | 8.006 |
| | HeavyWater | 21.48 | -2.188 | **8.089** |
| | Simplex | 21.90 | **-4.186** | 4.985 |
| | Inv. Tr. | 21.27 | -1.189 | 3.687 |
| | RG, $\delta = 1$ | 21.25 | -1.094 | 1.456 |
| | RG, $\delta = 3$ | 21.19 | -0.809 | 7.078 |
| **Memorization** | Gumbel | 5.66 | -2.536 | 1.085 |
| | HeavyWater | 5.73 | **-3.804** | **1.605** |
| | Simplex | 5.71 | -3.442 | 0.977 |
| | Inv. Tr. | 5.38 | 2.536 | 0.792 |
| | RG, $\delta = 1$ | 5.35 | 3.080 | 0.482 |
| | RG, $\delta = 3$ | 5.82 | 5.435 | 0.912 |
| **Understanding** | Gumbel | 33.42 | **-9.574** | 0.396 |
| | HeavyWater | 32.59 | -6.852 | 0.308 |
| | Simplex | 31.50 | -3.279 | 0.920 |
| | Inv. Tr. | 27.93 | 8.426 | **1.045** |
| | RG, $\delta = 1$ | 32.96 | 8.066 | 0.184 |
| | RG, $\delta = 3$ | 33.83 | 10.918 | 0.300 |
| **MultiNews** | Gumbel | 25.69 | 2.579 | 3.172 |
| | HeavyWater | 25.67 | 2.655 | 3.491 |
| | Simplex | 25.86 | **1.934** | 2.701 |
| | Inv. Tr. | 25.74 | 2.389 | 1.586 |
| | RG, $\delta = 1$ | 25.85 | 1.940 | 0.963 |
| | RG, $\delta = 3$ | 25.74 | 2.389 | **3.781** |

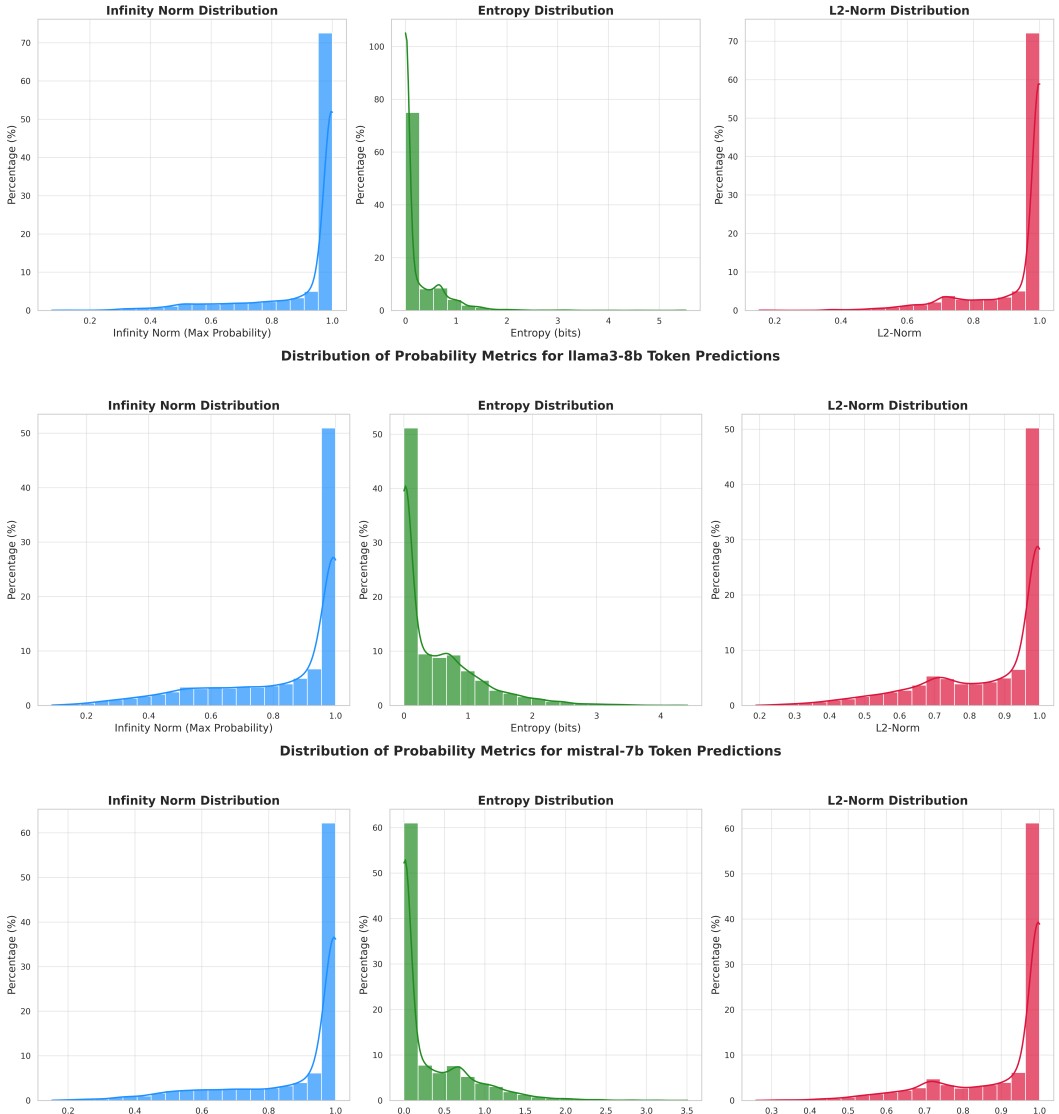

**Figure D.14:** Histograms of statistics of token distributions on **Q&A** dataset. 90% token distributions fall into the low-entropy regime with infinity norm greater than $1/2$, i.e. $\max_x P(x) \geq \frac{1}{2}$.

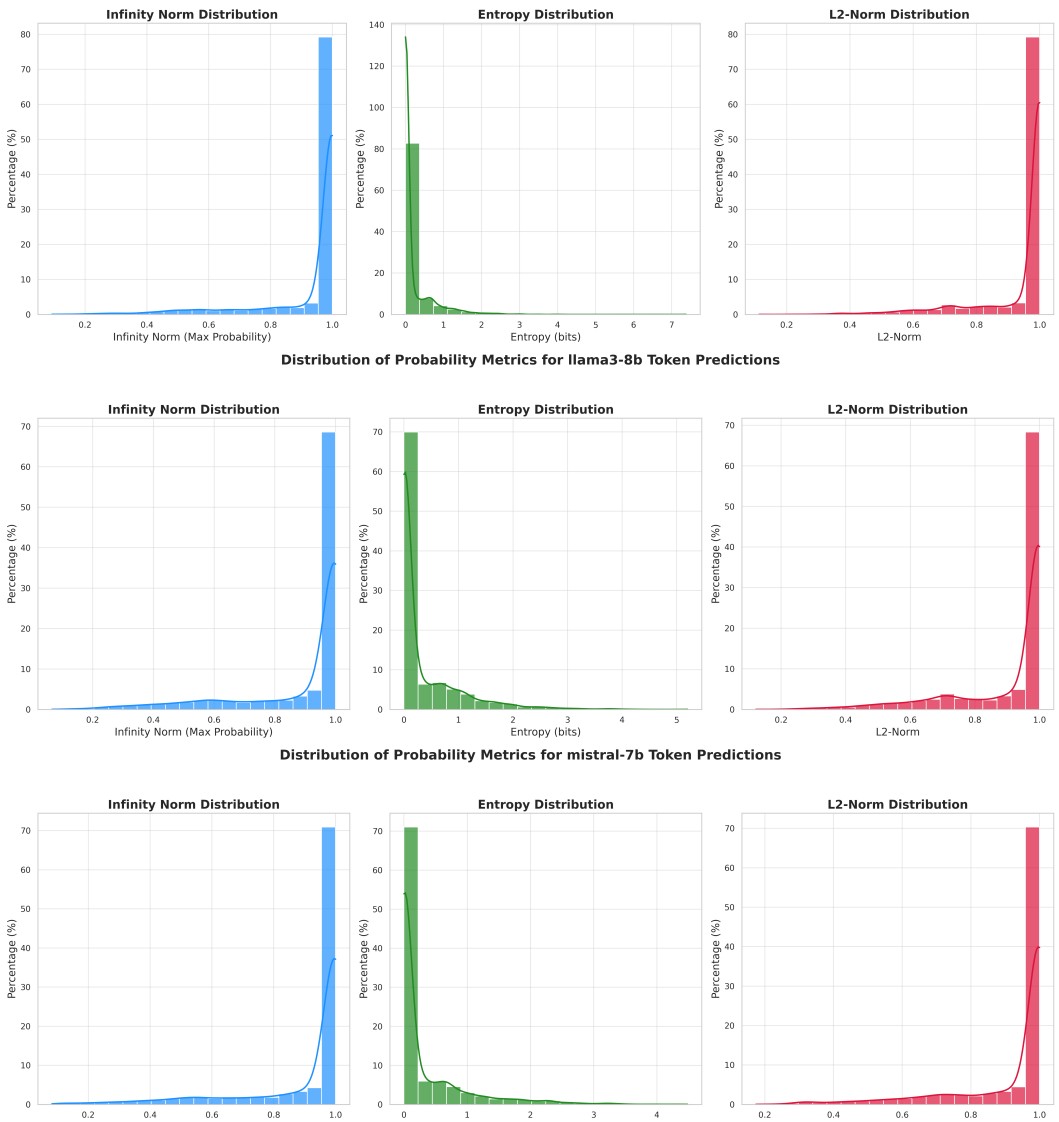

**Figure D.15:** Histograms of statistics of token distributions on **coding** dataset. 93% token distributions fall into the low-entropy regime with infinity norm greater than $1/2$, i.e. $\max_x P(x) \geq \frac{1}{2}$.

Table D.4: Results on LCC Code Completion Dataset (Edit_Similarity, higher is better)

| Watermark Method | Edit_Similarity ↑ |
|---|---|
| No Watermark | 0.45 |
| Gumbel | 0.52 |
| **HeavyWater (Ours)** | **0.52** |
| **SimplexWater (Ours)** | 0.44 |
| Inverse Transform | 0.45 |
| Red/Green $\delta = 3$ | 0.43 |

