# OpenReview forum: "HeavyWater and SimplexWater: Distortion-free LLM Watermarks for Low-Entropy Distributions"
_NeurIPS.cc/2025/Conference — NeurIPS 2025 poster_

### Official Review · Reviewer_dQ8C · 2025-07-02

**Clarity:** 3
**Significance:** 3
**Originality:** 3
**Rating:** 5
**Confidence:** 3

**Summary:**

This paper aims to design LLM watermarks that are effective even in low-entropy conditions. They model watermarks as modifying the distribution of each token in the response using some side information (for example, in the Kirchenbauer et al. scheme this side info is the output of the hash function). To do so, they formulate constructing a good watermark as an optimization problem which essentially involves maximizing the expectation of the score function applied to the response and the actual side information, and minimizing the expectation of the score function applied to the response and independent side information. This gap in expectations arises because the watermark modifies the response to be correlated with the side information.

There are two important choices this paper considers in instantiating such schemes: the score function, and the way in which the watermark modifies the distributions. The way in which the watermark modifies the distributions is derived from the score function, using Sinkhorn’s algorithm. This paper takes constructing the side information to be outside its scope. This paper proposes two instantiations of the scheme: HeavyWater and SimplexWater. SimplexWater uses a binary simplex score function rooted in coding theory. HeavyWater is an improved version of SimplexWater that uses a continuous score function. The schemes also include a tunable parameter that controls how much distortion the watermark introduces.

In addition to proving properties of the proposed schemes, this paper shows empirically that their detectability out-performs existing schemes, especially on coding models.

**Questions:**

- How computationally expensive is the watermark generation algorithm/Sinkhorn’s algorithm?
- In Figure 3(b), what is being measured on the y-axis? It is labeled “improvement,” but under what metric is this improvement?
- The experiments seem to focus on two kinds of tasks: Finance-QA (Q&A) and LCC (coding). There are many other low-entropy scenarios, though. What evidence do you have that your watermark performs well in low-entropy scenarios in general?

**Ethical Concerns:**

["NO or VERY MINOR ethics concerns only"]

**Final Justification:**

This paper nicely formalizes the problem of constructing a good watermark in terms of an optimization task. It then shows that solving this problem yields practical watermarks.

The authors' rebuttal assuaged my primary concern, which was efficiency. The watermark involves solving an optimization problem when generating each token. As the authors point out, this optimization problem is over the next-token distribution (importantly, not the model itself). Therefore, this complexity does not grow as the model grows. So for larger models, this optimization cost becomes small compared to the cost required to run the model.
It would be great to see more of this efficiency discussion in the final paper.

The authors also included many experiments addressing the generality of the scheme in performing well in many low-entropy scenarios. They raised my attention to these in the rebuttal.

Overall, this is an interesting theoretical formulation of a practical problem, with practical applications. I recommend acceptance.

**Limitations:**

Yes

**Quality:**

3

**Strengths And Weaknesses:**

Strengths
- This paper nicely formalizes the problem of constructing a good watermark in terms of an optimization task. It then shows that solving this problem yields practical watermarks.
- This paper has both theoretical and empirical evidence for the effectiveness of the watermark.
- The paper is clearly written.

Weaknesses
- The watermark scheme involves solving a *different instance* of an optimization problem to determine how to modify the distribution of each token in the response. In practice, this seems like it would be quite expensive. In contrast, existing schemes involve changing each distribution according to a simple rule, such as increasing the probabilities of tokens on a green list.
- Due to the above, the paper should include more discussion about the efficiency of the proposed scheme.

Overall, the paper shows a nice theoretical perspective on the task of watermarking. This perspective yields (seemingly) practical schemes that perform well empirically, in certain low-entropy settings. Since low-entropy settings have posed a challenge to watermarks, this contribution has good significance if the schemes are truly practical (see question about efficiency below). The quality seems reasonable, and the paper is reasonably clear. The main question I have is about efficiency.

---

> ### Author Rebuttal · Authors · 2025-07-31
>
> Thank you for the thoughtful review and encouraging feedback! We address your comments and questions below, including those related to efficiency and entropy. Please note that computational overhead measurements were included in our Supplementary Material (Appendix D). These results show that the computational overhead of our watermark implementation is comparable to existing watermarks. We discuss these results below and will highlight them in the main body of the paper.
>
> We are happy to answer any follow-up questions you may have.
>
> ---
> ### **Q1 and Weaknesses: Efficiency of the algorithm. “How computationally expensive is the watermark generation algorithm/Sinkhorn’s algorithm?”**
>
> We compared the computational overhead of our watermarking algorithms (which involve OT/Sinkhorn iterates) against several popular watermarks, including Red-Green, Gumbel, Inverse-Transform, and SynthID. The results are included in Appendix D.4 and Figure D.10 of our Supplementary Material.
>
> To summarize, we reported that schemes with closed-form watermarked next-token distributions (including Red-Green, Gumbel, and Inverse-transform) have a generation overhead of approximately 10% in the Llama2-7B model. Methods that equire some optimization to watermark the next-token distribution – including Deepmind’s \textit{SynthID} and our’s SimplexWater and HeavyWater – incur a higher overhead of approximately 30%, with SynthID incurring the highest cost across all methods.
>
> We highlight that this overhead cost is relative to the generation time of a Llama2-7B model. This relative overhead will be **significantly amortized** for larger models, as the watermarking computation time **remains constant regardless of model size** (it is a logit post-processor), whereas the next-token computation time increases.
>
> ---
> ### **Q2: Figure 3(b) y-axis metric for improvement**
>
> In Figure 3(b), we measure the improvement of SimplexWater and HeavyWater’s detection power relative to the Red-Green watermark [15]. Specifically, we measure detection improvement in terms of percentage drop of p-value $\frac{(-\log(p)\_{\text{RedGreen}}) - (-\log(p)\_{\text{Ours}})}{-\log(p)\_{\text{RedGreen}}}$. This figure illustrates that our watermarks consistently demonstrate higher detection power compared to Red-Green across a range of hashing schemes for generating side information.
>
> ---
> ### **Q3: Beyond Finance-QA (Q&A) and LCC (coding), “what evidence do you have that your watermark performs well in low-entropy scenarios in general?”**
>
> Great question! In **Appendix D** in the Supplementary Material (D.5 Experiment: Alternative Text Generation Metrics), we reported the performance of our watermarks on **four additional datasets** used in WaterBench [41]: LongformQA, Multi-news Summarization, Knowledge Memorization, and Knowledge Understanding. We emphasize that the favorable performance of our watermarking schemes is supported by both our theoretical analysis and our experiments. Next, we summarize the theoretical guarantees and practical evidence that demonstrate the effectiveness of our watermarks in low-entropy scenarios.
>
> **Theoretical Performance Guarantees.** Our theoretical guarantees imply favorable watermarking performance in low-entropy scenarios. Note that, by entropy we mean *min-entropy*: $H= -\log_2 \max_x P(x)$, where $P$ is the next-token distribution. *Low-entropy* refers to next-token distributions whose average entropy $H(P) \leq 1$ bit/token (see lines 166-173 and 223 of the paper). In the low-entropy regime, at least one predicted token has a probability greater than $\frac{1}{2}$.
>
> The ‘low-entropy’ constraint used in our analysis is not merely a theoretical assumption. It is based on the empirical observation that LLMs’ next-token distributions are **inherently low entropy,** as shown in our measurements reported in the Supplementary Material (Appendix C.1). From these measurements (Figures D.14 and D.15  – see also [52, Fig.1] for further evidence), over 85% of next-token predictions satisfy $H(P)\geq 0.95$, and over 90% satisfy $H(P)\geq 0.5$  for practical temperature values. Consequently, the vast majority of next-token predictions fall well within the regime considered in our theoretical analysis – which proves optimality guarantees for $H(P)\geq 0.5$. We observed similar patterns in the additional datasets we tested. **Even though coding, Q\&A, and summarization tasks are commonly understood as having varying degrees of ‘entropy’ – with coding assumed to have lower entropy – we still observe $>90\%$ of next-token predictions across these tasks falling well within the low-entropy considered in our theoretical analysis.**
>
> Our theoretical results give provable performance guarantees for the low-entropy regime. Theorem 1 and Theorem 2 in the main text prove that, in the low-entropy regime, SimplexWater is minimax optimal across all binary score watermarks. In other words, SimplexWater maximizes the binary per-token detection score for the worst-case distribution that satisfies the low-entropy constraint. For real-valued scores and randomly generated score functions, we demonstrate that watermarking in the low-entropy regime depends on the tail of the score distribution. This allows us to generalize the Gumbel Watermark (Theorem 3) and propose HeavyWater – which provably improves upon the Gumbel watermark (Theorem 4).
>
> **From Theory to Practice and Empirical Measurements:** Please note that, in addition to the Q\&A and LCC datasets included in the main paper, we report performance of our watermarks on **4 additional datasets in the Supplementary Material:** longform Q&A, knowledge memorization, knowledge understanding, and multi-news summarization (see Appendix D.5 and Table D.2). Our watermarks demonstrate better or comparable performance across all datasets tested, as reported in Table D.2.
>
> In summary, by focusing our **theoretical analysis on the low-entropy regime** $H(P)\leq 1/2$, we derive **new watermark designs with provable optimality guarantees, namely SimplexWater and HeavyWater**. The low-entropy assumption (lines 166-173) is reflected in practice, with **over 90% of LLM next-token distributions across Q&A, coding, and summarization tasks satisfying this constraint** (Supp. Material Figures D.14 and D.15) per our measurements. HeavyWater and SimplexWater display favorable performance on the LCC and Q\&A datasets used in the main text, as well as on four additional dataset benchmarks reported in our Supplementary Material (Appendix D.5 and Table D.2). The combination of theoretical guarantees and empirical benchmarks together provides significant evidence that our watermarks perform well in low-entropy regimes.

---

> > ### Author Response · Authors · 2025-08-05
> >
> > Thank you again for the comments and helpful feedback! We have submitted a detailed rebuttal addressing the points you raised. We would be grateful if you could let us know whether your concerns have been resolved. We are happy to answer any follow-up questions you may have.

---

> > ### Comment · Reviewer_dQ8C · 2025-08-05
> >
> > Thank you for your helpful response. I have only one comment:
> >
> > Q2: This clarification is helpful. This description needs to appear in the paper, as the current y-axis label "Improvement (%)" is ambiguous.

---

> > > ### Author Response · Authors · 2025-08-05
> > >
> > > Thank you for your positive review! Yes, we will absolutely include the label explanation in the paper.

---

### Official Review · Reviewer_3Nda · 2025-07-03

**Clarity:** 3
**Significance:** 3
**Originality:** 3
**Rating:** 5
**Confidence:** 4

**Summary:**

The paper considers the important problem of LLM watermarking in the case of low-entropy regimes such as in coding tasks. In particular, the question of high detectability with low distortion is tackled and two algorithms are proposed by invoking connections to coding theory and optimal transport. Experiments validate the approaches on various LLMs and settings.

**Questions:**

(1) Is there a performance metric for the quality of the generated code? Are there sample outputs for the coding tasks before and after watermarking?
(2) How easy is it to scrub/spoof the approach? Also, how robust is it various types of attacks such as substitution, deletion, and addition of tokens?

**Ethical Concerns:**

["NO or VERY MINOR ethics concerns only"]

**Final Justification:**

Based on the engagements with the authors as well reviewing the discussions, I maintain my score.

**Limitations:**

Yes

**Quality:**

3

**Strengths And Weaknesses:**

Overall, the paper is very interesting and introduces novel ideas in the watermarking literature.

Pros:

(A) In the case of binary score functions, a connection to the coding theory is made and shown to be minimax optimal. This encompasses the popular red-green settings.
(B) For score functions which are continous, a framework is introduced to tackle this settings and includes the popular Gumbel approach but beats it in performance.
(C) In both settings, they are adequately supported by theoretical results and quite a few experimental results.

Cons:

(i) The watermarking schemes target detectability and low-entropy regimes but it is unclear how they perform along the robustness and security angle.
(ii) The performance metrics are limited to detection and distortion, and downstream utility metrics of the generated code are not presented.

---

> ### Author Rebuttal · Authors · 2025-07-31
>
> Thank you for the thoughtful comments and the positive feedback. Below, we address the four points raised in your review (two weaknesses and two questions). Please note that we included **additional robustness results** for **HeavyWater** and **SimplexWater** in **Appendix D.2** of our Supplementary Material (SM). Our SM also includes additional results on the impact of watermarking on the quality of generated text (**Table D.2** in SM). Finally, we also provide new results on how **HeavyWater** and **SimplexWater** impact generated code. We present the answer to your first question (Q1) at the end of our response, since it includes a code snippet and a table.
>
> We would be happy to answer any additional questions or comments.
>
> ---
>
> ### **W1: Robustness and security** and **Q2: "How easy is it to scrub/spoof the approach? Also, how robust are various types of attacks such as substitution, deletion, and addition of tokens?"**
>
> Thanks for the question! In our SM (**Appendix D.3**), we evaluated the robustness of our watermarks by considering three types of attacks that involve modifying characters or words: **Lowercase attack**, **Misspelling attack**, and **Typo attack**.
>
> We selected these attacks based on the **MarkMyWords benchmark** (reference [42] in the paper). In **Figure D.9**, we report the detection power (as measured by p-value) of **HeavyWater** and **SimplexWater** post-attack, and compare it against the **Gumbel** and **Red-Green** watermarks. These results demonstrate **equal or superior detection** of **HeavyWater** relative to competing baselines under these attacks. We will highlight these results in the main paper.
>
> **Robustness Analysis**
>
> The robustness of most watermarking schemes (including ours) is tied to how random side information is generated (please see the discussion in **Appendix A.1** and **A.3** of the SM, particularly line 1017). Note that both **HeavyWater** and **SimplexWater** are agnostic to side information generation, in the sense that they can be paired with any hashing scheme (e.g., computing S based on hashes of previous tokens and a secret key).
>
> Consequently, we also evaluated the performance of our watermarks under different hashing strategies and reported the results in **Figure 3b** and **Appendix D.2** of the SM (**Figures D.7** and **D.8**). These results show that the performance of our watermark is **consistent across an array of hashing schemes and context window sizes**, including **min-hash**, **sum-hash**, **prod-hash**, and **Markov-1 scheme**.
>
> **Security Guarantees**
>
> Security guarantees of watermarks also stem from the properties of the side information generation procedure (i.e., how $S$ is generated). This is because, under the assumption that the watermarking strategy is public, the primary source of secrecy is the shared information used to generate the side information $S$ (see **Appendix A.1** in SM). Our watermarks can be incorporated with different seeding strategies (see **Appendix D.2**). Consequently, their security guarantees will stem from whichever side generation backbone is being used.
>
> Side information generation can potentially be optimized to prioritize security by leveraging **cryptographic mechanisms** for randomness generation and key sharing. This is an interesting and important direction of future work, and we are aware of recent work from the cryptography community that explores these aspects [56]. In our paper, we focus instead on how to **optimally** use random side information to maximize detection and minimize distortion. We recognize the importance of side-information generation in our limitations section (line 208), and will include this discussion about security in an expanded future work section.
>
> ---
>
> ### **W2: "Lack of performance metric beyond detection and distortion; downstream utility metrics of the generated code are not presented"**
>
> While the main paper focuses on the **detection–distortion tradeoff**, we do report additional generation quality metrics to evaluate the utility of the generated text, as detailed in **Appendix D.5**. In **Table D.2**, we assess the impact of watermarking on generation quality across four additional tasks from **WaterBench [41]**: **LongformQA**, **Multi-news Summarization**, **Knowledge Memorization**, and **Knowledge Understanding**.
>
> For these, we report **ROUGE-L scores** (LongformQA and Multi-news) and **F1 scores** (Knowledge Memorization and Knowledge Understanding). As shown in **Table D.2**, our methods (**SimplexWater** and **HeavyWater**) maintain competitive detection performance across all datasets, with a generation metric **comparable to unwatermarked text**. Together, these results provide significant evidence that our watermarks **preserve output quality** in textual tasks.
>
> We agree that utility metrics should also be emphasized for the code generation tasks used in the main paper. To that end, we are reporting two additional metrics, **Edit Similarity** and **Pass@K**. See our response to Q1 below for detailed information. We will highlight these more clearly in the revised version.
>
> ---
>
> ### **Q1: "Is there a performance metric for the quality of the generated code? Are there sample outputs for the coding tasks before and after watermarking?"**
>
> Thanks for the suggestion! We introduce **two performance metrics** to evaluate the quality of generated code:
>
> #### 1. **Edit_Sim (Edit Similarity)**
>
> We use this metric on the **LCC coding dataset [41]**, which focuses on code completion: generating the next few lines of code given a long context. Since the dataset is sourced from GitHub, human-written ground-truth completions are available. **Edit_Sim** measures **Levenshtein similarity** between generated and ground-truth code. A formal definition will be included in **Appendix D**. We benchmark our watermarking methods against prior work using this metric, and **HeavyWater achieves the highest Edit_Sim score** for the code completion task.
>
> #### 2. **pass@k**
>
> To complement Edit_Sim and better evaluate the successful execution of watermarked code, we also report results on the **HumanEval dataset [44]**, which is a popular benchmark for code generation. Since LCC is not suitable for pass@k (as it does not contain standalone problems and unit tests), we use HumanEval to measure **pass@k with k ∈ {1, 5, 10}**, where we execute the generated code against unit tests provided in the dataset in a sandbox environment. As shown in the results table, **SimplexWater achieves the highest score on pass@5**, while **HeavyWater performs best on pass@10**. In all cases, performance is **comparable to unwatermarked code**.
>
> These two metrics support that **HeavyWater** and **SimplexWater** have **no significant impact on the quality of generated code** relative to unwatermarked code, and perform favorably when compared to competing methods both in terms of detection accuracy and generation quality.
>
> #### **Results on LCC Code Completion Dataset**
>
> | Watermark Method | Edit_Similarity ↑ |
> |------------------|-------------------|
> | No Watermark | 0.45 |
> | Gumbel | 0.52 |
> | **HeavyWater (Ours)** | **0.52** |
> | **SimplexWater (Ours)** | **0.44** |
> | Inverse Transform | 0.45 |
> | Red/Green δ=3 | 0.43 |
>
> #### **Results on HumanEval Functional Coding Dataset**
>
> | Watermark Scheme | pass@1(%) ↑ | pass@5(%) ↑ | pass@10(%) ↑ |
> |------------------|-------------|-------------|---------------|
> | No Watermark | 14.0 | 20.7 | 28.1 |
> | **HeavyWater (Ours)** | **13.1** | **18.9** | **27.8** |
> | Gumbel | 14.3 | 20.5 | 25.6 |
> | **SimplexWater (Ours)** | **13.7** | **22.7** | **25.3** |
> | Inverse Transform | 13.8 | 22.0 | 27.5 |
> | Red/Green δ=3 | 11.6 | 19.5 | 23.2 |
>
> #### **Code Examples**
>
> #### Unwatermarked Output:
>
> ```python
> def has_close_elements(numbers: List[float], threshold: float) -> bool:
>     # Sort the list of numbers
>     numbers = sorted(numbers)
>     # Find the first two numbers in the list that are closer to each other than the threshold
>     min_distance = float('inf')
>     for i in range(1, len(numbers)):
>         distance = abs(numbers[i] - numbers[i - 1])
>         if distance < min_distance and distance < threshold:
>             min_distance = distance
>     # If we found a pair of numbers that are closer to each other than the threshold, return True
>     return min_distance < threshold
> ```
>
> **Unit Tests result:** Pass → **Pass@1=100%** on this example prompt.
>
> #### HeavyWater's Watermarked Output:
>
> ```python
> def has_close_elements(numbers: List[float], threshold: float) -> bool:
>     close_pairs = []
>     for i in range(len(numbers)):
>         closest = None
>         closest_distance = float("inf")
>         for j in range(i+1, len(numbers)):
>             distance = abs(numbers[i] - numbers[j])
>             if distance < closest_distance:
>                 closest = numbers[j]
>                 closest_distance = distance
>         if closest and closest_distance <= threshold:
>             close_pairs.append((numbers[i], closest))
>     return len(close_pairs) > 0
> ```
>
> **Unit Tests result:** Pass → **Pass@1=100%** on this example prompt.

---

> > ### Comment · Reviewer_3Nda · 2025-08-05
> >
> > The authors have addressed my questions and I will keep my score. As a minor comment, the generated code is O(n^2) while the original is O(nlogn). I take it this is due to using a generic LLM and not one optimized for coding tasks.

---

> > > ### Author Response · Authors · 2025-08-05
> > >
> > > Thank you for your thoughtful comment and positive review!
> > >
> > > You're right in noting that the generated algorithms are sometimes not optimized for complexity. This is due to the limitations of the base model (Llama2-7B), which isn't fine-tuned for coding tasks — it's not a consequence of the watermark. In fact, across K=10 generations, we observe both O(n²) and O(n log n) implementations from both the unwatermarked and watermarked models. It just happened that the examples we presented diverged in this regard. For instance, here is another example of a watermarked code using HeavyWater with O(n log n) complexity:
> > >
> > > ```python
> > > def has_close_pairs(numbers, threshold):
> > >     # Sort the list of numbers
> > >     numbers = sorted(numbers)
> > >
> > >     # Check if there are any pairs of numbers closer to each other than the threshold
> > >     for i in range(1, len(numbers)):
> > >         diff = numbers[i] - numbers[i - 1]
> > >         if diff < threshold:
> > >             return True
> > >
> > >     return False
> > > ```
> > > Hope this clarifies your concern, thanks again!

---

### Official Review · Reviewer_udiS · 2025-07-05

**Clarity:** 3
**Significance:** 3
**Originality:** 3
**Rating:** 5
**Confidence:** 4

**Summary:**

This paper proposes a minimax optimization framework for large language model watermarks. It aims at maximizing watermark effectiveness while minimizing text distortion under low-entropy constraints. This work designs two watermark methods, SimplexWater and HeavyWater, derived from the optimization framework. The former is a binary-score watermark that achieves the optimal detection gap, based on the observation that the minimax problem could be formulated as designing codewords with maximal Hamming distances. The latter is a generalized continuous-score watermark that samples watermark scores from heavy-tail distributions to improve watermark detectability. Experimental results show that the proposed methods achieve better performances in terms of effectiveness-distortion trade-off than existing LLM watermarks.

**Questions:**

- Is it possible for an adversary to mount reverse-engineering or spoofing attacks by observing token frequencies of SimplexMark or HeavyMark, especially under the uniform i.i.d. assumption?
- How do SimplexMark and HeavyMark perform (in terms of downstream task metrics) on low-entropy generation scenarios, such as code generation (pass@k) and math reasoning (accuracy)?
- How do the proposed methods perform against more advanced watermark removal attacks?
- Another question following the previous one on robustness: could robustness be potentially integrated into the optimization formulation as an objective?
- Could this optimization framework or the two proposed methods be potentially extended to multi-bit watermarking?

A few other minor issues:

1. Markov-1, Product, h=4, Sum, h=5 in Figure 3(b) is not defined in the main text (although the definitions could be found in the appendices).
2. Some figure titles seem to be inconsistent. E.g., in Figure D.13, do "Coding" and "LCC" refer to the same dataset (LCC)?

**Ethical Concerns:**

["NO or VERY MINOR ethics concerns only"]

**Final Justification:**

I would like to thank the authors for their detailed response and clarifications, including the additional results on low-entropy generation, the clarification on spoofing attacks, as well as the sketches on incorporating attack channels and multi-bit watermarking. The responses have addressed most of the concerns.

**Limitations:**

Yes.

**Paper Formatting Concerns:**

None.

**Quality:**

3

**Strengths And Weaknesses:**

**Strengths**

+ This paper proposes a unified optimization framework for LLM watermarks. The optimization framework includes watermark effectiveness, text distortion, and low-entropy constraints, and would help provide theoretical insights into LLM watermarks.
+ This paper further proposes two concrete watermark implementations under the proposed framework. Both watermarks are distortion-free and could work under different settings.
+ Apart from empirical results, the two implementations also contribute to additional theoretical insights: SimplexWater establishes connections between the watermark framework and coding theory; HeavyWater shows that continuous-score watermarks could benefit from sampling from heavy-tail distributions.

**Weaknesses**

- **The i.i.d. requirement for the side information might be restricted in real-world scenarios**. It would add to the difficulties of sharing the secret information between the watermarker and the detector, and might lead to vulnerabilities against reverse engineering. For example, the authors mention in the appendix that a counter and a secret key are used to generate the i.i.d. side information. In this case, the watermark behavior is only tied to the i-th token's position (i.e., the counter value). Each token at the same location would have similar watermark behavior, regardless of the context. Consequently, the watermark might be vulnerable to reverse engineering, where an adversary queries the watermarker multiple times and counts word frequencies at each position to infer high-score watermark tokens. Nonetheless, the authors do mention this in the limitation section and include results using more realistic non-i.i.d. side information. While the empirical results are promising, some of the theoretical analysis might not hold in non-i.i.d. cases.
- **Additional tasks and metrics for low-entropy generation could be included to further evaluate generation quality**. While this paper features low-entropy watermarking, the evaluation for low-entropy generation is limited to code generation (on the LCC dataset), and the main performance metric for generation quality is limited to the estimated Cross Entropy. While indeed additional metrics are included in the appendices, they mostly focus on general text generation tasks rather than low-entropy ones. Hence, it would be more convincing if the evaluation could be extended to include more low-entropy tasks and metrics (e.g., pass@k on code generation or accuracy on math reasoning, etc.).
- **The robustness evaluation does not consider stronger removal attacks**. The watermark removal attacks have mainly focused on rather primitive attacks, such as changing cases and adding typos. Other attacks, such as word substitution, copy-paste attacks, or paraphrasing, could also be included for a more comprehensive evaluation. This is a rather minor issue since robustness is not the primary focus of this work.

---

> ### Author Rebuttal · Authors · 2025-07-31
>
> Thank you for your comments and positive review!  We address your questions next and highlight additional results in the Supp. Material (SM). Please note that we included in Appendix D.2 experiments on the impact of non-i.i.d. side information on SimplexWater and HeavyWater, additional robustness results in Appendix D.3, and reported alternative performance metrics in Appendix D.5. We also provide new results on code generation metrics below.
>
> We are happy to answer any follow-up questions you may have!
>
> ---
> ### **W1, Q1:The i.i.d. requirement for side information restricted (…) While empirical results are promising, theoretical analysis might not hold in non-i.i.d. cases.  Is it possible for an adversary to mount reverse-engineering or spoofing attacks (..) under uniform i.i.d. assumption?**
>
> Thanks for bringing this up. Our theoretical results indeed assume i.i.d. side information, as mentioned in lines 208-218. Importantly, we address this issue head-on by evaluating our watermarks’ performance against several methods for non-i.i.d. side information generation in App. D.2.
>
> In App. D.2, we benchmark different side information generation methods based on hashing previous tokens – min-hash, sum-hash, prod-hash, and Markov-1 – extending beyond counter-based approaches. As shown in Fig. 3(b), both SimplexWater and HeavyWater outperform Red-Green in terms of detection under a range of non-i.i.d. side information generation methods. Fig. D.7 in the SM indicates that varying the number of preceding tokens used to generate side information has minimal effect on both text quality and detection of our watermarks. Figure D.8 shows that both Markov-1 and prod-hash-4 achieve detection performance very close to that of i.i.d. (fresh) randomness. These results show that SimplexWater and HeavyWater maintain detection accuracy and text quality under non-i.i.d. side information, comparable to the i.i.d. setting.
>
> Regarding the question on the possibility of adversarial attacks under i.i.d and non-i.i.d side information: if the side information $S$ is uniform i.i.d. and generated fresh for each token and across prompts, then, it is **not** possible for an adversary to mount any reverse-engineering or spoofing attack. Here’s why: the optimal transport (OT) optimization that is solved for each token results in a coupling between the (uniform) side information distribution $P_S$ and the next token predictions $P_X$. This coupling *ensures* that the marginal distribution of next-token predictions **remains exactly the same**. Consequently, an adversary that only observes tokens $X_1,\dots,X_n$ will not observe any statistical deviation from the original model’s distribution since, on average across $S$, samples are still drawn from $P_X$. Thus, it is **information theoretically impossible** for an adversary (no access to $S$) to distinguish between the unwatermarked and watermarked distribution.
>
> This breaks under non-i.i.d. Assumptions. As the reviewer points out, in the extreme non-i.i.d. case where $S$ is deterministic for a given token position, an adversary could spoof the watermark by querying the model multiple times.
>
> Current watermarks aim for an intermediate solution. For instance, side information is generated using previous tokens, secret keys, and hashing (potentially with salt), resulting in side information that is non-i.i.d., but not deterministic. Non-i.i.d side information generation impacts the robustness and security of watermarking schemes – not only ours, but also of all other competing methods, such as SynthID, Gumbel, Inverse Transform, and Red-Green. This was one of the main motivations in our choice to taxonomize watermarking into three different components in lines 130–149, namely (i) randomness generation, (ii) randomness use, and (iii) watermark detection, and optimize for the last two components assuming ideal randomness generation. This assumption enables a tractable theoretical analysis, leading to new information-theoretic performance bounds (Thms 1, 2, and 4).
>
> ---
> ### **W2, Q2: Additional tasks and metrics for low-entropy generation could be included.**
>
> Thanks for pointing this out. We do have additional text quality evaluation metrics in **Appendix D.5** and **Table D.2**, where we consider four additional datasets from **WaterBench [41]** along with either **ROUGE-L and F1 scores for textual quality**. These results show that our watermarks maintain strong detection performance without compromising text generation quality.
>
> During the rebuttal, we also computed additional generation metrics targeted towards code generation tasks (reported below). We will highlight these results in the main paper.
>
> **New experiments:** Below, we report additional tasks and metrics for low-entropy generation in coding. We include a new coding experiment using the **HumanEval dataset [44]** and evaluate it with **pass@K with k=1,5,10** generation metric, using Llama2-7B as the baseline. For LCC code completion tasks, since human-written codes are used as the ground truth, we use Edit_Similarity as the generation-quality metric. Under both Edit_sim and pass@k, our methods have no significant impact on the quality of generated code relative to unwatermarked code, and perform favorably when compared to competing methods. Below, we give two tables with the results.
>
> **Task 1: Code generation - Generating code for a given task using HumanEval dataset and evaluate with pass@k**
>
> #### **Results on HumanEval Functional Coding Dataset**
> | Watermark Scheme           | pass@1(%) **↑** | pass@5(%) **↑** | pass@10(%) **↑** |
> |----------------------------|------------------|------------------|-------------------|
> | No Watermark               | 14.0             | 20.7             | 28.1              |
> | **HeavyWater (Ours)**      | **13.1**         | **18.9**         | **27.8**          |
> | Gumbel                     | 14.3             | 20.5             | 25.6              |
> | **SimplexWater (Ours)**    | **13.7**         | **22.7**         | **25.3**          |
> | Inverse Transform          | 13.8             | 22.0             | 27.5              |
> | Red/Green $\delta=3$       | 11.6             | 19.5             | 23.2              |
>
>
> **Task 2: Code completion - Auto-complete a given code snippet using LCC dataset and evaluate with Edit_Sim**
>
> #### **Results on LCC Code Completion Dataset**
> | Watermark Method       | Edit_Similarity **↑** |
> |------------------------|----------|
> | No Watermark          | 0.45     |
> | Gumbel                 | 0.52     |
> | **HeavyWater (Ours)**              | **0.52**     |
> | **SimplexWater (Ours)**            | **0.44**     |
> | Inverse Transform                 | 0.45     |
> | Red/Green $\delta=3$     | 0.43     |
>
> ---
> ### **W3, Q3, Q4: More advanced watermark removal attacks. Integrate robustness into the optimization objective.**
>
> Currently, we have tested our watermarks on misspelling attacks, typo attacks and case-change attacks in Appendix D.3, benchmarking their performance against competing watermarks with comparable detection rates. We select these attacks since they are the standard in the MarkMyWords benchmark [42]. Fig. D.9 shows that our methods result in superior/comparable performance with respect to existing watermarks under the above attacks.
>
> Although robustness is not the main focus of this work, we consider the theoretical analysis of watermark robustness – particularly under complex attacks such as copy-paste and paraphrasing – to be an important future direction which we are actively exploring. We can incorporate robustness into our framework by modeling edits and other perturbations explicitly. Right now, we treat the problem as finding an optimal tilted next-token distribution $P_{X|S}$​, considering the worst-case $P_X$. To capture robustness, we can introduce an additional channel $P_{Y|X}$​ that describes how an adversary might transform each token X into a perturbed token Y (e.g., typos, paraphrasing, etc). With a suitable model for $P_{Y|X}$​, we can then choose $P_{X|S}​$ to maximize the worst case  $E_{P_{X,S}} ⁣[f(Y,S)]−E_{P_XP_S} ⁣[f(Y,S)]$. This formulation would allow us to optimize the watermark while explicitly accounting for robustness under perturbations, albeit at the cost of a more involved mathematical analysis.
>
> ---
> ### **Q5: Extend the optimization framework to multi-bit watermarking**
>
> Yes – this is an area that we are also exploring. A natural way to extend the proposed watermarks from single-bit to multi-bit is to assign random, independent side information strings to each of the $n$ messages. Denote the string corresponding to message $i$ as $S_i=[S_i^{[1]},...,S_i^{[k]}]$. To embed $S_1$, the watermarker applies SimplexWater/HeavyWater by using $S_1^{[t]}$ as the side information at token instance $t$. Note that the marginal distribution of the next token remains the same as the (original) next token distribution of the LLM as each $S_1^{[t]}$ is sampled independently at random, which maintains the distortion-free property. Based on the $f(X,S_1^{[t]})$ scores, the detector arrives at a high true positive rate (for detecting $S_1$) as a consequence of the high detection performance of SimplexWater/HeavyWater. The false positive rate remains small (assuming that n is not significantly large) as each $S_j$ is independent of S_1, which results in smaller scores for $f(X,S_j)$ for all $j\neq1$. The gain in false positives due to multi-bit watermarking can be bounded by applying a union bound to our existing results.
>
> ---
> ### **Q6: Minor issues.**
>
> Thanks for the helpful feedback. We will bring the definitions to the main text from App. D.2. Yes, “Coding” and “LCC” refer to the same dataset. We will clarify this in the revised version.

---

> > ### Author Response · Authors · 2025-08-05
> >
> > Thank you again for the comments and helpful feedback! We have submitted a detailed rebuttal addressing the points you raised. We would be grateful if you could let us know whether your concerns have been resolved. We are happy to answer any follow-up questions you may have.

---

> > ### Comment · Reviewer_udiS · 2025-08-08
> >
> > I would like to thank the authors for their detailed response and clarifications, including the additional results on low-entropy generation, the clarification on spoofing attacks, as well as the sketches on incorporating attack channels and multi-bit watermarking. The responses have addressed most of the concerns.
> >
> > One minor note is that for robustness evaluation, word substitution (or addition/deletion) and paraphrasing are also common attacks evaluated in previous established works (e.g., Red-Green and SynthID), and therefore, including these attacks would make the comparison more direct. Nonetheless, this reviewer understands the author(s)'s choice to use attacks from other established benchmarks, given that the current framework does not explicitly consider robustness. The idea of including attack channels or multi-bit formulation into the optimization framework would be inspiring, although a formal analysis could be challenging given the intricacy of the existing framework.
> >
> > Since the primary concerns on low-entropy generation and spoofing attacks have been resolved with additional results and/or analysis, this reviewer would like to increase the score to 5 (Accept).

---

### Official Review · Reviewer_1jKu · 2025-07-06

**Clarity:** 3
**Significance:** 3
**Originality:** 3
**Rating:** 3
**Confidence:** 3

**Summary:**

his paper proposes two novel watermarking techniques, HeavyWater and SimplexWater, to address the challenges of reliably detecting machine-generated text from large language models (LLMs), particularly in low-entropy scenarios like code generation. HeavyWater employs heavy-tailed score distributions to enhance detectability while minimizing distortion, while SimplexWater leverages simplex codes from coding theory to maximize the Hamming distance between watermarked tokens, proving optimal for binary score functions. The authors frame watermark design as an optimization problem, linking it to optimal transport theory, and demonstrate that their methods outperform existing approaches (e.g., Gumbel, Red-Green watermarks) in detection accuracy and robustness, especially for low-entropy text.

**Questions:**

How is the code pass@k in the experiments?

**Ethical Concerns:**

["NO or VERY MINOR ethics concerns only"]

**Limitations:**

yes

**Quality:**

3

**Strengths And Weaknesses:**

Strengths:
1. The paper presents a rigorous and thorough theoretical analysis.
2.  The paper extend binary score system to a continuous distribution for better performance.

Weaknesses:
1. The baselines listed lack some previous works that are specially designed to tackle low-entropy scenarios.
2. The paper could demonstrate the entropy difference between the used datasets and high-entropy datasets such as C4 for better comprehension. And shouldn't the models include coding LLMs such as starcoder?
3. The experiments should also include performance of high-entropy scenarios.

---

> ### Author Rebuttal · Authors · 2025-07-31
>
> Thank you for your review. We are glad that you appreciated the rigorous and thorough theoretical analysis. Below, we address the three points that you raised. Please note that we provide new results on how SimplexWater and HeavyWater impact code generation accuracy, namely Edit_Similarity for code completion and pass@k for code generation. We would be happy to answer any follow-up questions or comments you may have.
>
> ### **Q1:How is the code pass@k in the experiments?**
> ---
> That’s a great question! To better emphasize quality of generated watermarked code, **we include two coding-centric evaluation metrics** that assess different aspects of code quality.
>
> **Pass@K Evaluation (Functional Correctness)**: We evaluate functional correctness using the HumanEval dataset [44], which is a popular benchmark for code generation. Pass@K measures the empirical probability of a solution passing all unit tests among K generated solutions. We measure **pass@k with k ∈ {1, 5, 10}**, where we execute the generated code against unit tests provided in the dataset in a sandbox environment. As shown in the table below, our methods outperformed the competitors: (i) **SimplexWater achieves the highest pass@5 (22.7%)**, outperforming all baselines including unwatermarked text (20.7%) and (ii) **HeavyWater achieves the highest pass@10 (27.8%)**, nearly matching unwatermarked performance (28.1%). In all cases, performance is comparable to unwatermarked code.
>
> **Edit_Sim (Edit Similarity)**: We use this metric on the **LCC coding dataset** [41] which focuses on code completion: generating the next few lines of code given a long context. Since the dataset is sourced from GitHub, human-written ground-truth completions are available. Edit_Sim measures Levenshtein similarity between generated and ground-truth code. A formal definition will be included in Appendix D. We benchmark our watermarking methods against prior work using this metric. The results are provided in the table below. Specifically, (i) **HeavyWater ties for best performance (0.52)**.
>
> **Key Insights**: These results demonstrate that our watermarks preserve both functional correctness and syntactic quality of generated code. The fact that SimplexWater and HeavyWater often match or exceed unwatermarked performance on these metrics, while providing strong watermark detection, highlights the effectiveness of our approach in maintaining code quality during watermarking.
>
> #### **Results on HumanEval Functional Coding Dataset**
>
> | Watermark Scheme           | pass@1(%) **↑** | pass@5(%) **↑** | pass@10(%) **↑** |
> |----------------------------|------------------|------------------|-------------------|
> | No Watermark               | 14.0             | 20.7             | 28.1              |
> | **HeavyWater (Ours)**      | **13.1**         | **18.9**         | **27.8**          |
> | Gumbel                     | 14.3             | 20.5             | 25.6              |
> | **SimplexWater (Ours)**    | **13.7**         | **22.7**         | **25.3**          |
> | Inverse Transform          | 13.8             | 22.0             | 27.5              |
> | Red/Green $\delta=3$       | 11.6             | 19.5             | 23.2              |
>
>
> #### **Results on LCC Code Completion Dataset**
> | Watermark Method           | Edit_Similarity **↑** |
> |---------------------------|------------------------|
> | No Watermark              | 0.45                   |
> | Gumbel                    | 0.52                   |
> | **HeavyWater (Ours)**     | **0.52**               |
> | **SimplexWater (Ours)**   | **0.44**               |
> | Inverse Transform         | 0.45                   |
> | Red/Green $\delta=3$      | 0.43                   |
>
> ---
> ### **W1: Lack of considering previous works that are designed to tackle low-entropy scenarios.**
>
> Please note that we compare our watermarks qualitatively and/or quantitatively against **several** methods that are designed to watermark challenging generation processes that are ``low-entropy,’’ such as 1) code completion using the LCC dataset (see fig. D.13 in App. D.7), 2) Question answering using the Longform QA dataset (Table D.2), 3) Factual recall using the knowledge memorization dataset (Table D.2). We also acknowledge that low-entropy watermarking is an active field of research, discussing prior work in the field in Section 1 (see lines 105-116), as well in the Supplementary Material (see the related work discussion in Appendix A).  **We would be happy to include additional references you may have!**
>
> In our work, we define the “low-entropy regime” in the very concrete and tangible sense described in lines 166-173 and 223 of the paper. Specifically, we adopt min-entropy as the entropy measure of choice:  $H(P) = -\log_2 \max_x P(x)$, where $P$ is the next-token distribution. Here, *low-entropy* refers to next-token distributions whose average entropy $H(P) \leq 1$ bit/token (see lines 166-173 and 223) — i.e. **at least one predicted token has a probability greater than $\frac{1}{2}$** (lines 166-173). While the literature acknowledges challenges in watermarking low-entropy distributions in general, to the best of our knowledge, this specific definition has only appeared in [52] and [R1]. We cite the former in the paper and will include a discussion of the latter as well.
>
> The ‘low-entropy’ regime considered in our analysis is not merely a theoretical assumption. It is based on the empirical observation that LLMs’ next-token distributions are **inherently low entropy** in the sense given above. This is supported by our measurements reported in the Supplementary Material (see Appendix C.1). From these measurements (Figures D.14 and D.15  – see also [52, Fig.1] for further evidence), over 85% of next-token predictions satisfy $H(P)\geq 0.95$, and over 90% satisfy $H(P)\geq 0.5$ for practical temperature values. Therefore, the vast majority of next-token predictions fall well within the regime considered in our theoretical analysis – which provides precise optimality guarantees for $H(P)\geq 0.5$. We observed similar patterns in the additional datasets we tested. Our theoretical results yield a new theory and interesting new connections between LLM watermarking and coding theory.  Importantly, it also yields provable performance guarantees for our watermarks (HeavyWater and SImplixWater) in the low-entropy regime (Theorem 1-4). These results are novel.
>
> **Justification of Benchmark Selection and Experimental Comparisons.** Our experimental evaluation was chosen based on established evaluation benchmarks such as WaterBench [41]  and MarkMyWords [42], which represent standard and well-known benchmarks in the LLM watermarking literature that ensure comparability among different methods. We evaluate our method on (1) code generation and completion (LCC dataset), (2) financial Q&A and (3) four additional tasks from WaterBench covering knowledge-intensive scenarios (Appendix D.5), which are tasks that fall within the low-entropy regime covered by our theoretical analysis. Moreover, we would like to emphasize that our theoretical analysis covers most next-token distributions in all tasks that we tested, since the min-entropy below 1 bit regime is actually most common as far as we can tell.
>
> ---
> ### **W2+W3: Entropy difference between datasets, including high entropy scenarios, including the C4 dataset and the coding LLMs**
>
> Thank you for this comment. We would like to point out that, in addition to the code completion and QA datasets presented in the main text, we provide results for 4 additional datasets in the appendix. As detailed in **Appendix D.5**. In **Table D.2**, we assess the impact of watermarking on generation quality across four additional tasks from **WaterBench** [41]: **LongformQA**, **Multi-news Summarization**, **Knowledge Memorization**, and **Knowledge Understanding**. For these, we report **ROUGE-L** scores (LongformQA and Multi-news) and **F1** scores (Knowledge Memorization and Knowledge Understanding). As shown in **Table D.2**, our methods (SimplexWater and HeavyWater) maintain competitive detection performance across all datasets, with a generation metric comparable to unwatermarked text. Together, these datasets demonstrate the effect of our watermark on presumably varying levels of entropy scenarios.
>
>  **Empirically, even though coding, Q\&A, and summarization tasks are commonly understood as having varying degrees of ‘entropy’, we still observe $>90\%$ of next-token predictions across all of these tasks falling well within the low-entropy considered in our theoretical analysis.** Again, recall that by ``low-entropy regime’’ we consider next-token distributions with min-entropy below 1 bit. In other words, though coding has lower entropy and tasks like Q&A have higher entropy (see Figures D.14 and D.15 for measurements), they both fall well within the regime for which our performance guarantees hold.
>
> Finally, we note that the C4 dataset is a text corpus. Most existing works consider standard prompt datasets, such as the six datasets used in our paper. The six datasets we used to benchmark our results are derived from standard watermark benchmarks, namely four datasets used in the WaterBench [41] and two datasets (FinanceQA [43] and LCC [44]).
>
> Regarding the choice of LLM, we mirrored choices made in competing watermarks that we benchmarked against, incl. Inverse Transform, SynthID, and Correlated Channel (see refs. [11,13,16] in the paper) -  namely, using Llama and Mistral models – allowing us to compare our methods with previously published benchmarks. We made this choice for replicability purposes and so our findings can be compared with existing published work. Note that our watermarks target all application scenarios, so we did not adopt LLMs designed specifically for coding.
>
> ---
> Reference:
> [R1] Fairoze, Jaiden, et al. "Publicly-Detectable Watermarking for Language Models." IACR Communications in Cryptology 1.4 (2025).

---

> > ### Author Response · Authors · 2025-08-05
> >
> > Thank you again for the comments and helpful feedback! We have submitted a detailed rebuttal addressing the points you raised. We would be grateful if you could let us know whether your concerns have been resolved. We are happy to answer any follow-up questions you may have.

---

### Note · Authors · 2025-08-13

We sincerely thank the reviewers and Area Chair for their time and effort.

Our paper introduces a novel optimization framework for designing LLM watermarks. This framework leads to two new watermarking schemes: **SimplexWater**, which achieves provable minimax optimality when watermark detection uses binary scores, and **HeavyWater**, which improves and generalizes the popular Gumbel watermark. Our theoretical contributions include formal optimality guarantees in the low-entropy regime (i.e., next-token predictions have min-entropy below 1 bit, where over 90% of LLM predictions occur) and new connections between watermarking, optimal transport, and error-correcting codes. Empirically, our methods achieve higher watermark detection rates relative to existing approaches (including Gumbel, Red-Green, and SynthID) on tasks such as coding and Q&A, while maintaining generation quality as measured by cross-entropy, pass@k, Edit_Similarity, and ROUGE scores. We benchmark our watermarks using multiple models and standard datasets drawn from previous work, such as WaterBench [41] and MarkMyWords [42].

We are pleased that reviewers **dQ8C**, **udiS**, and **3Nda** recommend acceptance. They engaged with us during the discussion period and confirmed that our rebuttal addressed their concerns. We are grateful for their dedication, effort, and constructive feedback throughout this process, which have helped us improve our paper. We will incorporate their suggestions into our camera-ready version, as noted in our discussion below.

At the time of posting, reviewer **1jKu** has kept their original weak reject score. Reviewer **1jKu** did not participate in the discussion period nor provide any additional comment beyond their initial short review, despite our detailed response and request for clarifications (e.g., they state “baselines lack some previous work” without providing specific citations). The only concrete question they raised was about pass@k performance, which we addressed in our rebuttal.

We kindly ask the AC to weigh the consensus formed among reviewers **dQ8C**, **udiS**, and **3Nda** – all of whom provided detailed reviews, participated in the author-reviewer discussion, and confirmed their concerns were resolved – when making their final assessment. We included additional information in the AC-only comments below.

Thank you so much for your support.

---

### Decision · Program_Chairs · 2025-09-17

**Decision:**

Accept (poster)

**Comment:**

This paper designs LLM watermarks for challenging low-entropy text by framing the task as an optimization problem. The goal is to maximize the watermark's detection signal by correlating the output with random side information, while simultaneously minimizing distortion to the original text. This framework yields two new, tunable watermarks: SimplexWater, which uses a binary score function rooted in coding theory, and HeavyWater, which uses a more flexible continuous score. Through both theoretical analysis and empirical experiments, the paper shows these methods achieve better detectability compared to existing schemes, particularly in low-entropy scenarios like code generation.

The paper's primary strength is how it nicely formalizes the problem of constructing a good watermark as an optimization task. This perspective is not only very interesting, but it also generalizes and improves upon popular methods. Solving this optimization problem yields two practical watermarks, HeavyWater and SimplexWater. The effectiveness of these methods is supported both in theory and in experiments. The latter was made stronger in the rebuttal when the authors added results on key downstream metrics like pass@k, demonstrating that the watermarks preserve the functional quality of generated code.

The most significant initial concerns centered on computational efficiency and the practical impact on generated code. However, the authors effectively addressed these by clarifying that the per-token optimization overhead is constant and becomes negligible for large models. The paper's focus is on detectability and quality and lacking on robustness to attacks, which is okay but something that should ideally be analyzed in a watermarking paper. Simple attacks like typos are studied, but not more sophisticated ones like rewrites or word subsitutions. Similarly, the theoretical analysis relies on an idealized i.i.d. assumption for side information, though the authors' experiments show the methods still perform well with more realistic randomness.